# Techno-functional and 3D shape analysis applied for investigating the variability of backed tools in the Late Middle Paleolithic of Central Europe

Davide Delpiano[1]*, Thorsten Uthmeier[2]*

**1** Dipartimento di Studi Umanistici, Sezione di Scienze Preistoriche e Antropologiche, Università degli Studi di Ferrara, Ferrara, Italy, **2** Department of the Old World and Asian Studies, Friedrich-Alexander-University of Erlangen-Nürnberg, Erlangen, Germany

* davide.delpiano@unife.it (DD); thorsten.uthmeier@fau.de (TU)

**Data Availability Statement:** All relevant data are within the paper and its Supporting Information files.

## Abstract

In the Late Middle Paleolithic of Central Europe, two main cultural complexes have been distinguished: the Micoquian or *Keilmessergruppe* (KMG), and the Mousterian. Their differences mainly consist in the frequence of some retouched tools and the presence of bifacial technology. When these industries coexist, one element of discussion is the application of different concepts to manufacture tools with the same techno-functionality. This is particularly true for backed artifacts, such as Keilmesser (backed, asymmetrical bifacially-shaped knives) opposed to flake-tools equipped with a natural or knapped back. We conducted a techno-functional analysis of the backed tools from the G-Layer-Complex of Sesselfelsgrotte, one of the main Late Middle Paleolithic sequences in Central Europe, characterized by a combination of KMG and Mousterian aspects. In order to better understand the morpho-metrical data, 3D scans were used for recording technical features and performing semi-automatic geometric morphometrics. Results indicate that the techno-functional schemes of Keilmesser show a moderate variability and often overlap with the schemes of other typological groups. Within bifacial backed knives, a process of imitation of unifacial flake tools' functionaly was recognized particularly in the cutting edge manufacturing. Keilmesser proved to be the long-life, versatile version of backed flake-tools, also due to the recurrent valence as both tool and core. This is why Keilmesser represent an ideal strategic blank when a mobile and multi-functional tool is needed. Based on these data, it is assumed that the relationship between Mousterian and KMG is deeply rooted and the emergence of KMG aspects could be related to constrained situations characterizing the long cold stages of the Early Weichselian. A higher regional mobility caused by the comparably low predictability of resources characterized the subsistence tactics of Neanderthal groups especially at the borders of their overall distribution. For this reason, Keilmesser could have represented an ecological answer before possibly becoming a marker of cultural identity.

**Funding:** The doctoral studies and the mobility of Davide Delpiano were supported by a scholarship funded by the Italian Ministry of Education, University and Research and based at the University of Ferrara. The authors acknowledge support by the Open Access Publishing Fund of Friedrich-Alexander-Universität University. The funders had no role in study design, data collection and analysis, decision to publish, or preparation of the manuscript.

**Competing interests:** The authors have declared that no competing interests exist.

# 1. Introduction: Defining Mousterian and Micoquian dichotomy in Central Europe

The technological variability through time and space that characterizes the 250,000 years of occupation of Central Europe by Neanderthal groups is a direct evidence for the plasticity of their behavioral strategies, which do not appear to have been static, but, on the contrary, complex and changeable, particularly during the last phase of their presence in Europe [1,2].

Numerous analysis of Middle Paleolithic innovations suggest that they were triggered by a complex combination of several factors, which have been integrated in explanatory models for the variability in some of the major techno-complexes within the Middle Paleolithic contexts [3–9]. The ecosystem in which these groups interacted, the major activities carried out with the knapped tools at different sites within the territories used, the cultural background that some groups shared: each one of these interdependent elements is a single component of a complex ensemble that may have contributed to common changing (and innovation) processes. Similarly, social factors, such as group size, the density of social networks or the long-term transfer of information, should not be omitted [10].

The two main lithic industries that are until today dominating the discussion about the Late Middle Paleolithic in Central Europe are the Mousterian and the Micoquian [11,12]. It was long thought that these two entities, which are in turn fragmented into further techno-complexes characterized by different technological concepts and/or tool types, correspond to distinct Neanderthal populations, at least in terms of technical behavior [12–14]. However, the specific definitions of the techno-complexes involved, as well as their chronological frameworks, were biased due to different histories of studies, terminological misunderstandings, and lack of reliable chronostratigraphic contexts. In the recent past, it became more and more difficult to explain the bonds recognized within this presumed dichotomy, and consequently to quantify how much of those are rooted in the cultural sphere or, on the other hand, in the behavioral economy [15].

The Micoquian concept itself incorporates different meanings: the Micoquian *sensu lato* is a Middle Paleolithic industry whose lithic assemblages are characterized by asymmetrical or elongated bifacial tools with concave edges [14,16]. This Micoquian s.l. was defined by Otto Hauser based on layer H, complex VI at La Micoque [17]; the currently unknown stratigraphic position and the distribution of the original material on numerous collections make a modern and general analysis difficult. According to a recent reassessment [17], Micoquian-type bifacials are indeed present in that layer and occur together with Levallois artifacts, but the questionable chrono-stratigraphy casts doubt on any large-scale hypothesis derived from the type site assemblage. The Micoquian s.l. at that time was considered as a late Lower Paleolithic industry, chronologically positioned between Acheulean and Mousterian.

Gerhard Bosinski linked the Micoquian to a series of Middle Paleolithic lithic industries typical of Central Europe characterized by different bifacial tool types and dated to the Eemain and the Early Weichselian (MIS 5) [12]. He developed a qualitative morphological approach by using presence or absence and typical combination of *fossile directeurs* to create *formengruppen*; the morphological variability of a *fossil directeur* within a formengruppe may lead to a further subdivision into *inventory types*. For the Micoquian, the most indicative *fossil directeur* was the asymmetrical backed biface or keilmesser (KM). Afterwards, this category was enlarged by Stephan Veil who included bifacial leaf-shaped tools in the typical Micoquian toolkit [18], and A. Pastoors [19], who could show that large handaxes are also an integral part of the Micoquian. Despite the fact that these—and other tool types such as small handaxes or *groszak* [9]- also occur frequently in the respective assemblages, the common denominator of Central European Micoquian is still the Keilmesser. In addition to the term "Micoquian",

different alternative terms have been proposed for assemblages from Central Europe, mainly stressing a strong correlation with some bifacial tool forms, e.g. Micoquo-Pradnikian or Asymmetrical Knives Assemblages (AKA) [20,21] or the most common *Keilmessergruppen* [18,22,23].

The chronological framing of the *Keilmessergruppen* in Central Europe is still debated. According to few debated Polish sites (mainly Dzierzyslaw I and Bisnik cave), some isolated Micoquian features may appear as early as MIS 6 [14,24,25]. However, at the moment two main schools of thought exist. Based on the stratigraphies of German sites (mainly Königsaue, Buhlen/Oberer Fundplatz, Balver Höhle, Neumark Nord-2), the "long chronology" postulates the presence of the Central European Micoquian during the entire first part of the last glacial cycle (starting after Eemian, e.g. from MIS 5d to MIS 3) [26]. To the contrary, a "short chronology" restricted to end of MIS 4 and the first half of MIS 3, based on the assumption that assemblages of *Keilmessergruppen* are confined to cold environments occuring in Central Europe not earlier than MIS 4, has been then proposed for the entire Central European *Keilmessergruppen* [9,27,28]. Sites central to the arguments for a "long chronolgy", such as the Balver Höhle, are still under debate between those who favor the first [29] or the second hypothesis [9,27] for the integrity of their stratigraphical sequences. Equally disputed are a number of supposingly early sites like Wylotne and Zwierzyniek [30] as well as those of Zwolén [31] and German sites of the Ruhr region [32]; the latter both represent fluvial archives with problematic site formation processes. Based on geological data, Königsaue was originally dated to MIS 5a [33], but direct AMS-dates on resin [34], and subsequently on bone [35] from the older layer A, has shifted the attribution to MIS 3 (44.5–46 ky). Recently obtained absolute dates (93±7 ky) from Neumark Nord 2/0 are in good agreement with the environmental studies and support an onset of the *Keilmessergruppen* during MIS 5b-c [36] and, therefore, give new arguments for the "long chronology". Both the two models agree that a large number of sites from secure stratigraphical contexts and with reliable absolute dates fall into a period between the end of MIS 4 and the first part of MIS 3. Among these sites are Sesselfelsgrotte [37,38], Pouch [39], Verpillière I and II [40], Ciemna and Oblazowa [41,42], Pietraszyn 49a [43], Wroclaw-Hallera Avenue [44] and Kůlna [45]. Other contexts with dates falling into MIS 3 age are Salzgitter-Lebenstedt [19,46] and Lichtenberg [18], however characterized by complex open-air sequences or unsecure association between dates and human occupation [26] (Fig 1).

Depending on the study approaches and analytical methodologies, the relationship between the *Keilmessergruppen* on the one hand, and the Mousterian on the other, has been considered with different points of view: two independent cultural units differentiated on a chronological base [12], in which Micoquian elements develop directly from late Acheulean assemblages [21], or two cultures characterized by a parallel development (*evolution buissonant*) [14], up to the recognition of a deep functional interrelation between both entities, expressed by the term "Mousterian with Micoquian Option (M.M.O.)" [9], which will be used in this paper as well. According to this interpretation, Keilmesser and, more broadly speaking, the Micoquian bifacial tool technology, is a specific strategy to prolongate the use life of lithics by resharpening, which is at the same time applied within particular functional- or seasonal-related circumstances. The fact that the Micoquian bifacial tool technology is combined with different technological concepts for the manufacture of unifacial tools also typical for the Mousterian, like the Levallois, Quina and/or Discoid core reduction concepts, questions the significance of Micoquian features alone as independent markers of Late Middle Paleolithic entities [47].

The flake-débitage technologies, the typology of the unifacial flake-tools and bifacial tools have always been considered the main variables in any differentation between the Mousterian and the *Keilmessergruppen*/M.M.O.. However, for a long period most attempts to identify and

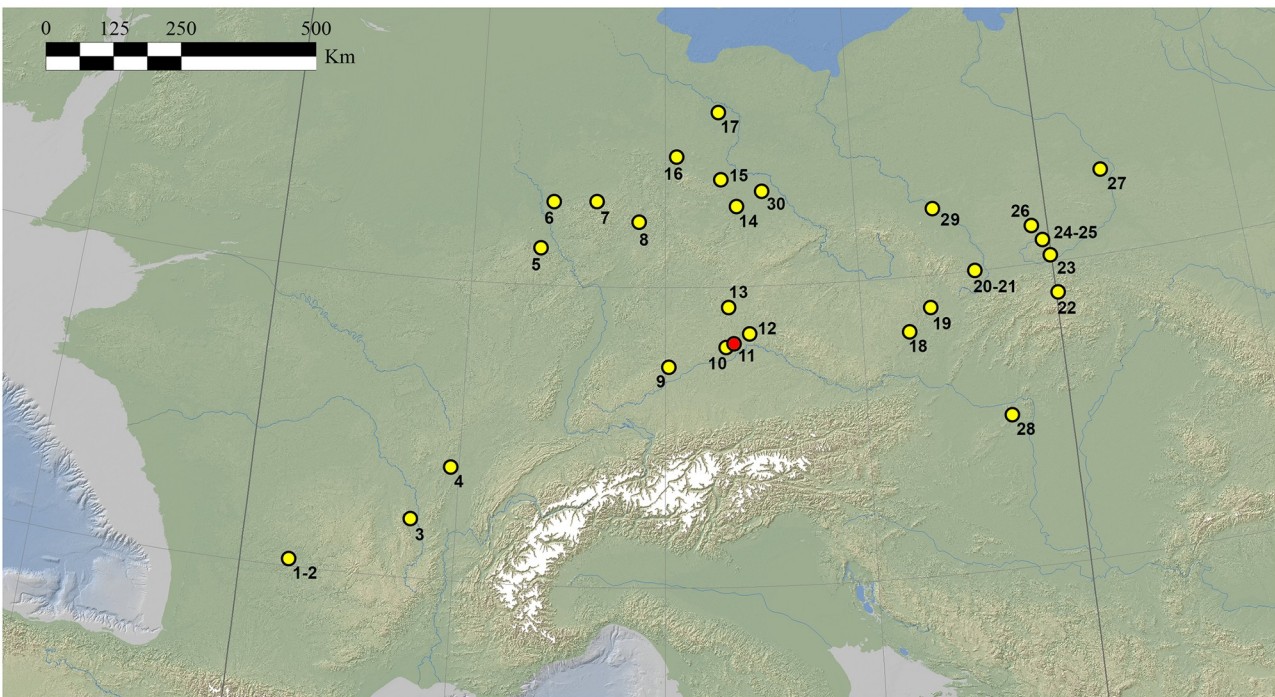

**Fig 1. Map of Central-Western Europe during GI 12 (46.8–44.2 ky) with Digital Elevation Model (base topography–ETOPO, 2011; ETRS_1989_LAEA_152 projected coordinate system) developed by Davide Margaritora and location of sites named in the paper: 1) La Micoque; 2) Abri du Musée; 3) Champ Grand; 4) Grotte de la Verpiliére; 5) Kartstein; 6) Neandertal; 7) Balver Höhle; 8) Bühlen; 9) Bockstein-III; 10) Klausennische; 11) Sesselfelsgrotte; 12) Zeitlarn-I; 13) Weinberghöhle; 14) Neumark Nord; 15) Königsaue; 16) Salzgitter-Lebenstedt; 17) Lichtenberg; 18) Moravsky-Krumlov IV; 19) Kůlna; 20) Dzierzyslaw-I; 21) Pietraszyn 49a; 22) Oblazowa; 23) Zwierziniek; 24) Wylotne; 25) Ciemna; 26) Bisnik; 27) Zwolén; 28) Tata; 29) Wroclaw-Hallera Avenue; 30) Pouch.**

classify Central European *Keilmessergruppen*/M.M.O. occurences focused on the presence or absence of bifacial tools and their variability, while putting less attention on simple unifacial tools. The main attention was on the eponymous Keilmesser, the leaf-shaped bifaces (*Faustkeilblätter*), the partially foliated (*Halbkeile*) and the small triangular bifaces (*Fäustel*), and a few typical, but rare tool types (bifacial scrapers, *groszaki*, etc.). This is in contrast to the Mousterian of Acheulean Tradition (MAT) [13,48], which was quite early understood as a combination of unifacial and bifacial tool concepts. The complexity of the general framework increased with the presence of a number of assemblages, mainly in North-Western Europe, that apparently could not be classified within one or the other entity (the "typological dilemma" according to Ruebens [49]), as they were characterized by the presence of Micoquian elements next to Levallois and laminar débitage systems, resulting in a new definition (Mousterian of bifacial tools–MBT [50]) recalling older ones such as the "Middle Palaeolithic Assemblages with Handaxes" (MPAH) by Kozlowski and Kozlowski [51]. Shifting the attention from the significance of the bifacial tools as *fossils directeurs* to a broader view, the importance of the relationship between the tools from *débitage* and *façonnage* and the productive concepts behind them started to emerge in different works of the so-called "techno-ecological approach" [6,52,53] aiming at techno-cultural distinction of the Neanderthal groups at the end of the Middle Paleolithic. The techno-ecological approach tries to place a tool into a context capable of understanding its complexity and effective cultural value. The tool fulfills functional needs, with different degrees of effectiveness and versatility, changing with direct relation to cultural and environmental constraints [54]. It is therefore necessary to integrate data from techno-

functional, economic/strategic and behavioral approaches in a research design that allows to up-scale the small-scale complexity and variability of individual tools and tool types to a broader model in favor of a global understanding of more general dynamics. In particular, the different technological responses to similar needs may represent forms of cultural choices and/ or environmental adaptation; in order to identify them, in-depth analyses are required besides precise chronometrical and paleoenvironmental framing.

Here we present a reassessment of one of the most important artifact types in the classical distinction between the Mousterian and the *Keilmessergruppen*/M.M.O., e.g. backed bifacial knives, integrated in a broader sample of backed items stemming from different technological concepts selected among unifacial and bifacial artifacts. As we shall see, these are analytical appropriate elements due to analogies in their ergonomics, functionality and technological characterization. The analyzed pieces come from the G-layer-complex of Sesselfelsgrotte, which is one of the most significative sequences when investigating the relationship between Mousterian and *Keilmessergruppen*/M.M.O. The study combines techno-functional and morphometric approaches with a broader perspective integrating the strategic, ecological and cultural significance of the toolkits and, on a larger scale, the respective techno-complexes.

## 2. Relevance of backed tools within Late Middle Paleolithic assemblages

Keilmesser is the tool type common to all the assemblages that are linked with the *Keilmessergruppen*/M.M.O. [23]. In general, Micoquian Keilmesser are widespread especially in Central and Eastern Europe, but also in the less frequent ocurrences of the *Keilmessergruppen*/M.M.O. in Western Europe (e.g. Verpilière I and II, Abrì du Musée in Les Eyzies de Tayac [40,55]). Apart from that, single pieces or related forms have been identified in very different, yet unrelated contexts from Africa to the Near East or even on the Iberian Peninsula [56,57], some of them dating back to the Middle Pleistocene [57]. However, it is only in the final Middle Paleolithic that they became increasingly standardized, numerically relevant and a common expression of more or less uniform lithic assemblages [26].

As suggested by the direct translation from the original German noun, the term Keilmesser describes a wedge knife and thus is merging the shape and the presumed function. In the literature, Keilmesser are sometimes referred to as "*Faustkeilschaber*" [58], backed bifaces (*biface à dos* [59]) or–in cases when combined with distinct outlines and/or features of manufacture—pradnik/prondniks [60] or Ciemna knives [51]. One of the most widely accepted definition [12] is that of a piece with an almost straight bifacial working edge opposite to a straight or angular back. This allows for a large variability of both in outline and cross-section asymmetrical tools with a carefully retouched, bifacial and sharp cutting edge, which is often shaping also a distal point, opposed to a usually rough and thick, often cortical prehensive part. The base of the tool is usually also thick and may have had a receptive function as well. If a distal point is present, a bifacially worked "bow" (distal posterior part) may form the "*trait d'union*" between the back and the cutting edge [23].

Typical in Keilmesser is the plano-convex/plano-convex cross-section shaped through flat retouch of the lower face to the cutting edge, and a corresponding convex direct retouch on the upper face, which can be simple or scaled. The cutting angle is usually acute and smaller than 60˚ [23,40,61] or only slightly larger [62]. In contrast to wider scrapers angles used with transversal motion, this is suggesting a function of a knife with longitudinal cutting motion. A reversal plano-convex section on the back is often applied in order to have a flat and a convex edge on each surface. Such an asymetrical volumetric scheme permits to maintain the effectiveness of the edge angles even after subsequent resharpening stages [53,61], and especially

when the tool is reoriented to reverse the edge functions by using a second working edge opposite to the first one [63]. The long stages of use, reuse and recycling of Keilmesser are testified by the detachments of core-reduction flakes, thinning products and orthogonal or longitudinal retouch as well as lateral resharpening flakes. Typical of the shaping of the cutting edge is the "tranchet blow" technique testified in several *Keilmessergruppe*/M.M.O. sites [29,64,65]. It follows that the final morphology can vary, depending on the raw material properties, the degree of reshaprening and/or other ecological and functional factors. Although the definitions of Bosinski´s tool types are still valid, they are no longer used as *fossils directeurs* for distinct entities, e.g. his "inventory types" [12]. It became common sense that simple generalizations on Keilmesser are misleading; instead, an in-depth working-step analysis of larger samples of objects is hence required to comprehend the different individual biographies before generating a general model for a given assemblage or site [9,19,52,66]. As a matter of fact, the Keilmesser tool concept contains objects which share the same ergonomic properties within the human–tool interaction system: the hand-held knife usage scheme is built around the direct opposition between the cutting edge and back. This backed tool, whose precise functions are however still to be confirmed by large scale use-wear analysis (e.g. cutting tools [18] or rather multi-functional [67]), already shows a marked variability in the macroscopical use-patterns that calls for an investigation from a techno-functional point of view.

However, within the same assemblages and in the coeval European landscape, other backed tools exist. These are mainly unifacial and obtained through different or complementary technological concepts. Although they can possess similar ergonomics and schemes of prehension and use, they are still technologically different as they result from other concepts of tool manufacture and determination of the back. Particularly relevant in this regard, also in the light of the present study, is to understand the process of decision-making that leads to the use (and potential resharpening and discard) of artifacts obtained by (bifacial) *façonnage* on the one hand, and/or the preference of artifacts manufactured by different concepts of *débitage* on the other. This is even more so, as almost always the assemblages in question are characterized by a considerable number of flakes derived from core-reduction concepts. The flake-oriented technologies in the *Keilmessergruppen*/M.M.O. are basically the same than those of the Mousterian complexes. Because of this, some explanatory models built on the relationship between unifacial and bifacial toolsets have been raised to explain the variability of *Keilmessergruppen*/M.M.O. complexes. In this regard, the Mousterian of Micoquian Option (M.M.O) after Richter [9,68], was initially coined on the base of the assemblages of the G-Layers-Complex from Sesselfelsgrotte. According to him, the "Micoquian Option" is an increase in the relative frequency of bifacial tools, which correlates to proxies interpreted as being indicative for a more intense use of the lithic artifacts discarded at a site, such as the ratio between simple scrapers and double/convergent scrapers or between notched and denticulated pieces. He concludes that the relative amount of discarded bifacial tools in a Micoquian context increases with a growing time of activity, e.g the length of the site occupation. This assumption has been questioned, among others, by Jöris, who argues with the absence of an independent confirmation of the model by faunal data, and emphazises some examples of ephemeral campsites where bifacial shaping is dominant (e.g. Lichtenberg) [23]. However, both base camps and ephemeral camps can bear long-life bifacial tools, whose origin and purpose can relate to several different tool biographies [52].

The technological nature of the flake industries of Mousterian tradition in *Keilmessergruppen*/M.M.O. contexts is quite diversified. Bosinski [12] and Kozlowski and Kozlowski [51] initially pointed out that Levallois concept was absent in the Micoquian assemblages of Central Europe. Quina débitage is present within the most ancient layers in Sesselfelsgrotte [9], in Bockstein III [69], while Discoid débitage is well attested in Kůlna cave all along the Micoquian

sequence [53], as well as in several Polish assemblages being part of the so-called "Bockstein group" according to Kozlowski and Kozlowski [51]. In the most recent *Keilmessergruppen*/M. M.O. sites, Levallois débitage is however usually dominant, in both its recurrent centripetal and recurrent parallel/unipolar variants, as attested in Sesselfelsgrotte, Königsaue, Salzgitter-Lebenstedt and other sites [9,19,33,35,52].

Each of these core reduction concepts result in sets of flakes, some of which present a back opposed to a sharp edge. These backed artifacts are not only strictly technical in that thay maintain the core shape and convexities, but at the same time may also represent the objectives of the knapping operations. Unaware of their respective frequencies, their production is in any case indispensable due to the technological function in the cause of the respective core reduction. Depending on the reduction concept, these artifacts can have different shapes: thick, short and asymmetrical when being part of the Quina concept [9,70,71], thick, short and at times pointed and/or asymmetrical within the Discoid one [53,72], and thinner and more elongated in case of the *éclat debordants* obtained by the Levallois concept [73]. The differences do not only lie in the morphology of the backed blanks, but imply distinct modification and curation strategies. In any case, the fact that they are used–indicated by the presence of retouched working edges—indicates that both their production and actual use results from different ecological and/or cultural responses to similar needs, e.g. a cutting edge opposite to a back. The reasons behind their manufacturing and discard need to be examined by taking into consideration their entire working life, from the conception to the techno-functionality until the (re)use potential. This is the starting point of the present study, which takes into exam the backed implements of a key sequence for the Mousterian/Micoquian relationship in central Europe.

In the final Middle Paleolithic, backed tools acquired a particular importance from both a behavioral and a cognitive point of view. The post-determination of the back by means of abrupt direct retouch is well attested in different European techno-complexes [48,74,75]. This additional technical investment is supposed to enhances the performance by improving the prehensive grip or to adapt the tool for hafting. Unaware of the precise function, it is the standardization of prepared backed implements which is considered as an indicator of "modern behavior" due to the degree of problem solving behind their manufacture and the growing importance they acquire within MP-EUP and Middle Stone Age contexts [76–80].

## 3. The case of Sesselfelsgrotte G-Complex

### 3.1 Stratigraphy

The Sesselfelsgrotte is a small rock shelter located in the Lower Almühl valley near to the village of Essing, Lower Bavaria. Its stratigraphy represents one of the key sequences of the Upper Pleistocene in Central Europe. The site has been investigated by the University of Erlangen under the direction L. Zotz and G. Freund from 1964 to 1977 and again in 1981. Of the excavated area of 57 square meters, approximately 20 square meters are under the roof of the rock shelter. The stratigraphical sequence of 7 m is mainly formed by limestone débris and was subdivided into 18 geological layers, which were again differentiated into 35 sedimentological sub-layers. Its relevance is mainly due to the stratified preservation of 22 Middle Paleolithic occupations, which are characterized by numerous evident features in the form of fireplaces, well preserved lithic and faunal remains and the presence of Neanderthal remains [37] (Fig 2). Together with layers of the Upper Paleolithic and Mesolithic, they are embedded in a stratigraphical sequence that yielded environmental data (e.g. pedology, small mammal fauna, malacofauna) and was absolutely dated by numerous radiocarbon and TL-dates. The archeological sequence starts at the base with eight human occupations of the "*Untere Schichten*"

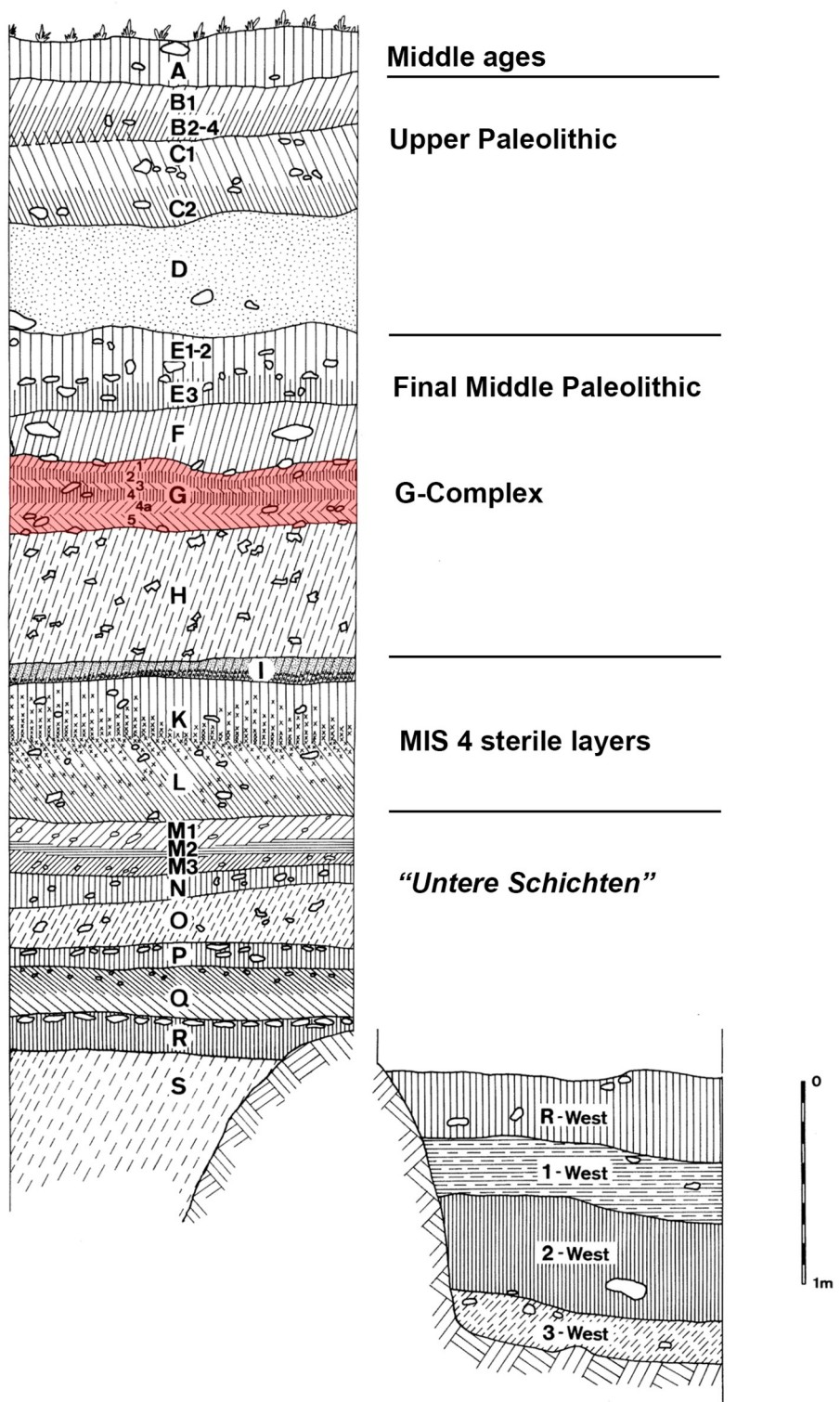

**Fig 2. Stratigraphical sequence of Sesselfelsgrotte: G-Complex layers are highlighted.** Modified from Freund, 1984.

("lower layers") from geological layer R to M, which are characterized by landscapes fluctuating from forestal to more open steppe [27] and dated to temperate interstadials at the beginning of the Weichselian glaciation (probably MIS 5c and 5a). The datation is confirmed by the development of the thickness of the enamel of molars in *Arvicola terrestris*, which clearly postdates the Eemian [81]. The Mousterian assemblages of the lower were classified as Mousterian with small dimensions ("Taubachian"), Charentien Type Ferrassie, Charentien Type Quina and Moustérien typique [27]. Above it follows the MIS 4, which is stratigraphically represented by sterile layers (L, K, I) indicating cold and arid conditions testified by the first appearance of mammoth and micro mammals typical of arctic tundra, accompanied by an increase of humidity in layer I [81]. The overlying layers H and G were termed "G-Complex" and contained thirteen subsequent occupations recognized in 60 cm of stratigraphy and dated to the onset of MIS 3 [9]. The following layer F is sterile and separates the G-Complex from a the last Middle Paleolithic frequentation of the site in layer E3 [82]. Above this, archaeologically sterile loess layers are correlated to the last glacial maximum (layer D). Human presence starts again in layers B and C with assemblages associated to the Magdalenian [83]. The Holocene Layer A with finds from the Mesolithic to the Middle Ages seals the sequence.

## 3.2 Lithic assemblage data

The G-Complex yielded the remains of recurrent and–in part–intensive human occupations testified by numerous fireplaces situated both under the roof and in front of the drip line. From a sedimentological point of view, the presence of evident features and at least two living floors (sublayers G2 and G4) underline the excellent preservation conditions. These are equally attested by the many faunal remains and the fossils of three Neanderthal indviduals, one of which was identified as a foetus or neonatus [84]. Approximately 85,000 lithic artefacts were recovered from six sublayers of the G-Complex (G1, G2, G3, G4, G4a, G5) and from the underlying layer H. To also consider vertical post-depositional movements of artifacts typical for cave and rock shelter fillings, Richter [9] used the compostion of raw material units in excavation units (e.g. sediment removals in square metres) to identify coherent assemblages. Based on the assumption that every occupation has a specific raw material procurement and using the underlying precondition of a spatial connectivity of the artifacts discarded during each occupation, a cluster analysis resulted in 13 lithic assemblages. Whereas some were conform with the geological layers, others strechted over two. The occupations are thus characterized by different stratigraphical nuclei, different raw material spectra and/or different spatial distributions, the latter (but not always) restricted to the inner or the outer part and correlating with a fireplace.

All 13 assemblages are associated to the already mentioned M.M.O., characterized by varying frequencies of bifacial tools and the presence of Levallois products. The occurence of backed bifacial knives in almost all assemblages allows to classify the archaeological record of the G-Complex as belonging to the *Keilmessergruppen*/M.M.O. Despite technological and typological close relations, the assemblages show a considerable variabilty: whereas some are dominated by Mousterian elements and have only few or none bifacial tools, others have frequencies of Micoqiuan bifacial tools that equal typical inventories of the *Keilmessergruppen*/M.M.O. The other aspect of variability in between the assemblages concerns the diversity of the raw material, which is based on a detailed sortation of artifacts according to outcrops and thought to mirror differences in the procurement strategies and/or the availability of raw material sources. The key for the model of the M.M.O. developed by Richter lies in his observation that the diversity of the raw material in both the Jurrassic cherts and quartzites underwent a cyclical change. Each of the four identified cycles begins with a high diversity of raw

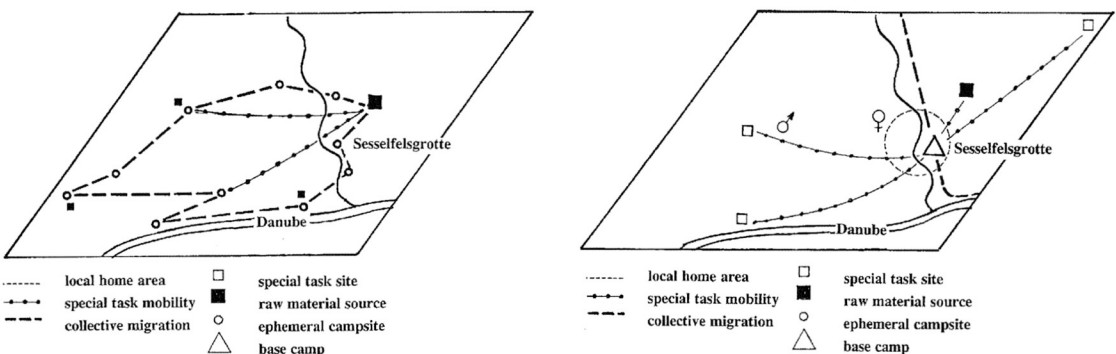

**Fig 3. Seasonal cycles of occupation recognized by Richter (1997) for G-Complex layers.** On the left, the mobility and resources exploitation patterns during the *initialinventar* (circulating land-use patterns during summer); on the right, the spatial behaviour during the *konsekutivinventar* (radiating land-use pattern during autumn). From Richter 2001.

material units termd "*Initialinventar*" ("initial assemblage"), and ends with assemblages with a low diversity of the exploited raw material resources named "*Konsekutivinventar*" ("consecutive assemblages"). The raw material diversity correlates with a more Mousterian-like charactersitic of assemblages at the start of a cycle, and a pronounced Micoquian character (cause by maximal frequencies of bifacial tools) at the end. The ecological interpretations favored by Richter mainly bring back to two hypotheses, the first based on the knowledge of the local territory and the selection rate of the better raw material, and the second based on seasonal changes in the availiabilty of resources [9,47,68,85]:

Whatever the interpretation, the different assemblages of the G-Complex clearly attest a diversity in the length of the site occupation, the mobility and the underlying land use strategies (Fig 3). This is indicated by the intra-cycle differences: each cycle begins with a small assemblage in which the typical Micoquian toolset is scarce and the tool ratio is strongly balanced towards scrapers. In the next phases of each cycle, an increase of the bifacial forms as well as of the laminar component and the denticulates is evident, the latter being–according to Dibble and Rolland [86]—proportional to the occupation length. The micoquian-denticulates-laminar package would hence increase according to this variable, attesting the shift from short-term campsites to base camps and accompanied by a change in land-use strategies [68]. Amongst the other main aspects of the lithic assemblage, it is worth mentioning that a micro-lithic component exists, which is manufactured on flakes obtained from advanced stages in the reduction of cores of local raw materials, e.g. from curated cores without cortex left. According to use wear, these tools, which are either "*raclettes*" (also termed "groszaki" or informal tools), served locally for the processing of soft materials and–most probably—of plants [87]. Finally, some Upper Paleolithic tool types are present, mainly untypical end-scrapers and burins. Regarding diachronic inter-cycles differences, a change in the technical systems of core redcution is attested: whereas the Quina concept is evident in the first cycle, the second is characterized by both Quina and Levallois products. During the third cycle, a preference of the Levallois recurrent centripetal method can be observed, and in in the fourth cycle cores of the Levallois recurrent parallel method gain more importance. A significant change is seen in the presence of the Quina concept in the lower part of the G-Complex and its lack in the upper part, resulting in the postulate of a M.M.O. A with Quina and M.M.O. B without Quina core reduction concept. This chronological sequence has been virtually extended to all the late Middle Paleolithic *Keilmessergruppen*/M.M.O. sites [9].

### 3.3 Environmental data and chronology

The large mammal fauna of the G-Complex has been examined for the respective sublayer. Due to the occasional lack of correspondence between the geological sub-layers and the stratigraphical nucleus of lithic inventories, a verification of the model based on seasonal changes in the availiabilty of resources will not be possible for every assemblage [88]. However, an analysis of the meso- and microwear of horse teeth from assemblages that show a stratigraphical conformity with excavation layers is under preparation. In sum, the G-Complex faunal assemblages contain the frequent remains of *Mammuthus primigenius*, documented starting from layer M (MIS 4), and woolly rhinoceros, but are dominated by reindeer (*Rangifer tarandum*) and horse (*Equus sp*.). Among the other species, chamois, ibex, wolf, fox and arctic fox are frequently found, too, while the remains of giant deer, bison, red deer and cave bear are few.

Information about environmental and climate indicators which accumulated without human agency in the different layers is significant. Studies of small and micro mammals [81], birds [89], molluscs [90], pollens and charcoals [91] unanimously attest a progressive growth in temperature, humidity and extension of fores cover from layers L to H. In particular, the increase of pine, birch and mugwort indicates a transition from a glacial to a more temperate phase. In the G-layers, charcoals document the spread of *Larix/Picea* and the presence of *Prunus*, while the scots pine is the only left pine tree. However, among the pollen a decrease of arboreal species in favor of an increase of Poaceae is noted if compared to layer H. At the same time, small mammals' species typical of cold environment and arctic conditions confirm a certain degree of climatic instability and indicate an open vegetation with only sparse forested areas.

The G-Complex has been dated by 14C, TL and IR-OSL methods. Radiocarbon dates on bones and charcoals provided dates between 30.770 and 47.860 calBP [38], but from these, only dates coming from the inner part are supposed to be reliable due to contermination by running water in the oute part in front of the dripline. Therefore, the reliable radiocarbon time window is reduced to between 39.950 e 47.860 Cal BP [82]. Except for one deemed date, a series of TL dates on burnt flints of 56.0 ± 4.7 ky support the validity of the older absolute dates from the inner part [92]. In general, the chronological position of the G-Complex in MIS 3 is verified by absolute dates from the underlying M-layers, which provide a medium date of 73.2 ± 11.7 ky, consistently with an attribution to MIS 5a.

All the available radiometric and environmental data are in agreement with a datation of the G-Complex (from layers H to G) to the beginning of MIS 3, framed within the Oerel-Glinde interstadial. The layers formed in a milder phase if compared to the previous ones (from L to I), but still characterized by a probable cooling between layer H and the G-layers. Therefore, a certain climatic instability is documented; though more humid and temperate, these conditions indicate a landscape of open *larix/picea* boreal forest (taiga), populated mainly by mammooths, reindeers and horses.

## 4. Materials

The sample of artifacts analyzed in the present study was selected from the assemblages of the G-Complex of Sesselfelsgrotte, stored in the Prehistoric collection of the Friedrich-Alexander-Universität Erlangen-Nürnberg (FAU), Institut für Ur- und Frühgeschichte, Kochstr. 4/18, 91054 Erlangen. Since one of the authors (T.U.) is responsible for the collection, no permits were required for the described study, which complied with all relevant regulations. The criteria for the basic population, from which the analyzed items were sampled, were the presence of a raw or retouched cutting edge, opposed to a blunt edge forming a back, which could be cortical, the result of knapping prior to the detachment of the piece, or prepared. These "backed

tools" *sensu lato* fall into three main categories: Keilmesser, backed scrapers and backed flakes. The classifications were taken from Richter [9]. The first two groups consist of tools *sensu stricto*, determined according to an empirical-inductive approach by the identification of the technical investment to obtain the cutting edges. This investment, which we read as retouch, is a direct evidence of their intended or actual use. The third category is composed by potential tools, determined according to a hypothetical-deductive and technological approach. Through a series of techno-functional attributes (regularity, dimensions, ergonomics, handling, functionality) and by analogy with items of the aforementioned groups of tools *sensu stricto*, it is assumed that these were or could be used.

The present study is focused on backed tools s.l. because:

- Their ergonomics and functionality are apparently simple and linear, because they are characterized by a clearly identifiable active and equally distict passive part; the potential function is ranging from knives to multifunctional tools. The broad analogies between retouched and unretouched as well as unifacial and bifacial backed tools allow different analytic approaches ranging from the techno-functional to the geometric morphometric.

- Backed items represent one of the main objectives of the lithic reduction concepts of both *débitage* and *façonnage* operations.

- They perfectly exemplify and summarize the two technological and cultural worlds of the Mousterian and the Micoquian: Keilmesser on the one hand are the eponymous tool of the *Keilmessergruppen*/M.M.O., and backed flakes (obtained via Discoid, Levallois or Quina technologies) on the other hand are, alongside with backed scrapers, the common Mousterian toolset.

All in all, 347 backed tools were selected within the assemblages of the G-Complex, including 58 Keilmesser, 118 backed scrapers, 155 backed flakes and 18 bifacial tools without a lateral back that conventionally are classified as different variants of handaxes (e.g. *Faustkeile*, *Halbkeile* and *Fäustel*) (Table 1; see also Supplementary Information for a detailed database). These later differ from Keilmesser because of a general symmetry and the lack of a back opposite to the cutting edge. However, one potential prehensive part is usually at the base; in cases where the base was thick, handaxes were included in the sampled artifacts. Given their significance, almost all Keilmesser from the G-Complex (assemblages A01 to A10) were analyzed. For the other artefact types, a twofolded sampling strategy was employed: for assemblages from the last cycle (assemblages A01, A02 and A03), it was tried to include almost all artefacts that meet the defining parameters described above. For the remaining three cycles, tools with specific interventions on the back and those representing the dominant tool or blank types with a back were chosen. In sum, the sampling allows to analyze not only a complete cycle for quantitative and qualitative comparisons, but at the same time to draw a complete picture of the techno-functional variability of all occupations.

**Table 1. Distribution of analyzed tools for techno-typological categories and inventory (from Richter 1997).**

|          | A01 | A02 | A03 | A04 | A05 | A06 | A07 | A08 | A09 | A10 | Other | Total |
|----------|-----|-----|-----|-----|-----|-----|-----|-----|-----|-----|-------|-------|
| Keilmesser | 10 | 4 | 4 | 5 | 1 | 11 | 0 | 10 | 8 | 5 | | 58 |
| Scrapers | 36 | 24 | 10 | 6 | 5 | 6 | 2 | 17 | 3 | 5 | 2 | 116 |
| Flakes | 93 | 28 | 12 | 16 | 0 | 3 | 2 | 1 | 0 | 0 | | 155 |
| Bifacials | 5 | 1 | 4 | 4 | 0 | 1 | 0 | 2 | 0 | 0 | 1 | 18 |
| Total | 144 | 57 | 30 | 31 | 6 | 21 | 4 | 30 | 11 | 10 | 3 | 347 |

## 5. Methods of analysis

### 5.1 Techno-functional analysis

One method of analyzing and comparing lithic tools is the techno-functional approach according to Lepot [93] and Boëda [94], which has already been applied by several scholars for the study of Middle Paleolithic assemblages a whole, as well as for specific tool types [95–97]. The techno-functional approach considers a tool as an object for the transfer of energy applied by humans to alter material matter. To do so, tools are composed of three functional units or contacts: a prehensive unit, a receptive and a transformative one. This approach considers the entire lifetime of an implement from prior to its production until its abandonment, e.g. through all stages of the operational chain from planning of tasks, raw material procurement and reduction to the actual use and abandonment. The method is embedding morphological and technological aspects by identifying the tools' targets and usage schemes through the analysis of single units and their internal relationships. Each tool is subdivided into its structural elements. If viewed upon from the perspective of tool use, backed tools usually have a "passive" part, which includes zones for the receptive\prehensive contacts, and a corresponding "active" part, which allows for the transformative contacts to the worked materials.

The receptive\prehensive contacts (Boëda [94]: "*Contact Préhensif*"—**CP**) represent the tool's subsystem aimed at handling it and receiving the energy from the user. On the artefact, it generally corresponds to the portion of the blank opposite or adjacent to the natural or retouched working edge. The CP can be produced or shaped through technical investment (e.g. retouch, lower or upper thinning), but may also be represented by the striking platform (e.g. the basal part) or cortical or knapped areas of the laterals, which were unaltered or produced prior to the detachment of the blank and therefore result from the core reduction technology. The energy applied on the CP may also, be controlled by an additional handle or other means to enhance the prehension.

The transformative contact (Boëda [94]: "*Contact Transfomatif*"—**CT**) represents the tool's subsystem aimed at releasing the energy and transforming the material on which the tool is used. On the artefact, transformative contacts can be identified macroscopically by the existence of a sharp *tranchant* usually produced by retouch and opposed or adjacent to the prehensive contact. For one and the same artifact, single or multiple CT can exist on the laterals, but also forming a point. At the same time, parts defined as a continuous cutting edge in most typological classifications, as it is the case for simple side scrapers, may represents several CTs (Lepot [93]). The CT can be shaped by flaking or by bifacial *façonnage*, and it may be subsequently retouched for better performance.

Starting from the morphology and the organization of the contacts of the analyzed tools, the comparative part of the approach used here aims at the identification of recurrently applied technological and morpho-functional features, which allow to summarize the main aspects in one (or several) scheme(s). The schemes are reconstructed by a systemic approach that structures a series of elements and puts them into relation. Each resulting artifact system is reflecting a level of organization on a quite general scale, which allows to further investigate general questions related to its structural, technological and ergonomic elements. By integrating information about human-environmental interactions inferred by the site setting, the site function and the ecological frame of the occupations, it is possible to compare the mental concept of the tools and to interfere the relevance of the techno-functional needs and the various ecological constraints that their makers and users were confronted with.

We collected data for attribute and morphometrical analysis (See Supplementary Information for a detailed database). Whereas part of the data was collected from the 3D scans of artifacts, others were recorded manually. Among these is the following series of morphological

elements and technological features: the maximum width, length and thickness, the amount of cortex left on the surface (subdivided into six classes from 0 to 100%), the weigth, the blank type (e.g. flake, Kombewa-flake, plaquette, pebble or frost flake), the general shape and cross section. For each CT, the following data have been registered: cordal length, overall curval length, minimum and maximum active angle (measured with a manual goniometer in 5˚ intervals), the shape of the longitudinal profile (from both the top and the sagittal view), and– if present–the type and extension of the retouch (according to Bourguignon [71]). Particular attention was payed to the bevel, e.g. the dihedral surfaces which shape the cutting edge. Bevel types were recorded based on different classes numbered from 1 to 7 (from concave to flat to convex, with intermediate classes between), considering the relationship between cutting edge and the lower face first and then relationship between the cutting edge and he upper face; for example, a combination of flat and slighly convex results in the class 4–5 (Fig 4). For assessing the conception of the cutting edge, it has been distinguished between "predetermined" when formed by flaking prior to the detachment of the blanks and therefore embedded in the knapping system) or left raw and not retouched, and "post-determined", indicating a manufacture through flaking of the blank´s edge (mainly by retouch). For the documentation of techno-functional units forming a points, we recorded the overall shape, the angles in plane and section, the transversal cross-section (measured at about 5 mm from the point), the active bevel and, finally, the technological conception (in classes: natural, flaking, retouched and mixed). For each CP, we also recorded the cordal length, the overall curval length, the minimum, medium and maximum thickness of the back, the sagittal profile, the technological conception (subdivided in natural, fracture, flaking, retouched and mixed), the angles of lower and upper edges as well as the presence of anthropic modifications of the back using parameters already applied for the analysis of backed artefacts from Fumane Cave, unit A9 [75]. Finally, the relationships between the different techno-functional units were analyzed. In addition to the reconstruction of the aforementioned schemes, we caluclated the quantitative and morphometric relationships between the active and passive parts and the type of dexterity as well as the potential pressure applied to the back inferred from the degree of bifacial symmetry of the artefacts. These data have direct implications for hypothesis about the lateralization of the analyzed tool; however, to securely access this aspect, a systematic and statistical approach, corroborated by experimental tests, would be necessary. Data concerning the raw material and the techno-typology of the blank (keilmesser, scraper, flake tool, other bifacial tool) were taken from past studies [9,27]. For data management, analysis, charting and basic statistics, we used the "ggplot2", "MASS", "FactoMineR" and "Tidyverse" packages of RStudio Version 1.2.5001, and Past 3.15 version.

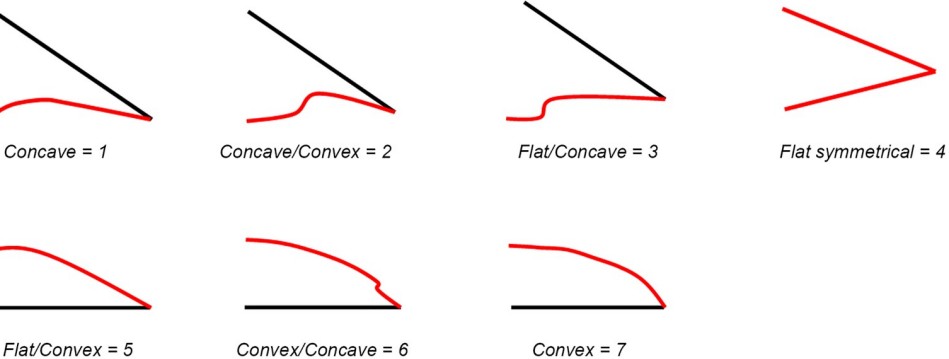

*Concave = 1*    *Concave/Convex = 2*    *Flat/Concave = 3*    *Flat symmetrical = 4*

*Flat/Convex = 5*    *Convex/Concave = 6*    *Convex = 7*

**Fig 4. Classes of cutting edge bevel in cross-section view.**

## 5.2 3D approach to techno-functional

The innovative and experimental methodological approach of this study is the investigation of different analytical parameters observed both manually and digitally directly on the artefacts' three-dimensional digital templates. 3D models were made for a large sample of the tools. The all in all 198 artefacts can be divided as follows: 55 Keilmesser, 82 backed scrapers, 49 backed flakes and 12 other bifacial tools. The 3D models were produced by one of us (D.D.), using a DAVID 3 structured light scanner at the Department of Computer Science at the Friedrich-Alexander-Universttät Erlangen-Nürnberg (FAU) in Erlangen. The scanner works with a camera and a beam light, which is projected onto the object and records it with a resolution up to 0,05 mm. After a calibration on small-size objects, each single scan was manually aligned in the DAVID PRO software. The obtained models were exported in Obj format for further processing and analysis (see below).

In general, 3D models permit a higher level of analytical interaction with the objects, e.g. the possibility to dissect the tool into parts or to investigate the three- or two-dimensional spatial relation between selected points, and to obtain very precise morpho-metrical measurements (Fig 5). Consequently, the main dimensions (length, width, thickness) were also taken virtually. However, the models were mainly processed in order to conduct computerized morphometrical and statistical analysis. The pieces were aligned along the main axis, with the back on the left, the point (if present) upward and the base downward, regardless from the position of the ventral or dorsal faces. Through the use of specific software (Autodesk Meshmixer and Geomagic Studio 2013), the pieces were dissected in precise and recurring points: generally, 3 transversal cross-sections were obtained in the proximal, mesial and distal portions, with the addition of a longitudinal cross-section for pointed tools. The sections allow the precise visualization of the active angle of the cutting edge, its lower and upper bevel, the thickness and the morphology of the back, and the relationship between active and passive parts. All these elements were recorded on the virtual model and not only used for analysis on their own, but also to correct the manual measurements whenever necessary.

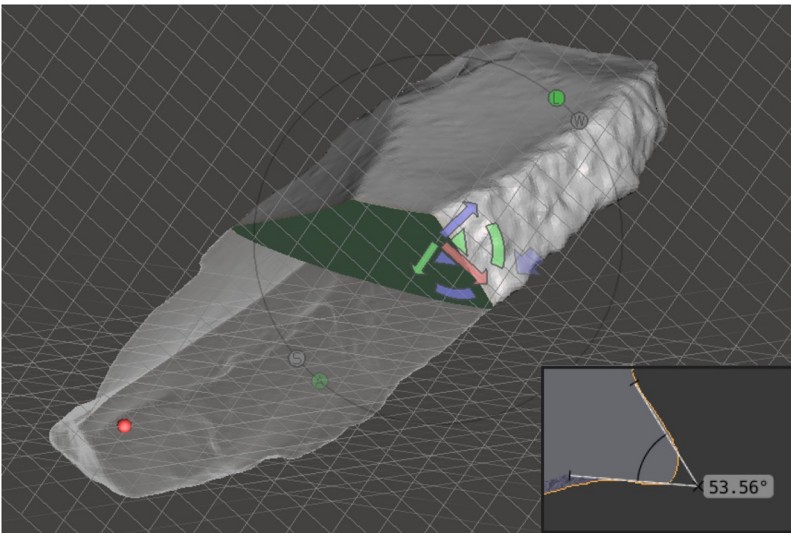

**Fig 5. Representation of cross-sectioning a virtual model of lithic tool and acquisition of angles' data in cross-section.**

In the next step, automated morphological analysis were applied on the 3D models of the artefacts. This approach has been developed in the last years, and successfully used in the analysis of bifacial tools in a growing number of case studies [62,98–100]. Shape statistical analysis were carried out with the software AGMT-3D, which has the advantage of performing several steps automatically [101]. Files of the objects in Wrl-format were imported into the software and at first processed for the acquisition of data: after a general alignment as described above, 3D semi-landmarks were obtained automatically with the construction of a grid formed by 50 meridians and 50 parallels, for a total of 2,500 landmarks on each surface. It followed the processing and statistical analysis of the data with the same software. It included the superimposition with the Generalized Procrustes Analysis (GPA) to normalize the variables of location, orientation and scale from the shape analysis. Based on this, a multivariate statistical analysis with Principal components (PCA) was carried out. The scores of the first two principal components (PC) are graphically presented in a two-dimensional scatterplot, while an additional report panel lists the absolute and relative variability of all PCs. Another function of the software used here is the "warp tool", which shows the trends in shape difference expressed by each PC until the hypothetical medium shape. Moreover, there is the possibility to assign each object of the sample to different groups using two separate attributes, a key function for the comparison of the variability of shape and differences in the mean shape between different categories of artefacts. Following this assignment, items on the scatterplot were color-coded according to their attributes, and ellipses indicating 90% confidence, as well as their centroids, were plotted for each group. In addition, the assemblage variability panel of the AGMT-3D software includes several analytical tools which were used in the statistical analysis and are explained in detail in the respective sections. Analytical tools used are the calculator for the group-mean distance (for the calculation of the multidimensional Euclidean distances between the groups' centroids) and the tool to compare the mean tool shapes. The results of the analysis are cross-checked by statistical tests for their significance.

The 3D morphometric and statistical analysis were developed on entire artefacts in order to comprehend the shape variability of the different typological and technological groups and the influence of raw materials and blanks. Furthermore, analysis also took into consideration the stratigraphical variability (to check for variability in time), and the degree of correspondence with the techno-functional schemes.

In the last step, the results of the 3D morphometric and statistical analysis were investigated for their technological and ecological context within the G-Complex of Sesselfelsgrotte, whose high-resolution permits to put the tools into a wider technological and techno-economical framework. In order to come to a more generalized picture, the results of the case study at Sesselfelsgotte are compared to the Mousterian and Micoquian context of Europe.

## 6. Results

On most cases, the results of the analysis of artifacts from the G-complex of Seselfelsgrotte refer to a basic techno-typological differentiation into the three main groups of backed tools present in the respective assemblages: Keilmesser, backed scrapers and backed flakes. Bifacial tools other than Keilmesser serve as a control group to evaluate the general variability within the bifacial tools.

### 6.1 Quantitative and qualitative data

**6.1.1 Metrical data.**   From a metrical point of view, the differences between the four groups are particularly evident in the weight. Moreover, kruskal-wallis tests for equal medians highlight that there are significant differences between samples medians (Table 2; Fig 6a–6d).

**Table 2. Data on tools' metrical dimensions and weight showed for techno-typological category.**

| Dimensions | Length in mm | | | | Width in mm | | | | Thickness in mm | | | | Weigth in g | | | |
|---|---|---|---|---|---|---|---|---|---|---|---|---|---|---|---|---|
| | Mean | min | Max | St. Dev. | mean | Min | max | St. Dev. | mean | min | max | St. Dev. | mean | min | max | St. Dev. |
| Keilmesser (N = 56) | 52.4 | 28 | 86 | 13.76 | 32.1 | 18 | 56 | 6.54 | 12.6 | 7 | 24 | 3.91 | 24.0 | 6 | 104 | 12.53 |
| Scrapers (N = 109) | 57.0 | 30 | 140 | 17.96 | 39.0 | 17 | 80 | 13.24 | 12.5 | 6 | 31 | 4.66 | 33.6 | 7 | 250 | 36.98 |
| Flakes (N = 140) | 47.0 | 16 | 113 | 14.32 | 33.8 | 9 | 95 | 12.87 | 10.8 | 3 | 26 | 3.67 | 15.8 | 1 | 113 | 15.56 |
| Bifacials (N = 17) | 58.5 | 36 | 87 | 14.29 | 41.9 | 25 | 60 | 9.61 | 14.3 | 5 | 26 | 5.23 | 34.1 | 6 | 70 | 17.64 |
| **Kruskal-Wallis test** | $p$ = 4.228E-08 | | | | $p$ = 1.995E-05 | | | | $p$ = 0.01004 | | | | $p$ = 4.206E-10 | | | |

However, if we perform Mann-Whitney U tests between each of Keilmesser's dimensional parameter and other tools' dimensions, we don't notice statistically significant differences especially between Keilmesser and scrapers (Table 3).

Despite the small range of the length of the Keilmesser, two peaks can nevertheless be noted: one around 40 mm and a second one just under 60 mm (Fig 6b). Backed flakes are on average the shortest tools with peaks in length around 35 and 50 mm (Fig 6c). At the same time, they are also the thinnest among the analyzed artifacts, which can be interpreted as resulting from the blank selection. Among the KMs, the overall smaller width mean and standard deviation comes to the eye, leading to more standardized and narrower outlines; the length/width ratio is 1.67. Regarding the mean thickness, an increase among KMs is recorded within the assemblages of the last occupational cycle, where it varies from 11.8 to 14.3 mm and the length/thickness ratio varies from 4.8 to 3.9 mm (Fig 6f). The dimensions of backed flakes do also not change considerably in between the assemblages of the different cycles. Contrary to expectations, with a length/width ratio changing from 1.74 to 1.43, flakes in assemblages of the last cycle are less elongated, although the Levallois recurrent parallel method is the prevailing technology, The weight of the artefacts of the analyzed sample varies from 1 to 250 g, with a significantly lower values for flakes and larger weights for bifacial tools and backed scrapers

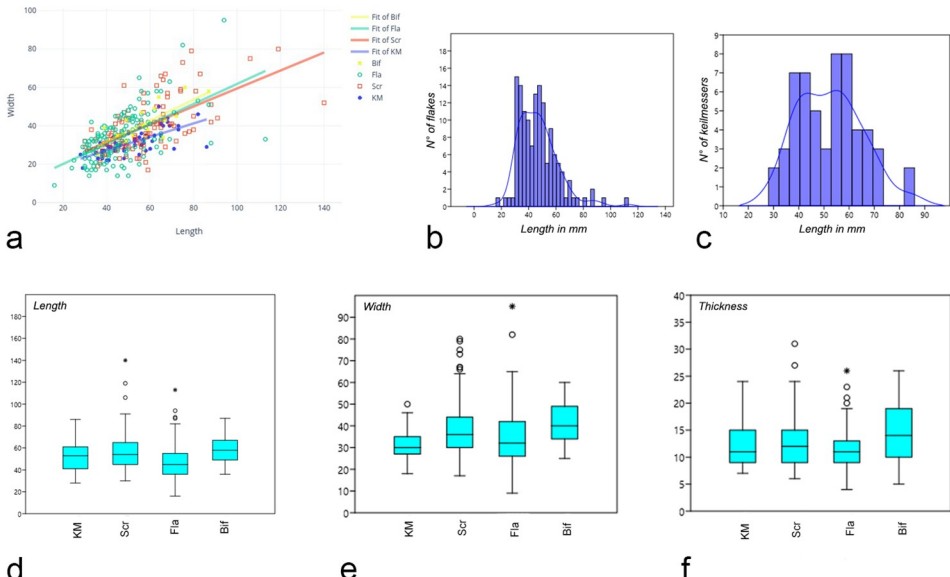

**Fig 6. Dimensional distribution (length/width) of the analyzed artifacts showed for typology (a); histograms of dimensional classes of flakes (b) and Keilmesser (c) length; boxplot showing respectively length (d), width (e), thickness (f) of the tool-types classes ("KM": Keilmesser; "Scr": scrapers; "Fla": flakes; "Bif": other bifacial tools).**

**Table 3. Mann-Whitney U tests on length, width, thickness and weigh between Keilmesser and other artefacts' typologies; in bold, *p* values > 0.05 attest no statistically significant differences.**

|  | Scrapers | Flakes | Bifacials |
|---|---|---|---|
| Length | ***p* = 0.6632** | *p* = 8.037E-05 | ***p* = 0.3234** |
| Width | *p* = 7.643E-05 | ***p* = 0.312** | *p* = 6.732E-05 |
| Thickness | ***p* = 0.8901** | *p* = 0.06851 | ***p* = 0.1126** |
| Weight | ***p* = 0.08548** | *p* = 4.081E-05 | *p* = 0.005044 |

which, however, record little homogeneity (Table 2). Keilmesser set halfway with a median of 24 g, and a range between a minimum of 6 g and a maximum of 104 g; despite this, no significant differences are noticed between scrapers and Keilmesser weight values (Table 3).

**6.1.2 Blanks and raw material used for the manufacture of backed pieces.** The typological difference between the tool categories is usually associated with a technological difference, which includes the blank as well as the technical system applied to obtain the form of the pieces.

*6.1.2.1 Keilmesser.* Most Keilmesser are manufactured from plaquettes (72.5% of the total, raising to 82.4% if only determinable blanks are considered). Because they are thin and have plane, and at the same time parallel, cortical lower and upper faces, plaquettes are usually well suited for the manufacture of flat bifacial tools (Table 4). Plaquettes chosen for the production of Keilmesser in the Sesselfelsgrotte often had several natural lateral fractures before being flaked, which could be exploited in two ways: first, as a striking platform to begin the bifacial knapping, or second, as a unaltered section at the base and/or the back (Fig 7). The shaping phases of the tool manufacture began on the lower face with the removal of the cortex, followed by the retouch of the cutting edge on the upper face and–if necessary—some other technical arrangements, such as the modification or the thinning of the back or the shaping of the point. Almost all plaquettes used during the G-Complex are Jurahornstein (Jurassic chert). Characteristic features of the cortex and of the color allow to classify most of them as raw nodules taken out of the outcrop of Baiersdorf [102], where until today raw pieces can be collected from the paleosoil of Franconian Jura plateau approximately 4 km north-west of the site as the crow flies (Table 5). The fairly high number of undeterminable blanks (n = 7 or 12.1%) it is mainly due to extensive retouch that in some cases characterizes the lower face.

Not every asymmetrical backed biface originates from the direct *façonnage* of raw nodules. In fact, five flakes have been recognized as starting blank. In addition, two rather large natural flakes, which derived from frost cracking, have been identified as blanks, too. The flakes, although heavily modified by retouch, can be reconstructed as originally being of medium size. It is highly likely that they were obtained from medium or advanced stages of the reduction of different raw materials, including Jurassic and Cretaceous cherts as well as quartzite and lydite. None of the flakes have a cortical dorsal cover of more than 50%, and one of them (n° S.2222) is the only keilmesser without natural surfaces.

**Table 4. Raw blank for techno-typological category of tools.**

|  | Pebbles | | Plaquettes | | Frost flakes | | Flakes | | Kombewa-flakes | | Indet. | | Total | |
|---|---|---|---|---|---|---|---|---|---|---|---|---|---|---|
|  | N | % | N | % | N | % | N | % | N | % | N | % | N | % |
| KM | 2 | *3.4* | **42** | **72.5** | 2 | *3.4* | 5 | *8.6* | \ | \ | 7 | ***12.1*** | 58 | *100.0* |
| Scr | 2 | *1.7* | 20 | *17.2* | **17** | **14,7** | **68** | **58.6** | 3 | *2.6* | 6 | *5.2* | 116 | *100.0* |
| Fla | \ | \ | \ | \ | \ | \ | **148** | **93.5** | 6 | *5.8* | 1 | *0.7* | 155 | *100.0* |
| Bif | \ | \ | **13** | **72.2** | \ | \ | 2 | *11.1* | \ | \ | 3 | *16.7* | 18 | *100.0* |
| Total | 4 | *1.1* | 75 | *21.6* | 19 | *5.5* | 223 | *64.3* | 9 | *2.6* | 17 | *4.9* | 347 | *100.0* |

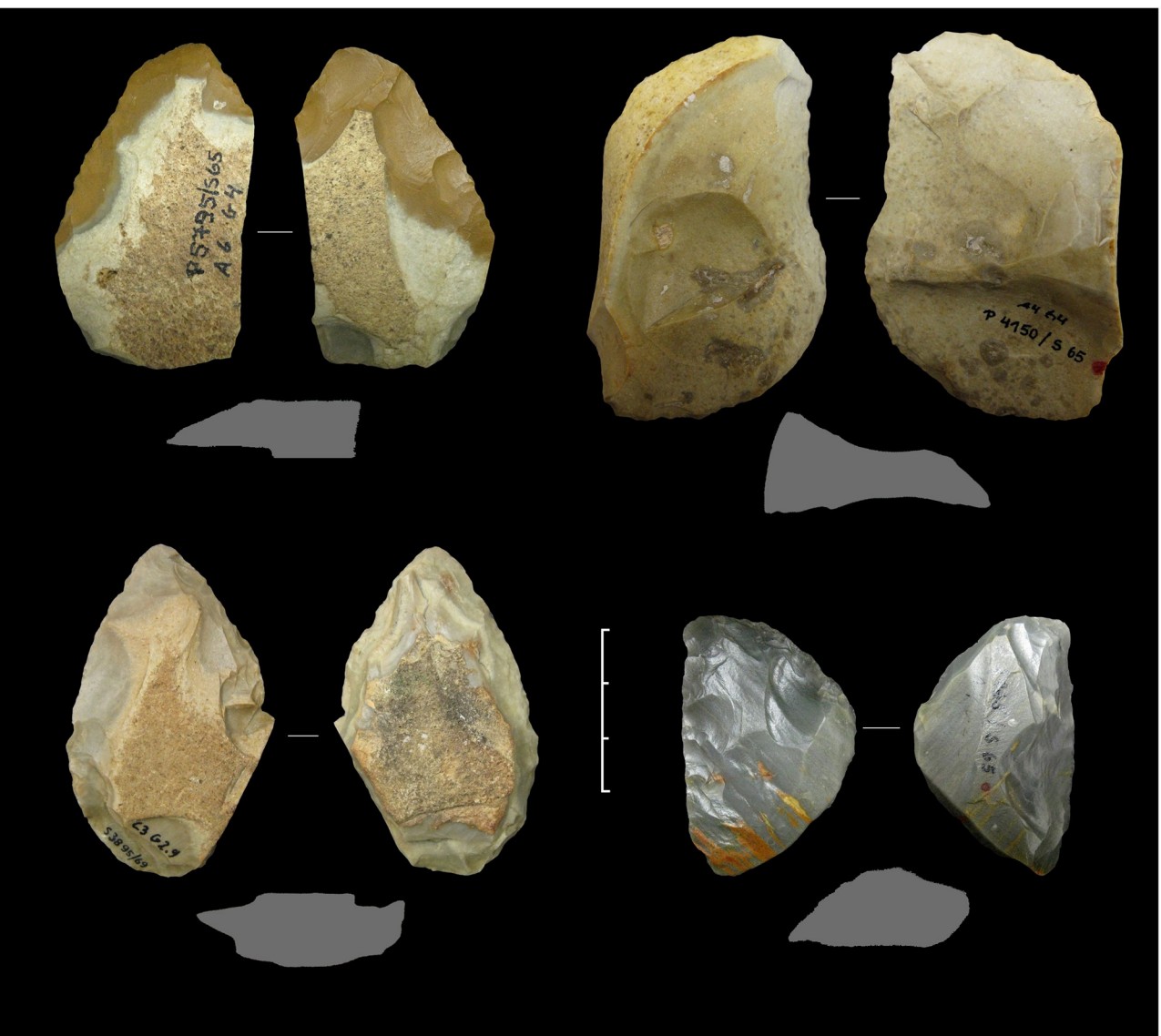

**Fig 7. Keilmesser from Sesselfelsgrotte G-Complexes; from the top left: P5795/S65; P4150/S65; S3895/69; P2965/S65.**

**Table 5. Raw material for techno-typological category of tools.**

|  | Jurahornstein |  | Cretaceous chert/ quartzite |  | Radiolarite |  | Total |  |
|---|---|---|---|---|---|---|---|---|
|  | N | % | N | % | N | % | N | % |
| KM | 40 | **69.0** | 9 | 15.5 | 9 | **15.5** | 58 | 100.0 |
| Scr | 53 | 45.7 | 56 | 48.3 | 7 | 6.0 | 116 | 100.0 |
| Fla | 32 | 20.6 | 122 | **78.7** | 1 | 0.7 | 155 | 100.0 |
| Bif | 15 | **83.3** | 3 | 16.7 | \ | \ | 18 | 100.0 |
| Total | 140 | 40.3 | 190 | 54.8 | 17 | 4.9 | 347 | 100.0 |

Besides the 40 Keilmesser manufactured from Jurahornstein (69.0%), others are made from different varieties of Cretaceous chert or quartzite (n. 9, 15.5%) and from lydite (n. 7, 12.1%) or radiolarite (n. 2, 3.4%). In the case of Cretaceous chert or quartzite, the selection of raw pieces is targeted to larger nodules or plaquettes to produce the larger, thicker and, thus, heavier Keilmesser. Compared to Keilmesser from Jurassic chert, the thickness is 17.7 mm vs 11.5 mm. The average weight of items made from Cretaceous chert and quartzite of 43.9 g is almost double than the weight of items made from Jurassic chert, which is in average 22.3 g. Because of their size, the round raw nodules and the flat plaquettes from Baiersdorf are also preferred for the flake production from cores. Seven Keilmesser are manufactured from black-bluish variety of lydite whereas two of them consist of a red radiolarites. On average, Keilmesser from lydite and radiolarite are small and lightweight (around 4 cm and 12 g, respectively); this is not only a consequence of the small size of the raw nodules, but mainly related to a far-reaching reduction of the tools. Judging from the presence of original surfaces, it becomes clear that in most cases the reduction occurs mainly in length and width, while the thickness (in average: 12.3 mm) is maintained as resource for potential resharpening. In cases were cortical parts allow for a classification, is can be securely said that the raw nodules were fluvial cobbles. The radiolarites originally stem from outcrops in the Alps, but after fluvial transport they are available as small pebbles in the gravels of the river terraces of the nearby Altmühl [27]. Constraints related to the raw size of the blanks must be therefore considered.

*6.1.2.2 Backed scrapers.* In general, backed scrapers are made from plaquettes, technical flakes, natural flakes and pebbles. The majority of the backed scrapers were produced from flakes, which account for 71 pieces or 61.2% (equaling 64.5% if only determined blanks are counted) (Table 4). The blanks used to manufacture these tools are therefore usually embedded in a technical system of the *débitage*. In this regard, the presence of three Kombewa-type flakes should be mentioned. However, the majority of blanks for backed scrapers is represented by technical flakes to remove the core–edge or partially corticated flakes. These are transformed into simple scrapers with rectilinear or convex edge through simple or scaled direct retouch. In most cases, the back consists of cortical surfaces or fracture planes, but in some cases it is pre-determined by negatives resulting from previous knapping of the core. Among the directly shaped blanks (*façonnée*), several frost flakes (N = 17 or 14.7%) are reported. These are well suitable blanks, since they can be used immediately after a simple direct lateral retouch, making it a very expedient conception that leaves cortex on most of the upper face. In addition, also plaquettes are directly used for scrapers without much investment. In these cases, the natural fractures served as striking platform for the retouch of the working edge, which is—when necessary—bifacial.

Most of the flake blanks used for the production of backed scrapers consist of Cretaceous and Jurassic raw materials in almost equal measure (Table 5). Flake *débitage* is also present in the reduction of Jurassic plaquettes, where mainly flakes from very early stages are used. However, directly shaped plaquettes and frost flakes are nevertheless preferred. In contrast, blanks from Cretaceous chert and quartzite mainly come from flake *débitage* is produced in every stage of the core reduction. Thick flakes from Quina cores technology or thinner pieces from Levallois centripetal and unipolar reduction patterns are clearly preferred. As for Keilmesser, the backed scrapers made from Cretaceous chert are larger, thicker and heavier than those from Jurassic chert.

Blanks from other raw materials are present in much smaller quantities. The four radiolarite specimens are slightly thinner and more elongated if compared to the three lydite pieces. In general, these tools were obtained either from flakes or directly from the shaping of pebbles. Despite the fact that the original blanks are heavily reduced, very piece still bears traces of fluvial neocortex, showing the overall low investment in the manufacture of the original blank.

*6.1.2.3 Backed flakes*. It is already implied by the definition that backed flakes are produced in the course of different *débitage* technologies. Among these, the reduction of raw nodules and plaquettes dominates, but in six cases the blanks stem from the reduction of other flakes in the frame of a *kombewa*-like conception (Table 4). Most of the original blanks are represented by *débordant* flakes (core-edge removal f.), which removed a knapped or partially cortical core-edge in the course of centripetal, unipolar or orthogonal *débitage* systems. Blanks resulting from the Levallois concept were securely identified in slightly less than thirty cases. Levallois blanks are either large quadrangular or oval core-edge removals or *dejeté* points and flakes. Fifteen pseudo-Levallois points, typical of the Discoid method as well as centripetal core reduction, are also present.

Among the backed flakes, the frequency of Cretaceous chert is five times higher than that of Jurassic raw materials (Table 5). This means that a choice opposite to the one represented in the Keilmesser is documented. The different raw materials also lead to differences in the variability of the blank dimensions and weight: the flakes produced in Cretaceous chert are in average larger (48.5 x 35.3 mm and 17.4 gr) than in Jurassic chert (41.0 x 28.4 mm and 9.8 gr). From the selected sample, only one flake is manufactured from radiolarite collected on palaeosoil.

*6.1.2.4 Control group*: *Other bifacials*. The strategies recorded for bifacial tools with a basal or lateral prehensive part are similar to those of the Keilmesser. The vast majority is produced from plaquettes. Only two specimens are on flake. With regard to the raw material, Jurassic chert is by far the most preferred material (83.3%) (Tables 4 and 5). There are no particular morpho-dimensional differences between the different lithotypes: all of them are bifacially worked until the anticipated shape was reached (*Faustkeilblatt*, *Halbkeil*, *Fäustel* or generic bifacial tools).

## 6.2 Techno-functional data

**6.2.1 Techno-functional schemes.**   Distinct combinations of morpho-functional features allowed to distinguish at least seven different techno-functional schemes (TFS), which include some variants (Fig 8). All schemes are characterized by at least the combination of back (forming the prehensive unit) opposite to a cutting edge (forming the transformative unit); the first three schemes also have a distal point.

*6.2.1.1 Techno-functional scheme 1* (*TFS 1*). Artefacts belonging to TFS 1 are characterized by the presence of at least one lateral prehensive unit opposed to a transformative unit, both connecting to a distal point. Given the features of single units, the point is a robust trihedron with a transverse cross-section that shapes a rectangle or a scalene (acute-angled) triangle. The morphology oscillates from trapezoidal to triangular with a variable longitudinal asymmetry ranging from minor to pronounced. If bifacially worked, artifacts of the TFS 1 recall *Balver Messer* type of Keilmesser [12].

*6.2.1.2 Techno-functional scheme 2 (TFS 2)*. Items following the TFS 2 have at least one lateral prehensive unit opposed to a transformative unit. In contrast to TFS 1, the distal point is formed by the connection of two cutting edges, the main one and the distal posterior part [66], and is therefore flat. The general morphology is asymmetric. Two variants can be distinguished. Variant TSF 2a has a trapezoidal to triangular shape. If bifacial, this scheme resembles Keilmesser types such as *Klausennischemesser* or, if the elongated and flat, *Königsauemesser* with a very short back. Artifacts of variant TFS 2b have a more rounded distal extremity, because the point in variant TFS 2a is replaced by a convex *tranchant*. It can be assumed that the convex outline of the distal part is caused by subsequent stages retouch and reshaping.

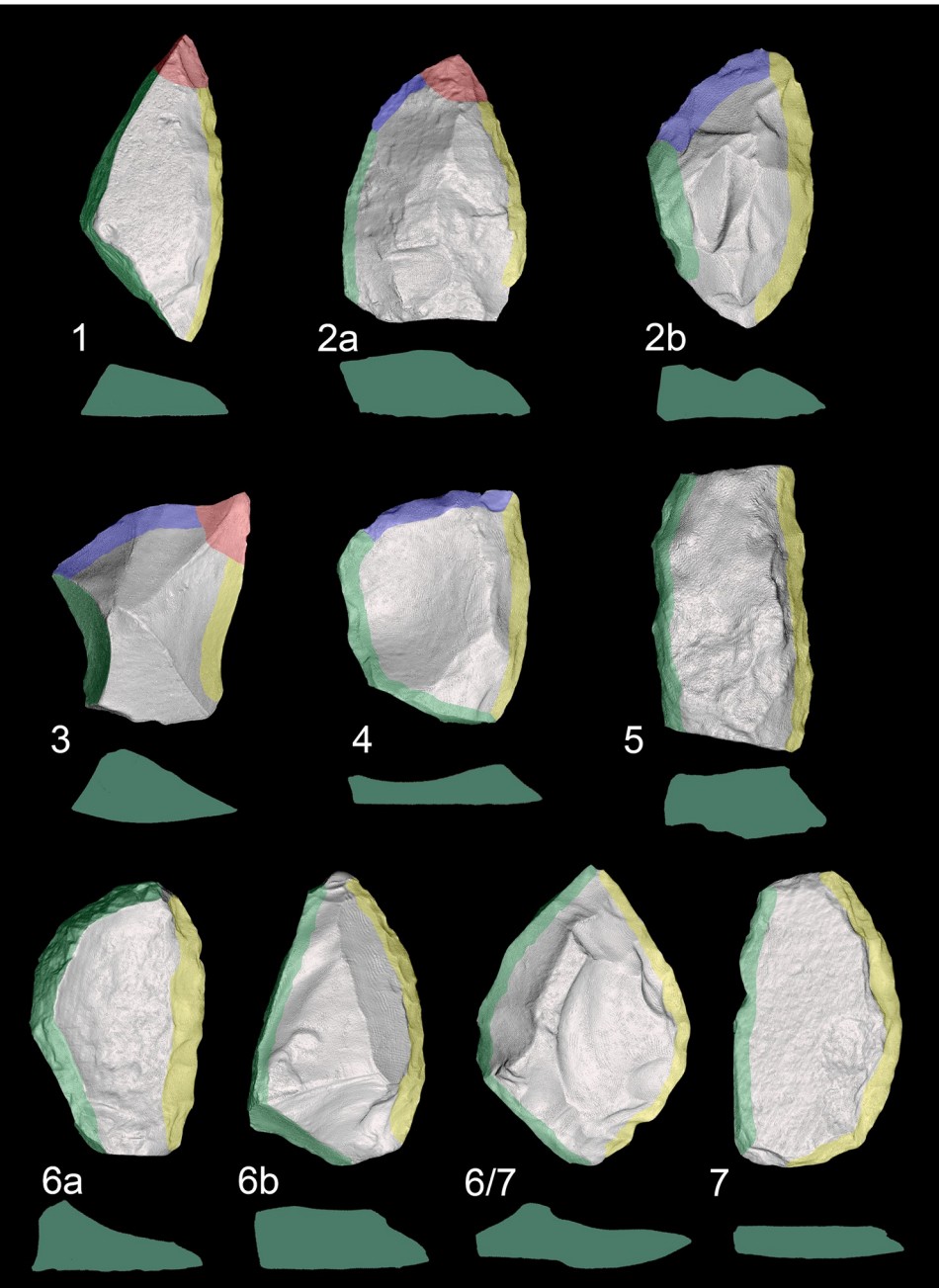

**Fig 8. Techno-functional schemes evidenced on samples' 3D scans.** The point is marked in red, the main cutting edge in yellow, the distal cutting edge or bow in blue, the back in green.

When bifacial, items of TFS 2b are similar to *Lichtenberger* Keilmesser, even if the variability of the latter could fall also into TFS 2a.

*6.2.1.3 Techno-functional scheme 3 (TFS 3).* Artefacts of TFS 3 are usually asymmetric with a tool axis strongly divergent from the knapping axis ("off-axis"). A (usually small) latero-proximal back is opposed to a point, which is formed by two cutting edges. TSF 3 is often found on pseudo-Levallois points or related artefacts.

*6.2.1.4 Techno-functional scheme 4 (TFS 4)*. Items of TFS 4 are short and quadrangular pieces. In cases of TFS 4a, they're characterized by two thick prehensive contact units at the base and one being a lateral back opposite to two sharp transformative contact units corresponding to the distal and one lateral edge. This is a particularly distinctive scheme found on short core-edge removal flakes. The TFS 4b variant is composed of more irregular forms that have in tendency a triangular outline provided by one prehensive contact and two adjacent transformative contacts, which do not form a point.

*6.2.1.5 Techno-functional scheme 5 (TFS 5)*. This scheme includes mainly rectangular elongated artefacts characterized by two lateral contacts units, one being prehensive and the other one transformative. The functionally corresponding units are usually straight and parallel to each other, but located on the opposed sides of the blank. The pieces are symmetric in morphology, but not in functionality, and consist mainly of elongated and rectangular core-edge removal flakes (e.g. *débordant* products obtained from the unipolar Levallois method).

*6.2.1.6 Techno-functional scheme 6 (TFS 6)*. Pieces of TFS 6 are typically elongated and characterized by a prehensive contact unit formed by a lateral back with a convex to curved outline. The opposite transformative contact unit is generally straight. The final morphology of variant TFS 6a recalls a "lunate"-type (if the back is curved) or a trapezoidal one (if the curves are sharp). Among Keilmesser, the Buhlen type is quite similar. The second variant, TFS 6b, features a convex to curved back formed by two adjacent prehensive contacts, one is the base and the other one is on the lateral side, like in the *Bocksteinmesser* [12]. This variant lacks a point and the cutting edge can be slightly convex.

*6.2.1.7 Techno-functional scheme 7 (TFS 7)*. Conversely to scheme TFS 6, these quite elongated artefacts have a mostly rectilinear lateral back representing the prehensive contact unit, which is opposed to a transformative contact unit formed by a convex cutting edge. The pieces are asymmetrical and usually semicircular.

*6.2.1.8 Techno-functional scheme 6/7 (TFS 6/7)*. TFS 6/7 stands halfway in between the aforementioned schemes TFS 6 and TFS 7 and includes pieces characterized by a back and cutting edge that are both convex and curved, and an absence of a distal point. The morphology is oval to rhomboidal and more symmetrical than TFS 6 and TFS 7.

**6.2.2 Tool types and techno-functional schemes.** It is evident that Keilmesser are distributed only in TFS 1, TFS 2, TFS 6 and TFS 7, including their variants. TFS 3, TFS 4 and TFS 5 almost exclusively occur on flake blanks (Table 6). At the same time, the schemes observed on Keilmesser show the largest variability. The variability concerns the presence or absence of a point, the different paired or opposed contact units and possible combinations that make them a multi-purpose tool. Several Keilmesser types are defined through the presence of three features [23]: the back, the cutting edge and the distal posterior part, which may end in a point depending on the degree of exhaustion of the tool. In fact, almost 40% of the Keilmesser are included in TFS 2a and 2b. In cases when the distal posterior part is lacking, the Keilmesser are equipped with a robust point (26.8%) or are simple asymmetrical bifacial knives with different morphologies (TFS 6, 7 and variants).

Backed scrapers are very well distributed in all schemes and show a large variety in raw materials, blank types and shapes. Very often, they share convex profiles (TFS 6, 6/7 and 7) and short shapes. In some cases, they follow the typical schemes of Keilmesser, from which they differ only with regard to the typological classification, which is based on different supports, retouch types or the shape of the cutting edge.

Because of their conception and production embedded in centripetal or unipolar débitage technologies, backed flakes are characterized by straight or convex *débordant* shapes. Among backed flakes, TFS 7 and TFS 5 are very common, but due to many short and quadrangular supports, TFS 4 is also frequently present. In addition, there are items that belong to TFS 3

**Table 6. Techno-functional schemes of tools showed for techno-typological category.**

| TF scheme | Artifact types | | | | | | | | | |
| | Keilmesser | | Backed Scrapers | | Backed Flakes | | Other Bifacials | | Total | |
| | N | % | N | % | N | % | N | % | N | % |
|---|---|---|---|---|---|---|---|---|---|---|
| 1 | 15 | *25.9* | 13 | *11.2* | 12 | *7.7* | 2 | *11.1* | 42 | *12.1* |
| 2a | 11 | *19.0* | 7 | *6.0* | 3 | *1.9* | 2 | *11.1* | 23 | *6.6* |
| 2b | 12 | *20.7* | 10 | *8.6* | 4 | *2.6* | 3 | *16.7* | 29 | *8.3* |
| 3 | \ | \ | 2 | *1.7* | 10 | *6.5* | 3 | *16.7* | 15 | *4.3* |
| 4a | \ | \ | 7 | *6.0* | 13 | *8.4* | \ | \ | 20 | *5.8* |
| 4b | \ | \ | 9 | *7.8* | 17 | *11.0* | 2 | *11.1* | 28 | *8.1* |
| 5 | \ | \ | 14 | *12.1* | 32 | *20.6* | 2 | *11.1* | 48 | *13.8* |
| 6a | 5 | *8.6* | 20 | *17.2* | 10 | *6.5* | \ | \ | 35 | *10.1* |
| 6b | 9 | *15.5* | 6 | *5.2* | 4 | *2.6* | \ | \ | 19 | *5.5* |
| 6/7 | 2 | *3.4* | 8 | *6.9* | 13 | *8.4* | \ | \ | 23 | *6.6* |
| 7 | 3 | *5.2* | 18 | *15.6* | 30 | *19.3* | 2 | *11.1* | 53 | *15.3* |
| indetermined (other/Fragm.) | 1 | *1.7* | 2 | *1.7* | 7 | *4.5* | 2 | *11.1* | 12 | *3.5* |
| Total | 58 | *100.0* | 116 | *100.0* | 155 | *100.0* | 18 | *100.0* | 347 | *100.0* |

mainly due to the blank, which are pseudo-Levallois points obtained from discoidal or centripetal knapping methods.

Among the other bifacial tools, several pieces are typical representatives of TFS 2 and thus share important defining features with KMs. Other items have triangular shapes found in TFS 3.

It is noticeable that not a single TFS is exclusively correlating to one tool type only. However, the combination of this data with a detailed analysis of the specificities of the single contacts units allows better to isolate the desired tools from a techno-functional point of view.

**6.2.3 Tool types and prehensive techno-functional units: *Contact préhensif*—CP.** The data on the conception of the passive portion of the tools (*Contact Préhensif*"–CP) reveals a prevalence of natural backs for Keilmesser and backed scrapers (Table 7). In particular, 58.6% of the backs of Keilmesser are totally or partially made up of natural, often thick parts. Among the backed scrapers, even more than 60% show this conceptualization of the back. In these cases, the adaptation of the tool concept to existing surfaces of the raw pieces is also proven by a discrete number of backs selected on possibly natural conchoidal fractures very common in Jurahornstein plaquettes. A back consisting of negatives already existing on the selected flake and thus blank stemming from the respective method of the previous core reduction, are typical of backed flake tools. As a matter of fact, several the Mousterian *débitage* technologies

**Table 7. Technological conception and nature of backs for tool techno-typological category.**

| CP Type | Keilmesser | | Scrapers | | Flakes | | Other Bifacials | |
| | N | % | N | % | N | % | N | % |
|---|---|---|---|---|---|---|---|---|
| Natural | 30 | *34,5* | 65 | *40,6* | 50 | *27,3* | 6 | *31,6* |
| Natural/fracture | 3 | *3,4* | 3 | *1,9* | 2 | *1,1* | \ | \ |
| Natural/Débitage | 18 | *20,7* | 32 | *20,0* | 24 | *13,1* | 6 | *31,6* |
| Fracture | 2 | *2,3* | 9 | *5,6* | 3 | *1,6* | \ | \ |
| Fracture/Débitage | 15 | *17,3* | 13 | *8,1* | 7 | *3,8* | 5 | *26,3* |
| Débitage | 16 | *18,4* | 31 | *19,4* | 96 | *52,5* | 2 | *10,5* |
| Prepared by retouch | 3 | *3,4* | 7 | *4,4* | 1 | *0,6* | \ | \ |

include the detachment of *débordant* products in order to maintain the peripheral convexities of the core. These are the cases of the Discoid methods and the Levallois concept [72,103,104], as well as the Quina method [70], that are Qall attested at the site. Unaware of the concept of core reduction, the presence of cortical and natural surfaces on the back of about 40% of the backed flakes indicate that a considerable number of these artefacts were manufactured in the course of the first stages of core reduction.

While the thickness of the back, which is on average between 8 and 9 mm, records little variation between the tool types, data on the angles of the back highlight a faint asymmetry in cross section. This is best visible in the back of Keilmesser, where the lower angles are in average slightly more acute than the upper ones (88˚ vs 95˚). This is most probably a consequence of the plano-convex/plano-convex configuration of many pieces, which provides a further cutting edge in the case of exhaustion of the main one. This asymmetry is almost absent in scrapers and flakes.

The back is generally predetermined, either by natural surfaces or by the shape of the prepared core from which the blank had been detached. A post-determination of the back is rare, but present. Many technical arrangements are adopted to further configure, adapt or accommodate the already existing back. Among the analyzed material, Type 1 and Type 2 according to the classification by Delpiano et al. [75] are recorded. A direct intervention at the back through abrupt or semi-abrupt retouch is noticeable in 26 cases, which are mostly backed scrapers (Fig 9). This retouch, which is mostly partial, has been observed macroscopically only and has not yet been the subject of use-wear analyses. Beyond that, Keilmesser are to different degree characterized the thinning of the lower face, often associated with the abrasion and/or the preparation of a striking platform on the back, executed from the opposite surface.

This evidence links some of these pieces to the conception of core-tools, with the back having the not only the function as a prehensive contact units, but also that of a striking platform. Among the material from Sesselfelsgrotte, a peculiar preparation can be noted in cases where two (often orthogonal) backing units conjoin. In 18 Keilmesser, nine backed scrapers and one flake, the angle at this junction is modified through direct, inverse or (very often) bifacial detachments, producing an acute bifacial angle between the two backs (Fig 10). In eight Keilmesser and four scrapers, this specific kind of bifacial modification is only visible at the base.

Finally, the prehensive techno-functional unit of backed flakes is often left unretouched. It at all, only abrasion can be noted. Larger negatives on the back are attributable to the previous configuration of the periphery of the core from which the flakes have been produced.

**6.2.4 Transformative techno-functional unit: *Contact Transfomatif*—CT.** It is particularly significant to analyze and compare the tools' cutting edges (*Contact Transfomatif*—CT) under the perspective of the active angles and the presence or absence of a dihedral bevel, factors that greatly affect the functionality of past and present knives. The angles of the cutting edge show almost identical values if Keilmesser and backed scrapers are compared: the tools have angles of the cutting edge ranging between ~20˚ and ~70˚, while the mean is sligthly under 50˚; statistical tests confirm no differences between scrapers and Keilmesser (Table 8; Fig 11). These values are in stark contrast to the average angles recorded on flake tools of about 10˚. The low angles certainly result from the nature of these artefacts, which are almost always not retouched and therefore preserved the original active angle since the detachment form the core (unless micro-chipping). To the contrary, the post-determined creation of the CT by means of retouching typical for the manufacturing of Keilmesser or backed scrapers inevitably produces larger wider angles. A wide angle could be the main objective of a modification if the planned function of the CF is scraping. Alternatively, a wide angle could also be a consequence of resharpening after a cutting tool lost its sharpness and further retouching was

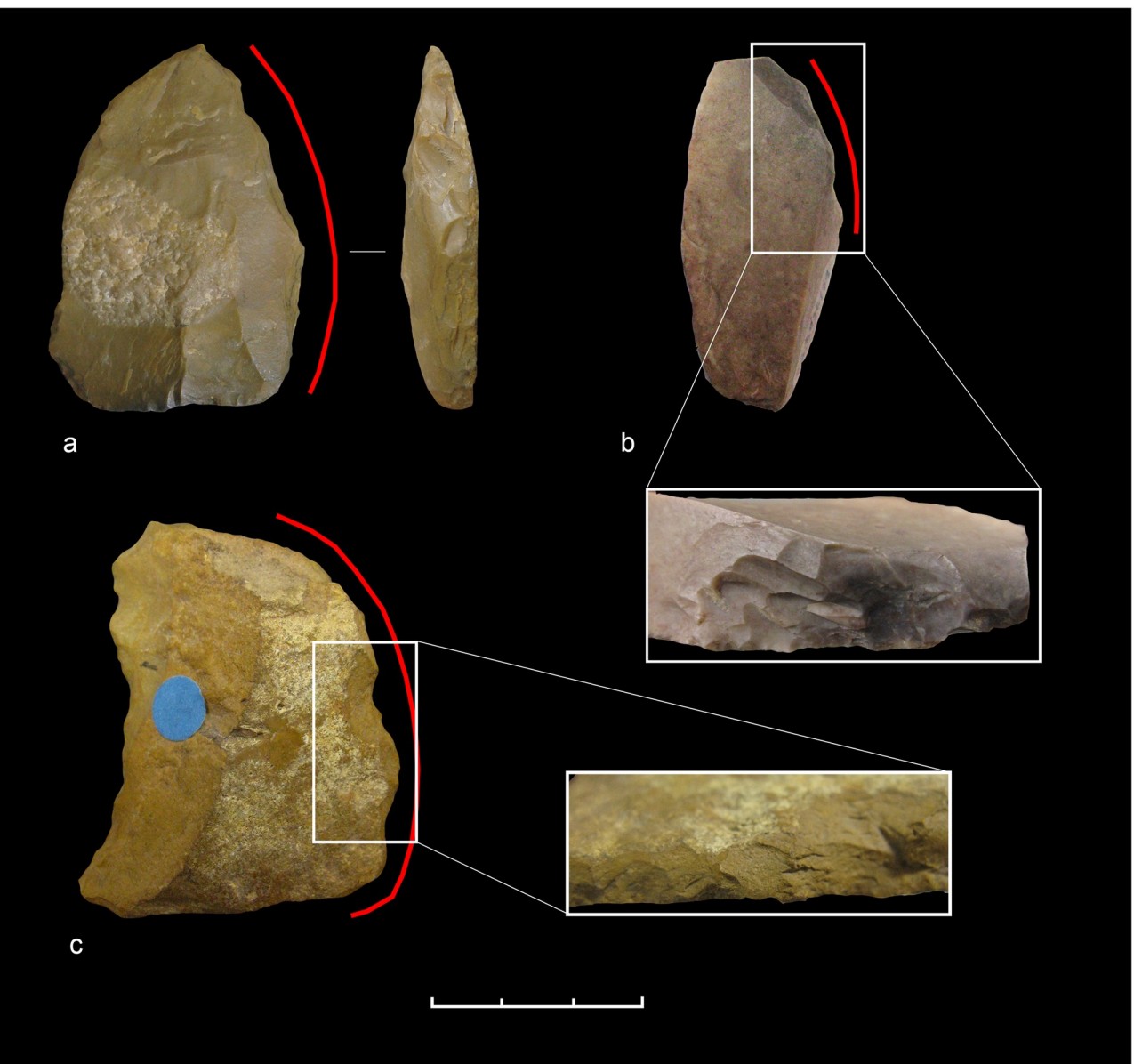

**Fig 9. Scrapers showing prepared back by means of total (a, c) or partial (b) direct retouch.** The specimens are S8918/73(a), S6436/71(b), P1551/S64 (c).

needed to avoid discard. In the literature, different arrangements to main the angle of cutting edges are reported [53,61,64].

With the help of transverse cross-sections, data on the bevel have been recorded. The cross sections confirm a certain degree of difference between the lower and the upper side (Fig 12). On their lower sides, Keilmesser share a high frequency of plano-concave bevels (class 3) followed by plane ones (class 4). Most upper sides of the bevels are plano-convex (class 5) and, to lesser extent, convex (class 7). The backed flakes and backed scrapers show similar values regarding the lower bevel that, probably due to the dominance of flake blanks, are often plane or plano-convex in the proximal sections caused by peripheral convexities of the bulb. On the upper sides of flakes, the bevels are rather plano-convex, whereas those of backed scrapers are

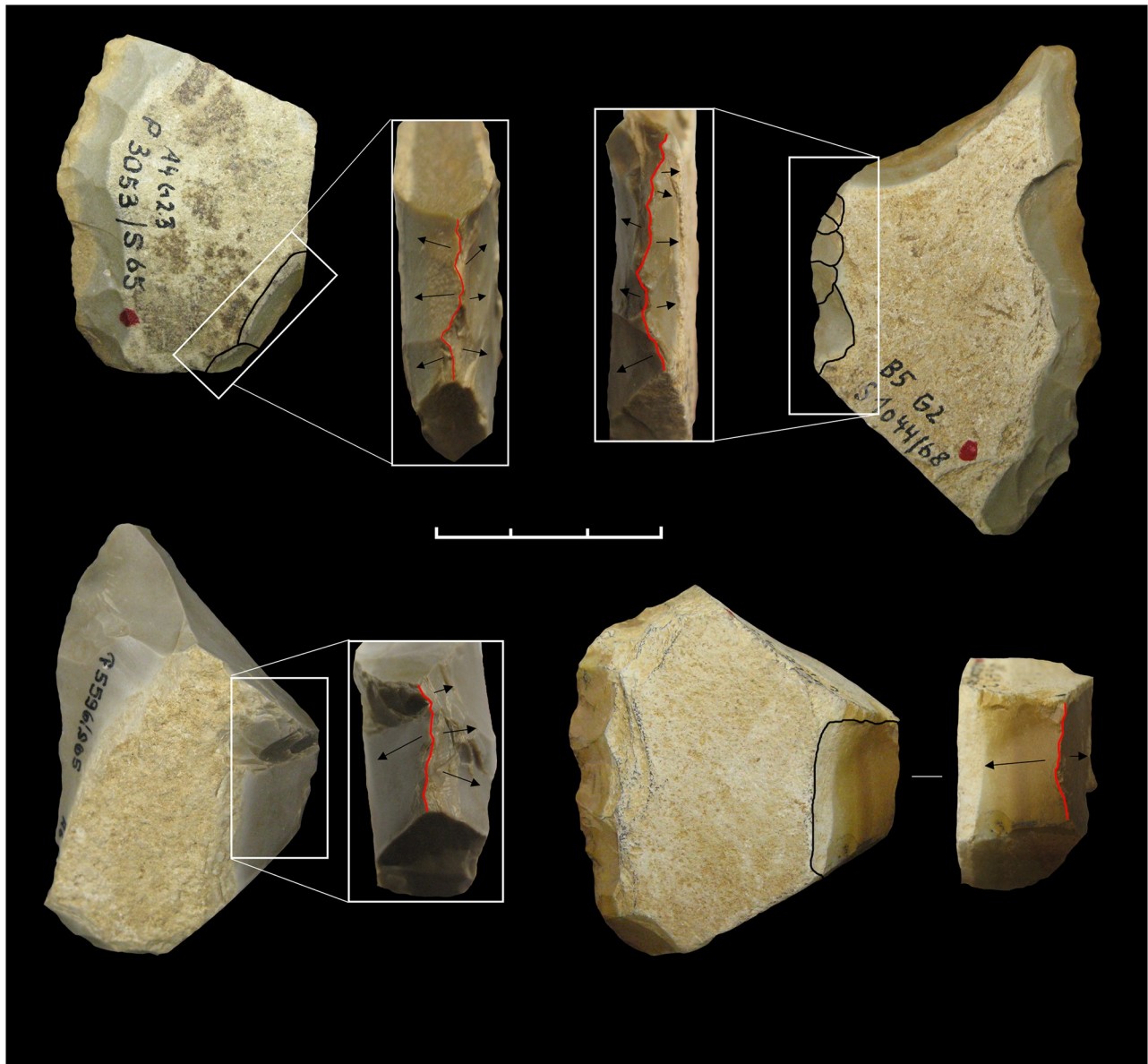

**Fig 10. Keilmesser and scrapers showing bifacial or unifacial detachments on the conjunction between the backed portions.** From the top left: P3053/S65; S1044/S68; P5596/S65; S5234/70.

convex due to the different order of the retouch. In sum, it is difficult to decide if the type of bevel is one objective of the configuration of the tool and, thus, aimed to meet a certain anticipated tool function, or if it is a situational consequence of the manufacture and use of the tool.

The data described above is directly connected with the origin and conception of the CTs, and therefore, with the typological classification. As previously said, the CTs of the Keilmesser are always post-determined, e.g. they are manufactured by retouch, which is almost always bifacial: concave and invasive in the case of the lower face, and convex on the upper. Backed scrapers are also characterized by a post-determination of the CT consisting of different types of scaled and stepped retouch, which in 25 cases (21.5%) is bifacial. Backed flakes are mainly characterized by a CT predetermined by the preparation of the core at the moment of the

**Table 8. Data on cutting edge mean angles of Keilmesser, scrapers and backed flakes.**

| Data on mean angles | Keilmesser | Scrapers | Flakes |
|---|---|---|---|
| N | 68 | 138 | 192 |
| Min | 27.5 | 17.5 | 15 |
| Max | 72.5 | 75 | 72.5 |
| Sum | 3332.5 | 6685 | 7200 |
| Mean | 49.00735 | 48.44203 | 37.5 |
| Stand. Dev | 9.888744 | 9.863128 | 10.303 |
| Median | 47.5 | 47.5 | 37.5 |
| 25 prcntil | 40 | 40 | 30 |
| 75 prcntil | 57.5 | 55 | 42.5 |
| Mann-Whitney U test with Keilmesser | \ | $p = 1$ | $p = 6.211E-12$ |

blank detachment; the few post-determined CTs (4.5%) or CTs with both modes of determination (10.3%) show a marginal, often inverse, retouch that has a little influence on the general morphology of the CT.

### 6.3 3D morphological data

Morphological analyses were carried out with the software AGMT3-D on 193 3D-models. In general, the morphological variability of scrapers and backed flake tools is higher than that of Keilmesser, respectively of about 12% and 18% (Table 9); this value is measured as the mean multidimensional Euclidean distance of the items in the group from the group's centroid (i.e. mean shape): the higher the value, the higher the variability [101].

The same morphological data was analyzed with the help of a principal component analysis. The visualization of the results is a scatter-plot presenting the distribution of the artefacts along the first two principal components, which alone account for 46.71% of the total variability. The two PC record changes in shape that are mainly located on the edge curvature (PC1) or on the base and the top (PC2) (Fig 13a). Keilmesser occupy a comparably small and more

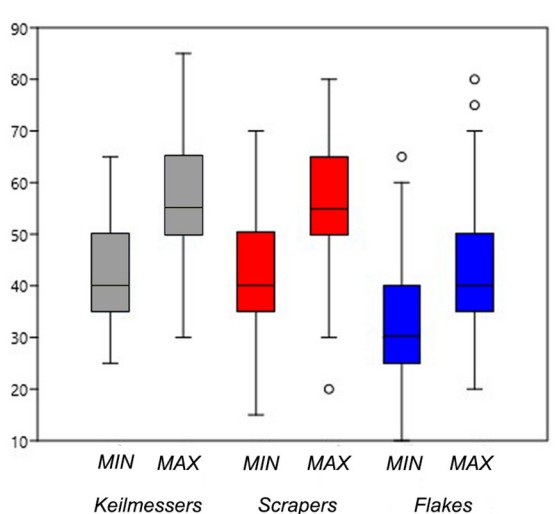
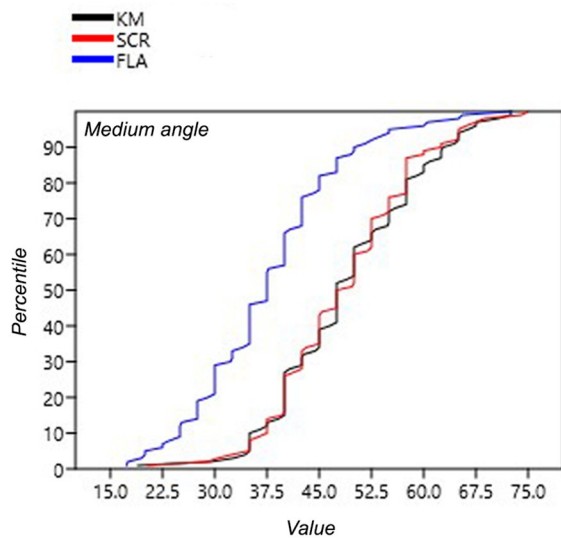

**Fig 11. Boxplot and percentile charts showing the distribution of cutting edge angles among tool types ("KM": Keilmesser; "Scr": scrapers; "Fla": flakes).**

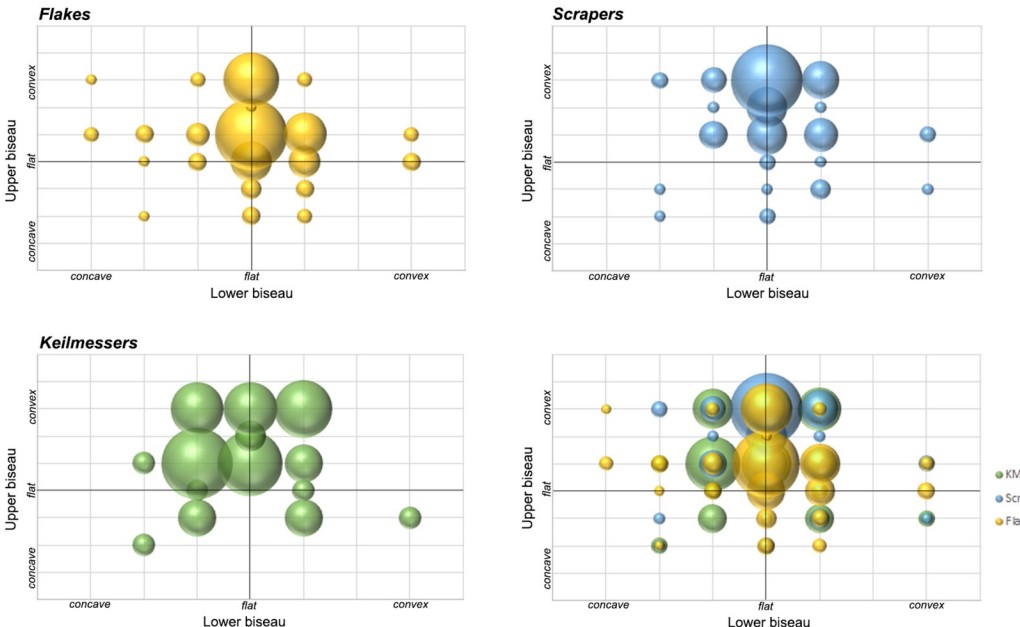

**Fig 12. Bubble charts showing the distribution of the classes of cutting edge bevel (lower and upper) for tool type.**

or less central area of the plot, confirming a certain level of standardization (Fig 13a). This is despite the fact that the number of analyzed 3D-models of Keilmesser is substantially higher than the number of other bifacial tools and also higher than the number of backed flakes.

Given the large numbers of both Keilmesser and backed scrapers, the observation of only minimal differences between the two tool classes gains more robustness. The results of different Wilcoxon rank-sum tests on the Interpoint Distances between Group Means and Centroid Sizes confirm that the mean shapes, shape variabilities and centroid sizes between Keilmesser and scrapers are not significantly different. The distance is greater between backed scrapers and backed flakes, unaware of the fact that they often share the conception of the starting blanks, and much greater between Keilmesser and backed flakes as well as between Keilmesser and other bifacial tools (Fig 13b and 13c). It follows that scrapers and Keilmesser are from a morphological point of view very similar. A comparison between their mean shapes highlight the fact that the latter are just a little narrower, less rounded and have a sharper termination (because of the many pointed tools). The most variable areas are, in fact, the mesial-distal lateral edges (Fig 13c).

There is the possibility that the observed variability derives from factors other than the aim to manufacture an anticipated artifact form. Instead, they could be more linked to the properties of the raw material or the blanks. For this reason, the attributes "blank type" and "raw

**Table 9. 3D statistical analysis on morphological variability and centroid size for tool techno-typological categories.**

|  | N | Variability | Centroid Size |
|---|---|---|---|
| Bifacials | 12 | 1771,186 | 10865,52 |
| Flakes | 47 | 1775,09 | 9787,911 |
| Keilmesser | 55 | 1500,649 | 10036,23 |
| Scrapers | 79 | 1674,179 | 11166,03 |

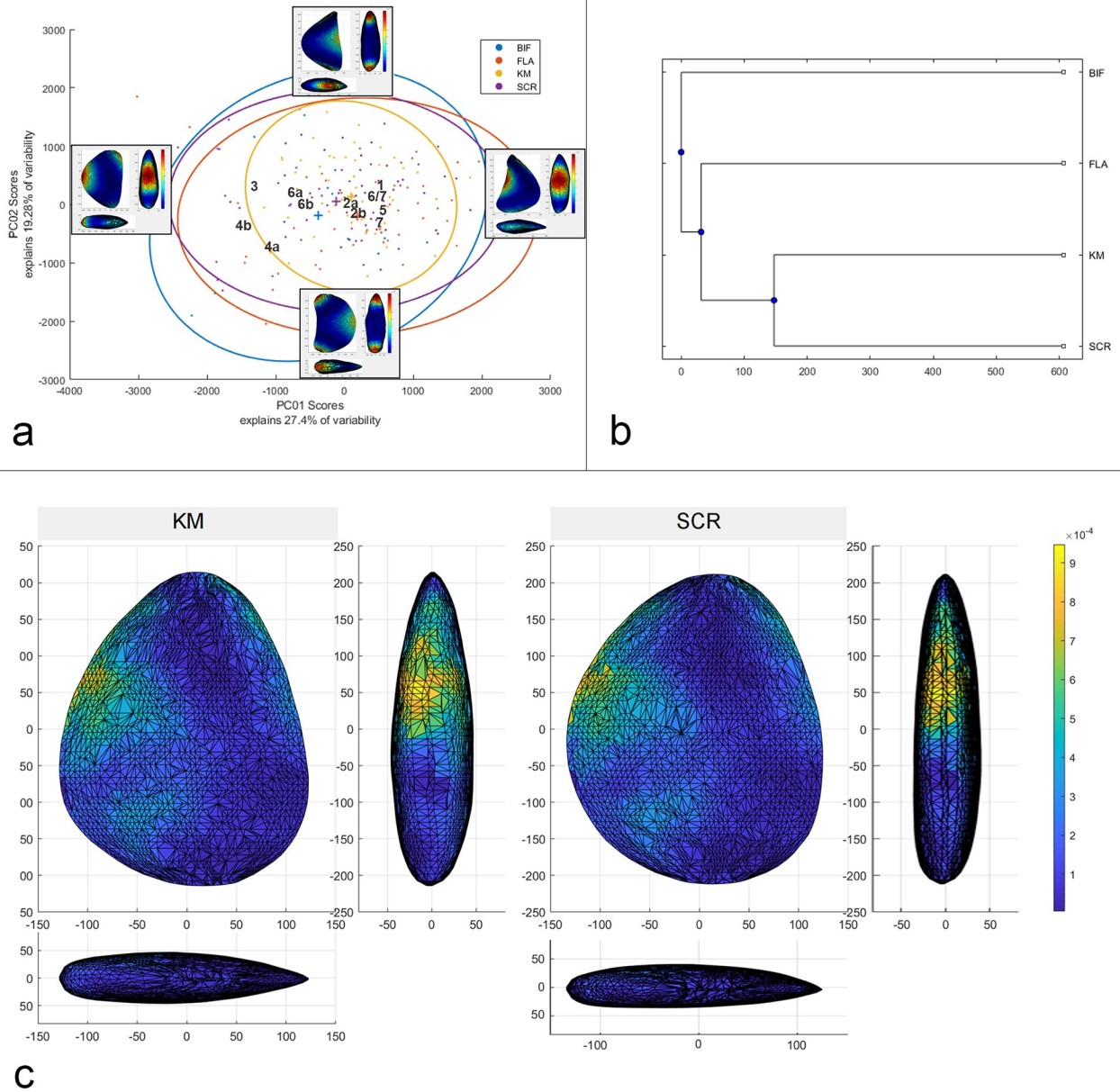

**Fig 13. Statistical morphological analysis on 3D models, compared for tool types: scatter plot based on principal component analysis (PCA); the numbers indicate the localization of the techno-functional schemes' centroids within the PCA (a); group distance between tool types (b) and groups mean comparison between the mean shapes of Keilmesser and scrapers ("KM": Keilmesser; "SCR": scrapers; "FLA": flakes; "BIF": other bifacial tools).**

material" have been integrated in the analyses and were analyzed for the single and combined (with tool types) variabilities. One result is that the morphology of Jurassic chert artefacts is similar to that of Cretaceous chert artefacts. The differences are mainly restricted to the point of the pieces, which is a consequence of the fact that Jurassic chert is the typical raw material for Keilmesser, whereas Cretaceous chert is heavily associated with flake tools where a point is missing. The morphology of artifacts made from radiolarite differs from all other materials, which implies that the raw material properties indeed influence the morphology of the tools (Fig 14a and 14c).

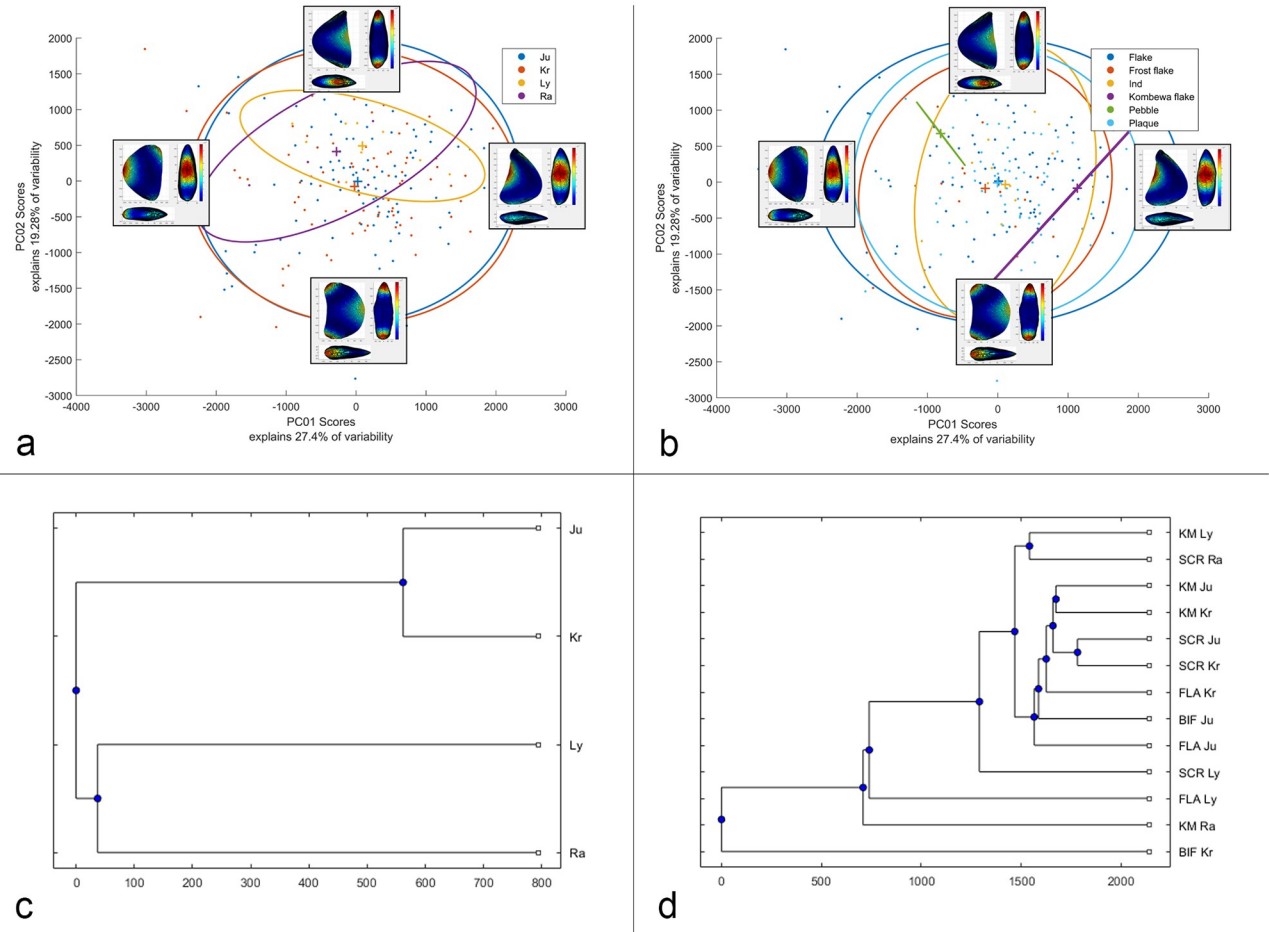

**Fig 14. Scatter plot showing the variability of tools for raw material (a) and raw blank type (b), group distance for raw materials (c) and tool types plus raw materials (d); ("KM": Keilmesser; "SCR": scrapers; "FLA": flakes; "BIF": other bifacial tools; "Ju": Jurahornstein; "Kr": Cretaceous chert and quartzite; "Ra": Radiolarite; "Ly": Lydite).**

If crossed with the data on the tool types, it is evident that the distance between the sub-groups is more related to tool types (with regard to Keilmesser and scrapers) and to a lesser extent depends on the raw material (Fig 14d).

Similarly, if raw blanks are included in the analysis, it can be seen that the majority falls into three categories: flakes, plaquettes and frost flakes. Within the scatter plot, we can see that their medium shape and centroids, according to the two main PCs, are directly comparable (Fig 14b). What is slightly changing is the variability. However, this is directly proportional to the sample-size for each type; the larger the sample-size, the higher is the variability (Table 10).

**Table 10. Morphological variability and centroid size of tools according to the initial raw blank.**

|               | N  | Variability | Centroid Size |
|---------------|----|-------------|---------------|
| Flake         | 96 | 1756,48     | 10421,15      |
| Frost flake   | 16 | 1453,148    | 11397,82      |
| Indet.        | 12 | 1591,752    | 8953,659      |
| Kombewa flake | 2  | 1013,591    | 8540,446      |
| Pebble        | 2  | 892,7342    | 7972,714      |
| Plaquette     | 65 | 1611,204    | 10788,64      |

Therefore, the raw blank only had a minor influence on the morphology, mainly regarding the blank size.

## 7. Discussion

### 7.1 One, no one and one hundred thousand types: Keilmesser in Sesselfelsgrotte G-complexes

The first topic discussed here is the definition of the Keilmesser tool-type according to a techno-functional approach. Despite a proclaimed standardization of these tools [18,23], their undoubted variability in shape led to the creation of several sub-types, which, at times, were interpreted differently, ranging from different chronological and geographical groups [12,50,105,106] to subsequent stages of reshaping and modification [20,23,61] and to distinct functional and/or aesthetical reasons [9,18,23,66,107]. Regarding the last hypothesis, bibliographical use-wear data concerning the Keilmesser from Sesselfelsgrotte confirm, despite the widely accepted notion of a function as hand-held knives, the presence of hafted pieces. In addition, there are examples for functions other than that of a knife: one item was identified as a scraper/rabot, and a second one is a hafted projectile [67].

The results of the analysis of the 3D morphological variability confirm a certain variability within Keilmesser, even if more limited than in the other tool-types. Our research protocol tried to overcome their classic typological division, which is mainly based on the outline, by elucidating their functional history. The cross-check of the identified techno-functional schemes by data on variability resulted in the consistency of three main sub-types (also corroborated by the group distances in the 3D morphology (Fig 15a):

1. One group defined by the actual or past presence of a distal point (schemes 1, 2 and 2b), which represents the majority of the investigated items (n = 38, or 65.6%)

2. One group defined by the presence of a curved, convex or segmented back (schemes 6 and 6b), comprising 14 tools (24.1%).

3. One group with a heavily curved cutting edge and the absence of a pointed end (schemes 6/7 and 7), comprising 5 tools or 8.6%.

A clear dimensional gap can be noticed between these groups, with the pointed-tools being larger in size. Among these, scheme 2 is associated to larger tools and narrower cutting edge angles (Fig 15b and 15c). If we accept the reduction-stage interpretation, this data would hypothetically set scheme 2 as the main initial scheme of Keilmesser tool-type. In this sense, Keilmesser of scheme 2 could be modified by resharpening to scheme 2b in cases when the tip reshaped into a rounded outline (after it became blunt). Alternatively, Keilmesser of scheme 2 could also be resharpened at times into scheme 1 through the gradual backward reshaperning of the cutting edge and the removal of the bow, following already known reduction processes [20]. However, scheme 1 Keilmesser could also represent an initial TFS for tools manufactured on chert plaquettes. In the more advanced stages of reduction, the tip could completely disappears and, based on the initial shape of the blank, schemes 7 or 6/7 and 6 or 6b can be achieved; this may apply especially to the latter, since these are generally smaller tools with wider cutting edge angles (Fig 16). Within each group, there was an increase of the active angles with the decrease in the blank sizes, thus associated to the reshaping of the tool (Fig 17). However, even in the group characterized by a convex back, the bigger tools (already quite small) have narrower angles, suggesting a direct manufacturing that hasn't undergone many resharpening

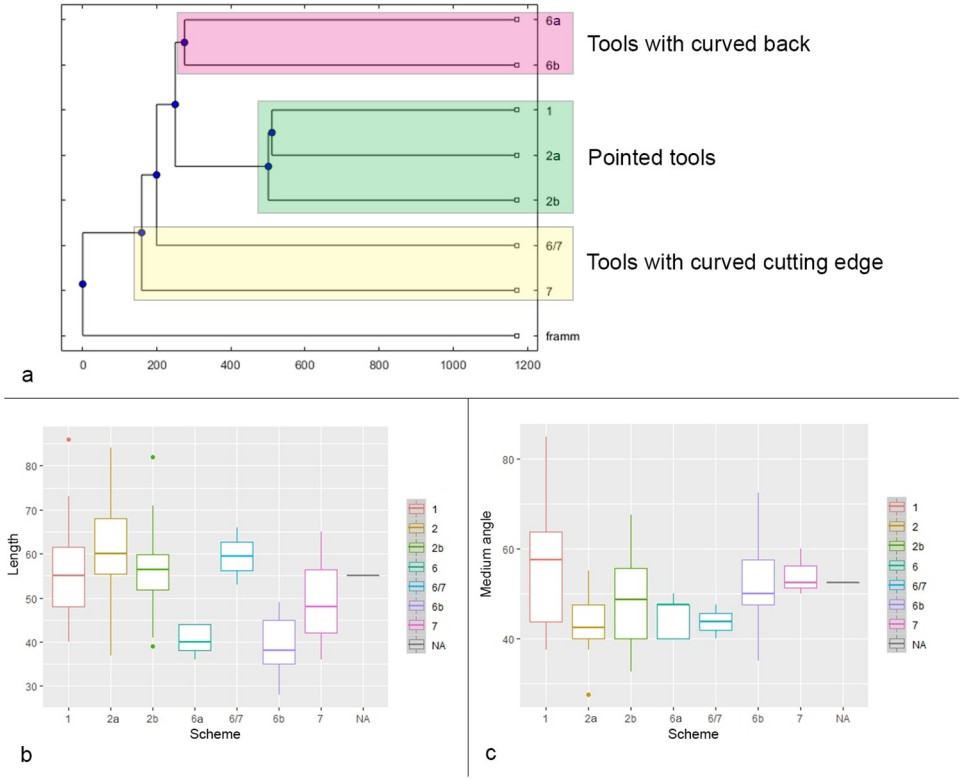

**Fig 15. Variability between Keilmesser techno-functional schemes showed by group distance (a), length (b) and medium angle (c).**

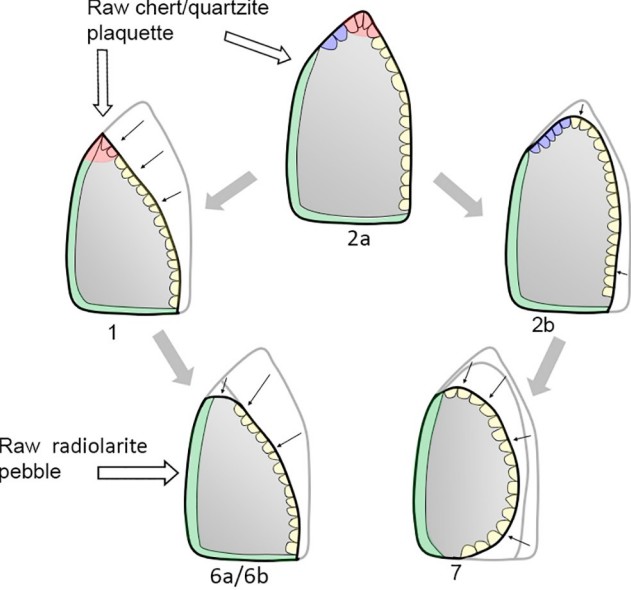

**Fig 16. Hypothetical diagram proposed for the manufacturing of the different techno-functional schemes typical of Keilmesser.** The nature of raw blank and their progressive modification through reduction and reshaping could have been the main reasons of this differentiation.

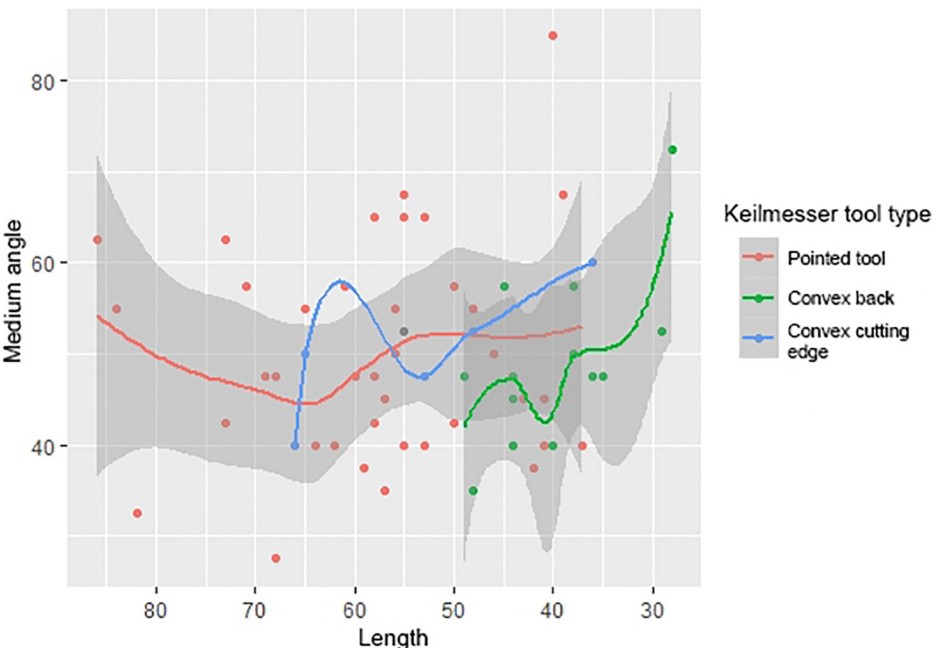

**Fig 17. Distribution of the cutting edge angle of *keimessers'* groups according to the blank length.**

phases. Alternatively, if the resharpening was present, it probably had little influence on angles and therefore on the techno-functional scheme. Therefore, schemes 6 and 6b could also represent Keilmesser variants adapted to small-size blanks and plaquettes made on Jurahornstein and particularly on small Radiolarite and Lydite pebbles, characterized by segmented and strongly convex natural edges (Fig 18). A certain correlation between tool shape, raw material type and dimensions could imply functional and ergonomic reasons: the smaller tools generally need a curved back to facilitate the manual handling [75]. Conversely, pointed Keilmesser are manufactured on a wider variety of blanks and raw materials, including Cretaceous Cherts and quartzites, which are thicker and heavier, allowing for the power needed by a larger cutting tool provided of a perforating point.

It is therefore possible that at the base of the Keilmesser diversification there are different techno-functional, handling and use-related schemes partially adapted to raw materials and blanks, but also technical consequences like the changing and evolving of schemes according to reuse and reshaping. If we apply a basic working-step analysis recomposing the (last) phases of production and modification of the tools, three <u>technological groups</u> are evident: core-tools (kerngeräte), blank-tools and exhausted tools. If we consider the Keilmesser techno-funnctional schemes, that grouping can reflect a direct requirement (both functional and/or ergonomical) as well as a consequence of tools' reduction; the grouping in technological groups, on the contrary, necessarily reflects a need in terms of technological versatility or strategic/economic potentiality, which indirectly affects the object and its morphometry.

In this sense, the <u>blank-tools</u> are pieces where the technical investment seems to be aimed exclusively at the first shaping of the tool from the selected plaquette, the subsequent achieving of a bifacial cutting edge (mostly plano-convex) opposed to a back, and the possible resharpening stages; each identified working stage regards the first manufacturing or the rejuvenation of the tool (Fig 19).

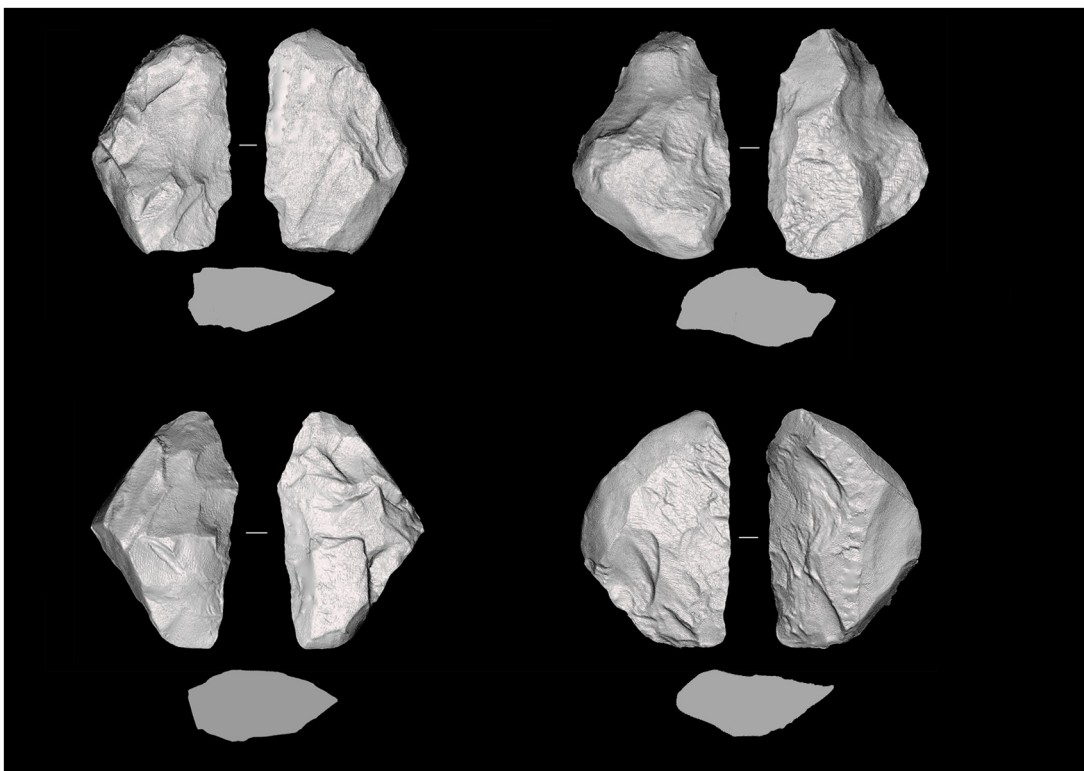

**Fig 18. 3D scans of Keilmesser manufactured on lydite or radiolarite pebbles showing the exploitation of the natural convex surfaces for the conception and the obtaining of the back.** From the top left: S8890/73; S246/68; P1568/S64; P2965/S65.

Conversely, <u>core-tools</u> are implements which also had a probable function as "matrix", source of raw material for producing smaller usable flakes. This ramified sequence has been recognized on 22 out of 58 blanks, with the exploitation mainly noticeable on the lower flatter surface but occasionally even on the upper one. According to the bifacial plano-convex volumetric concept defined by Boeda [53], a flat surface is produced in order to achieve the plano-convex active bevel; in this first stage, flat, wide and deep flakes are detached mainly from the cutting edge and, in several cases, from all over the periphery in a nearly-centripetal surface exploitation pattern, which can also be aimed at the production of functional flakes (Fig 20). Wide and flat detachments are also related to Keilmesser reshaping phases in order to maintain the angle of the cutting edge [61], according to a method applied also to much older bifacial tools [108]. However, an additional techno-economic objective has been recognized on these tools through the work-step analysis. For example, artefact P1963 records a first preparation of the plano-convex cutting edge, followed by a mainly upper reshaping that gradually erases the lower negatives. Flakes are then detached on the lower surface from the back that is used as a prepared striking platform. A certain irregularity in the back thickness suggests that the main goal was the production of flakes also 3 x 2 cm in size (Fig 20a). In artefact P5596, the working and faceting of the back from the lower surface is evident: the preparation of the striking platform and the surface exploitation resemble the Levallois concept (Fig 20c). Artefact P5791 however, records a core-like exploitation that exhausts the blank by removing most of the back, which also has the function of a striking platform (Fig 20b). In these and several other examples, the exploitation is bipolar to centripetal, and the Levallois-like conception is emphasized by the detachments parallel to the blank secant plane, the knapping angles around

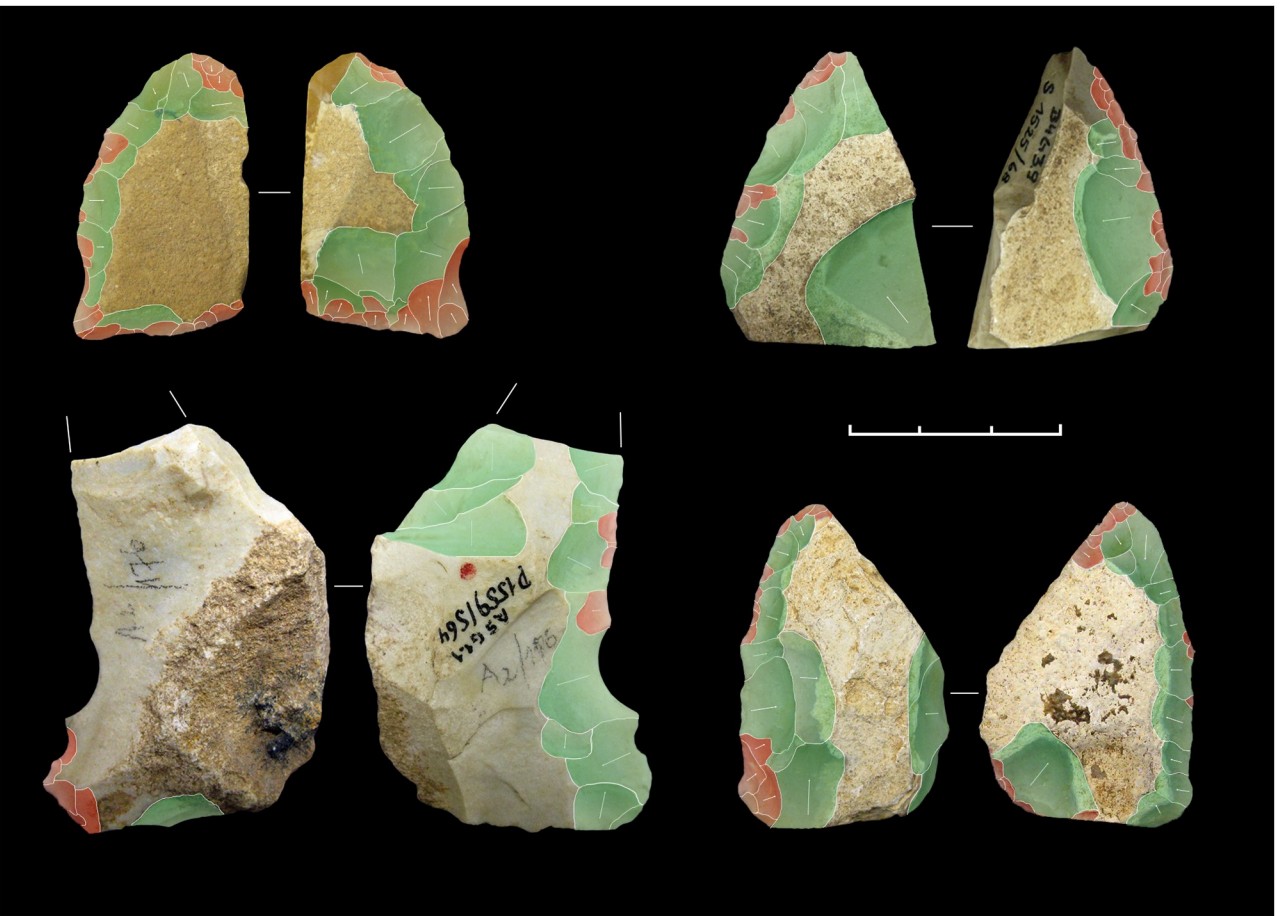

**Fig 19. Keilmesser blank-tools, that is the blanks shaped with the only function of manufacturing an asymmetrical backed bifacial tool.** In green, the detachments aimed to first tool shaping; in red, edge refining. From the top left: S2221/68; S1525/68; P1559/S64; S310/74.

90 degrees and the accurate preparation of the peripheral frame in correspondence to the blank's back [73]. This working can be applied, in some cases, also to the opposed surface thanks to the double plano-convex volumetric conception of the Keilmesser-tool [53]. The obtained products are oval or elongated small to medium sized flakes, suitable for immediate and short-term precision activities framed in situations of compelling needs and a shortage of raw materials. Keilmesser are generally known to be "core-tools"; their volumetric concept is well defined in the "fourth type of bifacial volumetry" based on Kůlna cave tool [53]. Their productive potentiality, derived from the ramification of their surface-exploitation, has been particularly investigated in the Crimean Micoquian; here, the production of usable flakes obtained from surface shaping of bifacial tools is assumed in their first shaping, in the re-tooling process and also after the discharge of exhausted tools through a recycling of the blank [109,110]. Within Central European *Keilmessergruppen*/M.M.O., this practice is rarely reported: some small and worn pradniks from Buhlen have been recognized as recycled cores after an abandonment as tools, but there is no mention of a possible alternation between the two functions [20]. Core tools have been also recognized in Neumark-Nord 2 and Königsaue [111].

Finally, <u>exhausted tools</u> correspond to the extremely reduced blanks, where most of the working stages are no longer visible except the last ones, until the complete exhaustion of the

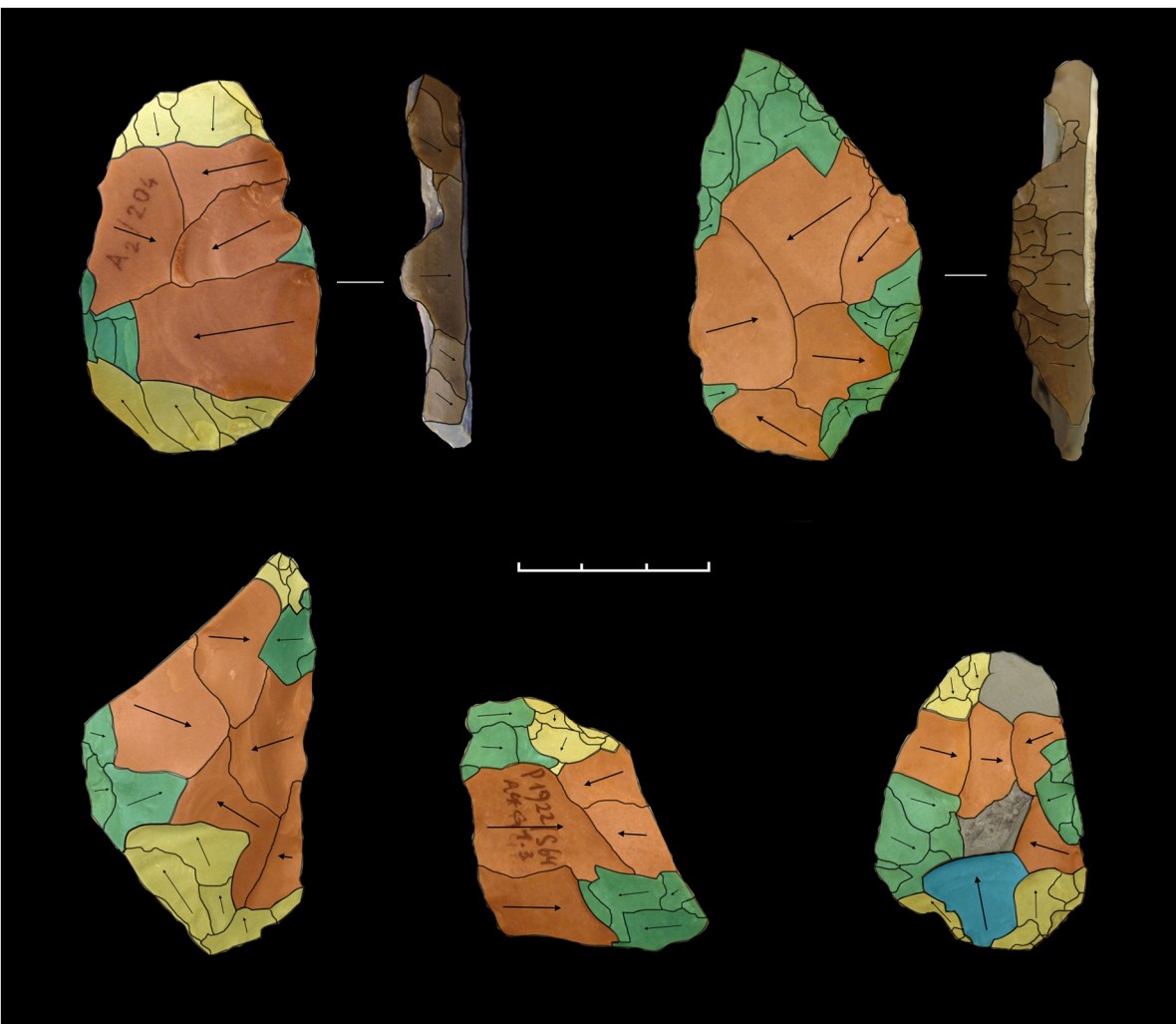

**Fig 20. Diacritic schemes of Keilmesser core-blanks showing the patterns of reduction and exploitation of the lower surface and, occasionally, the preparation of the back, used as striking platform.** The red and blue detachments may represent those aimed at creating usable blanks, while the yellow and green ones at managing the lateral convexities and regularizing the surface. From the top left: P1963/S64; P5791/S65; P5596/S65; P1922/S64; S3887/69.

usable volume. Very often these tools are made out of non-local raw materials, but possibly characterized from the beginning by smaller raw blanks, better suitable to some kinds of shaping (like schemes 6 and 6b) (Fig 18). These artefacts are often present in assemblages of the *Keilmessergruppen*/M.M.O. context: already recognized and defined "small *pradnik* knives" or exploited specimens by Krukowski [60], with the reshaping phases they may reach a reorientation of the blank, which at the end, often assumes a triangular shape.

Within the investigated assemblages from the G-complexes, those belonging to the first stages (layers G2 to G5) show high frequencies of core-tools, and every scheme is represented, indicating thus the requested versatility of these tools in these phases (Table 11). Exhausted tools show an increase in layer G2 and prevail in layer G1, where the longer tool use-life is probably linked to high requirement of raw material and mobility. Meanwhile, few core-tools suggest a lesser need to versatile and multi-purpose Keilmesser in the last layer. If we cross data with Richter's inventories indicative of occupation cycles [9], we note a difference

**Table 11. Distribution of Keilmesser's three different productional operational schemes for archaeological layer.**

| Layer | Blank-tools | Core-tools | Exhausted tools |
|---|---|---|---|
| G1 | 5 | 3 | 10 |
| G2-G3 | 5 | 10 | 6 |
| G4 | 5 | 9 | 3 |
| G5 | 1 | \ | 1 |

**Table 12. Distribution of Keilmesser's three different productional operational schemes for inventory type (from Richter 1997).**

| Inventory | Blank-tools | Core-tools | Exhausted tools |
|---|---|---|---|
| Initialinventar | 4 | 8 | 10 |
| Konsekutivinventar | 12 | 14 | 10 |

between initial and consecutive inventories mostly in the amount of blank-tools, which are three times more frequent in the latter (Table 12). This direct and more exclusive use of the Keilmesser tools attests more functional specialization, which goes well along with the assumed more stable occupation as base camp with logistic mobility recognized in the consecutive assemblages [28,68,85]. The selection of the best raw material [9] for blank-tools may also indicate a certain degree of planning aimed at achieving a greater tool-set effectiveness.

Blank-tools bear different functional operational schemes, but the most represented is by far scheme 1. Whether it has been directly manufactured or achieved after brief re-shaping phases from scheme 2, it is worth highlighting its high functionality, characterized by very distinct and specialized TF units such as the robust trihedron tip (formed by the intersection of three surfaces), suitable for incising, opening and directing the cut; the cutting edge, having medium angles just above 50 ° and a plano-convex bevel, functional to cut with longitudinal movements; the back, usually characterized by a little convex delineation, suitable for a possible manual grasping with the support of the finger. This is generally a very effective tool, mostly manufactured in Jurassic chert.

## 7.2 Differentiation and imitation: Backed artefacts within Keilmessergruppen

If the Keilmesser's internal variability can be related to the reasons discussed above, what is considered Keilmesser or not is an equally intricate issue. The limits of a sharp typological definition are clear, since Keilmesser often overlap with the other types, both in shapes and morpho-functional features. The identified TFS are differently represented within the tool types, but theoretically every backed tool scheme can be fulfilled through the use of non-retouched or retouched flakes, unifacial scrapers or bifacial knives.

This is also true in our case study, where no exclusive schemes for Keilmesser exist, although there are some clear preferences. This data is confirmed by the 3D morphology, where the overlap between types in the PCA representation is evident (Fig 13a). If we investigate and compare the back of these tool types, we can notice a generic difference in its conception even if its dependence from the initial blank is equally important. Jurassic chert plaquettes with natural thick parts are usually shaped into Keilmesser, a behavior that implies wider reasons of raw material management, but they are also used for other bifacial tools and scrapers. Paradoxically, the raw material doesn't seem to considerably affect the final form, since the artefacts produced in the most utilized Jurassic chert (Ju01) are more similar to the ones produced in Cretaceous cherts and quartzites (Kr) instead of other varieties of Jurassic cherts

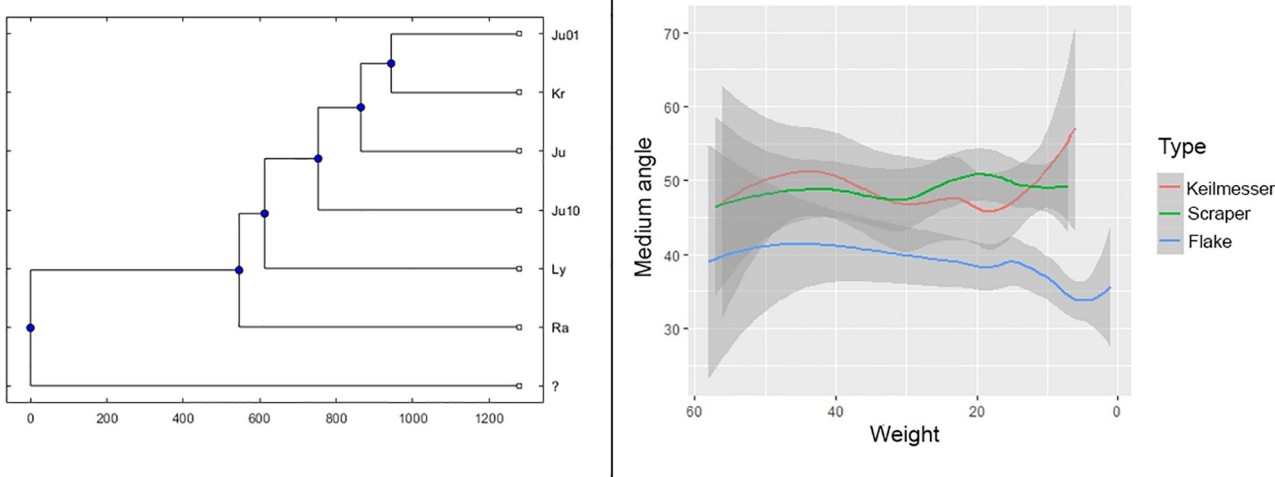

**Fig 21. On the left, group distance for raw material showing differences between the different varieties of *Jurahornstein*; on the right, distribution of the cutting edge angles according to the weigth diminution ("Ju", "Ju01" and "Ju10": Jurahornstein; "Kr": Cretaceous chert and quartzite; "Ra": Radiolarite; "Ly": Lydite).**

(Ju10) (Fig 21a). A typological preference related to the raw material supply can be noticed. Among Keilmesser, a number of radiolarite-lydite specimens are present, while among other bifacial tools they are missing. These are mostly exhausted tools deriving from a long use-life or manufactured on smaller blanks collected in fluvial gravels, probably utilized in periods of scarcity and "emergency situations". A number of scrapers are also made out of red radiolarites, while lydite scrapers are rare. The radiolarites, outcropping in the alps, were exclusively collected in semi-local Danube gravels, while the lydite outcrops in north-eastern Bavaria and was probably collected further afield, since secondary outcrops are also located downstream from the site. This is why mobility strategies, related to the tools' reduction concept and type of occupation, may have led to a choice in the raw material management [9].

The comparison of the tools' cutting edge, instead, provides the most interesting element, if related to their functional implications. The cutting edge angles and bevel are indirect indicators of functionality, penetrative potential and movement of use [93,99,112]. A strong consistency is noticed when comparing the angles formed by the upper and lower surfaces on the cutting edge of Keilmesser and scrapers. Simple backed flakes bear more acute angles; however, the first two types result from resharpening and retouching phases that could have modified and increased the angle. If the data regarding angles are crossed to the Keilmesser's weight, empirical index of their reduction degree, a sharp increase of the angles in the lighter exhausted tools can be noticed (Fig 21b). Scrapers, whose angles are always very similar to the Keilmesser, were found to have only a slight increase of angles, which can be related to the fewer influence of retouching respect to their volume. On the contrary, flakes don't record any increase but the opposite: since flakes were not furtherly reduced, we can assert that small backed flakes were manufactured with a narrower active angle probably for their high functionality for precision and fine activities, as demonstrated within Discoid assemblages [75]. For this reason, we found that weight could represent a measure of reduction if applied to heavily reduced Keilmesser tools, but in the case of flake blanks it is more likely to represent the initial blank size.

The cutting edge bevels show a recurrent asymmetry between the lower and upper profile. The more concave lower bevel of Keilmesser essentially derives from the need to detach flakes

in order to create a cutting edge in a raw blank. It thus represents the concavity of the flakes' negative; apart from this feature, the lower bevel is mostly flat in Keilmesser as well as in other tools. Concerning the intersection between upper surface and cutting edge, Keilmesser record a convex or slightly convex profile that is halfway between backed flakes and backed scrapers. This data fits well with the presumed functionality of Keilmesser [9,67,113], since extremely convex and asymmetric bevel related to angles > 60° seem to be the prerogatives of scraping tools, while more flat, symmetrical or slightly asymmetrical bevel should be predisposed for cutting related to longitudinal motion [93,95,99,114,115]. However, it is evident also in this case that a quite internal variability is recognized in each tool type, and the morpho-techno-functional features of the Keilmesser overlap in most cases with those of scrapers and, in lesser extent, of simple flakes. In this regard, the available use-wear data on G-complex tools states that hafted or hand-held knives are documented among bifacials, Keilmesser, and unifacial Mousterian tools, within quite different typologies of artefacts and among not retouched pieces [67]. The "tool-type" is thus never associated to a single use or specific function, nor the opposite, confirming in a preliminar way what has been assumed by the techno-functional approach.

In this sense, typology turns out to be fluid, relying on isolated features. The major discriminating factor between Keilmesser and other backed artefacts is thus the bifacial retouch, combined with their volumetric concept which allows a greater potential for reuse [53]. However, with regards to this factor, Krukowski [60] already focused his attention on scrapers related to *prodniks* as unifacial transitional forms. Also Jöris [23] states that not all Keilmesser are of the core-tools type but they can also be manufactured on flakes, including a unifacial variant resembling backed scrapers and therefore transitional. Similarities in this regard between Keilmesser and scrapers in their shaping biograhies have also been noticed in other *Keilmesser-gruppen*//M.M.O. assemblages [19,62]. These transitional forms however, possess low variability that is related to the limits of the flake blank, implying medium or low resharpening potential and excluding any other multi-functional ambitions or their exploitation as cores. In fact, the resulting tool is a simple or scaled scraper with Keilmesser-like morphology but differing from it for other elements; a sort of simplistic version of Keilmesser, since many morpho-technical requirements (plano-convex bevel, technical back) are already fulfilled by the flake blank [62]. Scrapers bearing traces of tranchet blow technique (*pradnik* scrapers [66]) are also considered smaller and shorter-lived versions of Keilmesser, besides being sometimes associated to less experienced knappers or children manufacturing [116].

Conversely, classical Keilmesser are bifacially retouched blanks with a very high reuse potential. Classical Keilmesser are generally long-lived, mobile and multipurpose tools, according to working-step analyses of several KM-bearing assemblages ([9,19,52,66]. The final shape is, as already seen and confirmed by our dataset, a consequence of numerous subsequent stages [20,23,61]. These stages can lead to the tools' reshaping or "remolding", a recycling that involves a different function [117]. In Moravski Krumlov IV is also hypothesized that, in Szeletian, Streletskayan and perhaps also Micoquian assemblage, Keilmesser-like forms can constitute itself a phase in the manufacture of unfinished leaf points [109,118].

In any case, it is significant that a large part of Keilmesser from Sesselfelsgrotte had elaborate biographies that included different functions and objectives, from the "simple" tool to the matrix for the production of small-medium size flakes. Moreover, these phases could have been both subsequent and alternating (Fig 20). The G-complexes assemblages are also known for the presence of a microlithic tools component, mainly round scrapers or raclettes manufactured on small flakes (Fig 22a) [9,119]. These flakes were mainly obtained from surface shaping and exploitation of chert plaquettes, the same blank for most of Keilmesser; there is thus the possibility that the two reduction sequences are intertwined (Fig 22b). After all, in the

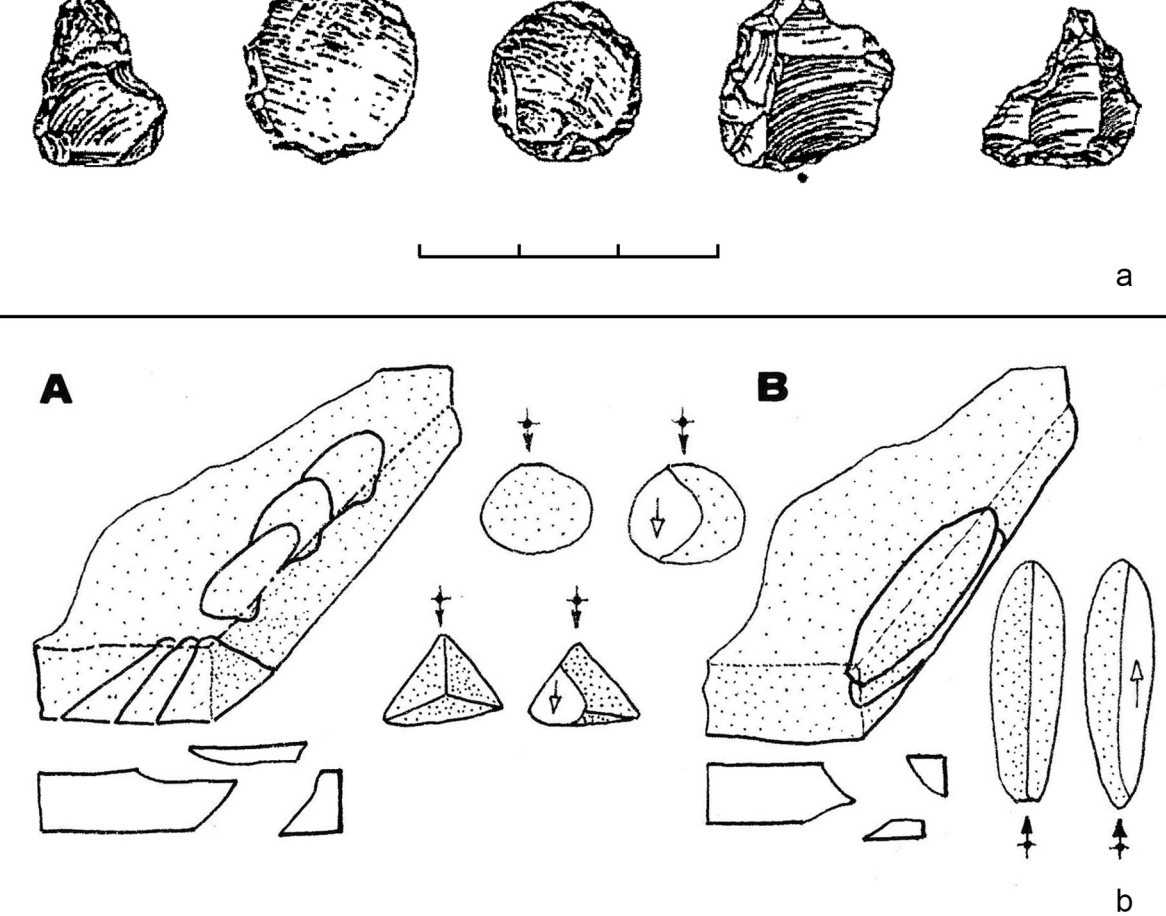

**Fig 22. Microlithic tools component recognized in the G-Complexes (a); flakes produced from the initialization and shaping of Keilmesser (b).** Modified from Richter 1997.

inventories where microlithic tools are numerous (A01, A06, A08, A09), the core-tools Keilmesser are very well attested; where they were less in number, core-tools are also absent or isolated. In the absence of refittings, techno-economic and functional data highlight the manufacturing of these tools out from the site and their very specific function specialized on working of vegetal materials [9,87]. This is why they go well with a possible obtaining from different stages of Keilmesser shaping, as the opening of chert plaquettes on lower surfaces or the mainteinance and reshaping stages, where the production of different flakes morphologies (round, ovalar or laminar) is attested [9].

The use of flakes obtained from bifacial tool shaping is attested in assemblages from all over the European Middle Paleolithic, from already mentioned Crimean Micoquian [110,120] to the Mousterian of Acheulean Tradition in South-Western France [48]. Keilmesser are the perfect tool adaptable to this bifacial multi-purpose conception, thanks to the easiness in which functional operational schemes can be modified through tool rotations and base-point or surface inversions in the volumetric exploitation [63]. The versatility of Keilmesser scheme allows the implement to be reorganized by extending its use-life and reshaping stages. Moreover, the double and alterned plano-convex concept allows both tool and core objectives to be met.

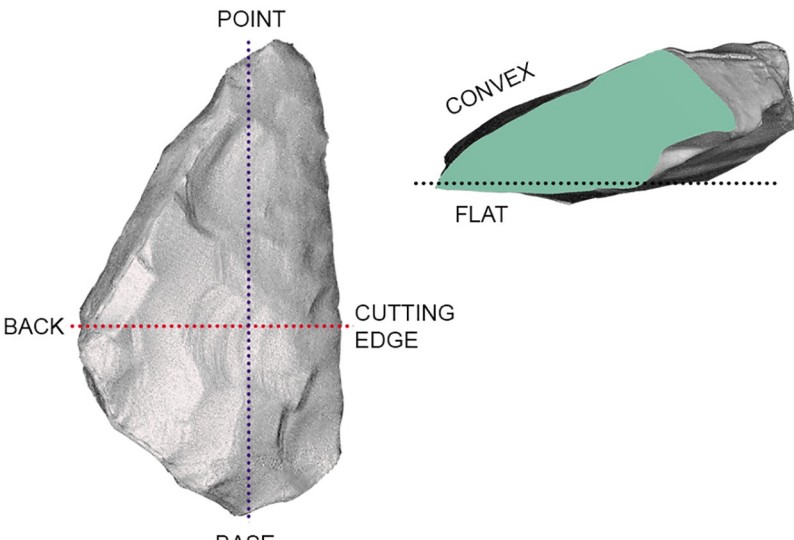

**Fig 23. The three main asymmetries of Keilmesser are showed: base/point, back/cutting edge and, viewed in cross-section, flat/convex surfaces.**

To sum up, the mental template behind the production of unifacial or bifacial backed tools is quite similar if not the same. The objective are cutting and/or scraping tools characterized by at least three asymmetries [20]: a longitudinal one (base vs tip, when present), a transversal one (back vs cutting edge), and the third on the cutting edge (flat vs convex) (Fig 23). This scheme may include most Keilmesser, many scrapers and some backed flakes. However, their maintenance and potential uses are different.

Therefore, we propose that at the base of this division there might be a sort of imitation process of bifacial backed knives towards the simple backed artefacts on flake. Boëda also pointed out that from a morphological point of view the edge of a plano-convex bifacial tool and of a modified knapping product are equivalent [53]. It has also been said that both unifacial and bifacial tools cover the same intentional sphere, despite having their own specific techno-functional features [19]. Moreover, 2D morphometrical Elliptic Fourier Analyses has shown that their reduction, in *keilmessergruppen*/M.M.O. assemblages, often follows quite similar trajectories [61]. This is also true for the bifacial tools with flat retouch; common types in late Middle Paleolithic assemblages, whose biographies have been related to simple unifacial artefacts [12,121,122].

If it has been said that unifacial backed knives are the simplified version of bifacial backed knives [62], we could assert that bifacial backed knives are the strategic, versatile versions of their unifacial counterparts. This "economic" interpretation enhances the tool as an exploitable volume, as a multi-purpose and multi-functional blank, as well as its possibility of angle maintenance [61], through intertwined and common working stages.

After all, the structure of the bifacial knife itself, independently from the context, has been related to products made in order to be rejuvenated [61,108]. In Bavarian *Keilmessergruppe*/M. M.O. contexts, the frequency of some types of bifacially shaped tools and leaf points has been interpreted as strictly dependent on the mobility strategies of the human groups that made them. Their frequency can be higher in palimpsests of recurrently visited, specialized leafpoint sites, where single pieces were used and abandoned after multiple import (like Weinberghohle, Zone 4 or Zeitlarn 1). They can be equally abundant, but with a larger variability of different forms, at residential sites with long times of activity [52]. The underlying idea is that the longer

a site is used in the course of the same occupation, and the more logistical the land-use pattern is, the more often long-life objects of otherwise mobile toolkits reach the very end of their use life at the same site. This scenario is based on the assumption that bifacial tools are used regularly and independent from the respective site function. Instead, the increased amount of discarded items is simply a consequence of staying at a camp site longer than the use-life of the single tools, irrespective of if they have been manufactured and used on-site or imported. Their effectiveness as long-life tools has been experimentally tested, with an estimate duration of at least several weeks [20]. The idea that parts of the Micoquian package, including Keilmesser, are more often discarded at base camps with long-term occupations therefore does not necessarily contradict to the well-proven notion that Keilmesser have a high strategic and versatile value and are part of a mobile tool-kit. Because it depends on the overall land use pattern, which is in general seen as being flexible, it does also fit to the high frequencies of Keilmesser observed at short-term and specialized sites like Lichtenberg or Zwolen.

Keilmesser, however, are not the only strategic tool-type in the European Late Middle Paleolithic, other tools imply similar duration and multi-purpose potential, like *limaces*. This tool involves an anticipatory behavior and, by integrating multiple functions, constitutes an original way of raw material circulation on long distances [123]. Limaces have sometimes been confused with Kartstein-type points, which are also characterized by upper invasive retouching, dorsal keel, plano-convex section and, moreover, flat detachments in the lower surface when the blank wasn't naturally provided. These two types are overlapping [124] and the similarities with Keilmesser-concept concerning the potential for resharpening is evident, besides the common presence in several sites [12]. Therefore, in contemporary contexts characterized by high and strategic mobility there were possibly different traditions in order to manufacture long-term tools potentially similar, but techno-functionally different: besides Keilmesser there were *limaces*, Quina scrapers, and possibly other bifacially-shaped tools [48,125].

We can therefore assert that backed unifacial and bifacial tools can derive from the same operational techno-functional concept. Their differentiation is instead an ecological consequence implying a specific requirement that is having a versatile volume that can be exploited for different purposes. This tendency is noticeable in several transitional bifacial forms (like bifacial scrapers or limaces), and finally in leaf points or leaf-knives, which in turn could have been a further consequence of reduction of bifacially shaped core-tools with or without a back [118,126]. However, this ecological variant doesn't deny the possible design of a specific mental template for Keilmesser that, like other potentially similar tool-types characteristics of other techno-complexes, may have assumed a strong cultural value, perhaps in a late phase of *Keilmessergruppen*/M.M.O. complex.

## 7.3 On the Keilmessergruppen/M.M.O. ecological and cultural implication

This study, though considering only G-Complex backed tools, provides data supporting the main interpretation by Richter, for which Mousterian and Micoquian technological traditions are actually deeply interrelated, according to his M.M.O. definition [9]. The *Keilmessergruppen*/M.M.O. could be framed within the Mousterian variability of Late Middle Paleolithic, since typological differences or the presence of bifacial technologies don't represent a sufficient cultural discriminating factor. Clear examples, in this sense, are Sesselfelsgrotte G-Complexes, along with the other main *Keilmessergruppe*/M.M.O. assemblages, where the technological and typological Mousterian substrate is always present or dominant, except in a few short-term and specialized contexts [15,47,52].

Within *Keilmessergruppen*/M.M.O., however, the relation between the so-called Mousterian "common background" and the "Micoquian option", is difficult to interpret. The sites length

occupation hypothesis isn't always adaptable, since Micoquian option appears to be related to frequent/high mobility strategies, thus emerging in contexts characterized by marked ecological constraints that could match with both long-term and short-term occupation. This dichotomy cannot therefore be explained only with a tool's reduction factor directly related to the type of occupation [127]. There is indeed a deeper element of ecological adaptation and predetermination of behavioral strategies, within which a different concept of technical object fits.

The Keilmesser, that in its variety of forms and concepts represents the main technical element of *Keilmessergruppen*/M.M.O., is indeed a precise ecological guide-fossil; it is the versatile, multi-purpose and strategical tool adapted to constrained situations. This fossil and the entire "option" emerge with nomadic seasonal occupations with cold or even harsh climates, characterized by the "mammoth steppe" environment, with the common presence also of reindeer, horse and rhinoceros [26,47]. These constrained landscapes represent a common denominator of *Keilmessergruppen*/M.M.O. human occupations, while those more properly Mousterian (without the "Micoquian" option) can be associated with broader periods and areas, often including milder environments.

This is true in a global Pan-European context, but is noticeable also within the same sites: in this sense, an example could be the interstratification of open-air sites in Königsaue. Königsaue-A and Königsaue-C, characterized by Levallois method next to Keilmesser and bifacial reduction, are interframed by Königsaue-B, where unifacial tools and core-reduction are exclusive. Here, raw materials management and other behavioral factors don't record marked differences, noticeable mainly in the frequency of bifacial and unifacial technologies [15]. Whether the stratigraphy represents the whole last glacial cycle [33] or climatic oscillations within early MIS 3 [47], the layer characterized by high rate of bifacial tools (K-A) record more harsh conditions, while layers where bifacial reduction is absent (K-B) or present but rare (K-C) takes place during a milder interstadial with forestal covering(Königsaue-A) or presence (Königsaue-C) [111].

If we take as example an interregional context, in the comparison between Polish, Moravian and Slovakian sites by Burdukiewicz [105], better ecological conditions characterized the regions south of the Carpathians, where the backed bifaces complexes are rarer even though the Middle Paleolithic sites are numerous. An ecological preference was therefore recognized, with Neanderthal groups going south mainly in colder periods. In Central-Eastern Europe the Ciemna stratigraphical sequence is particularly important, where the beginning of *Keilmessergruppe*/M.M.O. is related to the reoccupation of the region in still very cold periods, between MIS 4 and MIS 3; before this, only Mousterian and Taubachian (micro-Mousterian) are present [42]. South of Carpathians, the same broad behaviour is recorded in Kůlna sequence in Moravia [45].

*Keilmessergruppen*/M.M.O. are also present in more temperate areas, at the margins of their central European core-area, but in particularly cold stages; in south-western France, at Abri du Musée (ensemble V zone II), the Keilmesser with tranchet-blow technique appears together with cold and open environment fauna, like reindeer and mammoth [128]. In eastern France, Grotte de la Vérpiliére II is probably framed between the end of MIS 4 and the beginning of MIS 3, with a quite similar faunal assemblage [40]. The *Keilmessergruppen*/M.M.O. groups could have been pushed south and west in MIS 4, following large herbivores, when the plains of central Europe were covered by glaciers or permafrost and thus mostly uninhabitable (Fig 24). In these climatic-related population shifts, temporary refuge areas could have been south-western (from Burgundy to Dordogne or Brittany) and south-eastern (Pannonia basin) Europe, before MIS 3 reoccupation [26,106,113]. In this sense, the Central European *Keilmessergruppen*/M.M.O. settlement systems are thought to be developed in climatically challenged areas of central Europe, between the Alpine and Fenno-Scandinavian

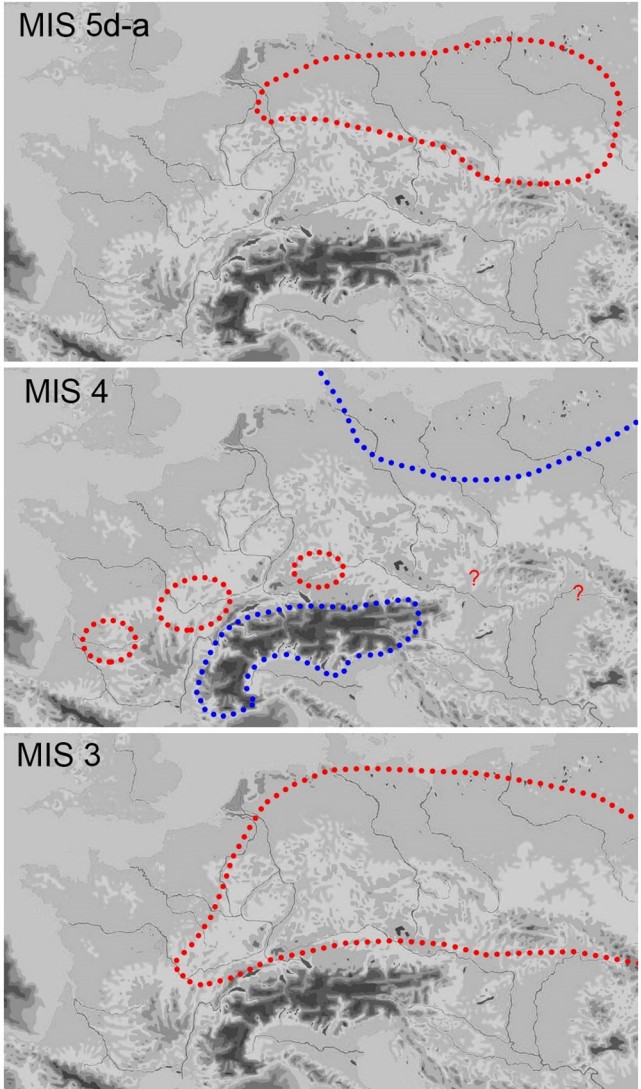

**Fig 24. Site distribution of *Keilmessergruppen* techno-complex in the early Weichselian, between MIS 5 and MIS 3.** The red zones were possibly populated, the blue zones were occupied by permanent glaciers.

glaciers, while classical Mousterian assemblages reflect different strategies of mobility and resources exploitation [129].

In any case, it seems that European *Keilmessergruppen*/M.M.O. tend to occupy environmentally demanding habitats, often related to stronger environmental constraints including (seasonal) permafrost, long winters and ecological reliance on migratory species. In these contexts, lithic raw materials are difficult to supply, and the Keilmesser could represent an excellent answer due to its long use-life, effectiveness and multi-functionality, good cutting ability, scraping, and butchering of carcasses. Its possible purpose as a core-tool can also imply the production of thin and sharp flakes or scrapers that can serve where the Keilmesser is useless (precision activities or wood-working). This seems to be particularly true for eastern contexts like Crimea, where, however, the climate in MIS 3 was characterized by faint fluctuation between humid/warm and dry/cool intervals, always in boreal forest-steppe landscape [130,131].

In a scenario, the late Middle Paleolithic of Central and Eastern Europe, where "Micoquian option" is generally known, different kinds of sites could be present. Within some open-air, short-term occupations, Keilmesser and related artefacts can represent the whole assemblage, with the absence of core-reduction technologies (i.e. in Lichtenberg [18], Zwolén [31] and Pietraszyn 49a [43]). In these cases, the strategic quality of this artefact as a mobile multi-functional tool is perfectly expressed. Conversely, after longer or more stable occupations, Keilmesser are accompanied by *débitage* products and were abandoned after more or less long biographies devoted to mobility [9,19,52]. The case of Sesselfelsgrotte G-Complex tells us that Keilmesser-bearing groups are not necessarily characterized by long-distance but frequent mobility patterns, especially in the "initial inventories" where exhausted tools or core-tools are numerous, becoming more logistic and organized during more stable "consecutive inventories" occupations, where resource availability is larger. Furthermore, according to the clear decreasing of strategic multi-purpose tools (core-tools Keilmesser) in relation to the whole backed bifacial tools (only 3 out of 18; Table 10) in the last G1 layer, it is possible that Keilmesser would have gradually increased their value as a cultural symbol and inheritance, rather than their ecological value, that was probably more important for their first conception and diffusion.

It is difficult to trace an evolutionary development of Keilmesser shape and function that could give us information in this sense. According to the chronological distinction by Jöris [23], after a debated phase A, the phase B is framed in MIS 4 and characterized by the typical resharpening tranchet blow technique, that could represent a strong cultural marker. Within the MIS 3 phase C this technique would considerably decrease, but this vision is partially revised and corrected [64]. Parallelisms with Sesselfelsgrotte are found in the long sequence of Ciemna cave, where the long-lived Keilmesser are typical of the more ancient layer (layer V), while in a second moment (layer III) these are less numerous and are found together with Levallois products [42]. In both cases, Keilmesser related to "ecological need" (here readable as bifacial core-reduction) would have been typical of a first, developing stage of *Keilmessergruppen*/M.M.O.

In order to advance diachronic interpretations, however, it is necessary to frame this techno-complex and its genesis according to a behavioral point of view. Valoch proposed, as origin of Micoquian assemblages, Eemian contexts like Tata where similar bifacial tools are present [132]. In the same way, the late Acheulean (Jungacheulean) would be at their base for Bosinski [12] and Kozlowski [14], while Jöris [26] asserts that they firstly emerge as a common fund toolkit in different and particular circumstances. The Hypothesis generally depends on whether the long or short chronologies are accepted; however, the early Weichselian (from MIS 5d but especially MIS 4) cooling episodes should be the key moments for the first definition and standardization of these techno-complexes. In these periods, demographic upheavals like displacement and decrease of human groups or mechanisms like bottlenecks may have contributed to form new technical objects or assemblages.

Western Europe, in MIS 4, is characterized by the widespread presence of Quina techno-complex, associated to high mobility patterns and exploitation of gregarious and migratory species adapted to harsh environments, like reindeer and bisons [6,133,134]. Lithic technology and economy are aimed at high mobile strategies, resembling Keilmesser-like behaviors; the technical investment is rarely employed to core reduction technologies, but to extend the life length of the flake blanks, here retouched into the typical Quina scrapers. This tool is also a possible matrix from which to detach flakes according to a ramified concept [135–137], thus a mobile blank with strategic value and a rather raw material source. Even if its reduction stages are different if compared to a bifacial tool, at the end of its functional life the Quina scraper

may bear bifacial retouching, bulb thinning and triangular shapes [71,135,138,139], being conceptually similar to bifacially shaped tools.

The "meeting" between the Quina technology and the *Keilmessergruppe*/M.M.O. package is well documented, especially in southern Germany, and dated to relatively ancient phases. The main contexts are Bockstein, firstly related to the passage between MIS 5 and MIS 4 [140] but then repositioned between MIS 4 and MIS 3 [69], and of course the most ancient layers of G-Complex in Sesselfelsgrotte, datable at the very beginning of MIS 3 re-occupation of the region after MIS 4. Similarities are noticed with the Rhodanian facies of Quina. Here, in addition to the high rate of retouched tools, tools resembling Keilmesser along with limaces are present and in contexts datable to the beginning of MIS 3 and at the limits of *Keilmessergruppen*/M.M.O. influence area [123]. Therefore, in this time span and in an area yet to be defined, a meeting and a partial coexistence of these two concepts have occurred. In the first occupational cycles of Sesselfelsgrotte G-Complex, the production of unifacial and bifacial blanks follow the same notions, that are the application of Quina concepts [9]. In the following phases, instead, there is the clear distinction between the core-reduction methods (mainly Levallois) for the production of flakes, and the manufacturing of simple or plano-covex bifacial tools on the other side. In Bockstein, the same behavior is documented, wherein small "bockstein-messer" are manufactured on Quina blanks [69]. This could represent the techno-functional integration of a tool-type, the backed knife, that in classical Quina assemblages is very differently shaped [70].

In a preliminary way, we can hypothesize that the first conception and production of the Keilmesser tool could have derived from ecological adaptations, which resulted in a series of actions functional to practical objectives. In this case, the aim was probably the manufacturing of a highly potential, multifunctional backed knife. Later in time, these tools could have become a cultural symbol, in a culturally and technologically fragmented Late Middle Paleolithic Europe, characterized by widespread regionalisms. Probably due to climatic and demographical shifts faced by late Neanderthal groups in the first Weichselian glacial maximum, a very diverse cultural mosaic is shaped and archaeologically perceptible [10]. In these phases the tool is more likely the result of the achievement of a conceptual scheme and, perhaps, equipped with aesthetic and symbolic value. The cultural and possibly ethnic value of these industries has recently been reinforced with data from the early-MIS 3 Neanderthals from Chagyrskaya in the Altai mountains: here, techno-typological affinities with Central- and Eastern-European Micoquian come together with genetic similarities with European specimens, while the same groups were unquestionably different from earlier Altai Neanderthals bearing Mousterian assemblages [141,142].

## 8. Conclusions

The techno-functional and morphological revision of backed implements from Sesselfelsgrotte G-complex allowed a broader understanding of the relationship between the different techno-typological entities. The main conclusions achieved through this work are the following:

• The Keilmesser tool-type is not always well defined but actually quite fluid. On a techno-functional base it includes several sub-types implying distinct functional operational schemes according to morphology, functional units, dimensions, etc. These schemes could both have direct functional or reductional values, since some of them (especially the pointed implements) could have represented the initial schemes, that eventually changed due to resharpening and retooling. This variability is confirmed by past use-wear analyses, perfectly showing the versatility of this concept. An approach that proved to be particularly useful is

the working-step analysis applied to techno-functional schemes, able to discern 3 types of Keilmesser distinct in their ecological biographies and potentialities. Among these, the core-tools recorded their multi-purpose concept as raw material matrix and asymmetrical backed tool.

- Enlarging the analyses to the other backed implements, including backed scrapers or simple flake-products, it is evident that a certain degree of overlapping affects these typological categories, firstly evident within 3D shape analyses. Even if some preferences are recorded among techno-functional schemes, no exclusive scheme for a single tool type exists. Moreover, comparisons between cutting edges' angles and active bevel analyzed in virtual section proved that Keilmesser's plano-convex section was probably manufactured in order to imitate the functionality of flake tools, besides the applicability of angle mainteinance techniques. Lastly, the backing type is related more to the initial blank than to the tool type. Past use-wear analyses provided consistent data in this sense, since typology is never associated automathically to a function, and backed knives are found among very different kinds of tools. The main discriminating factor between backed bifacial tools and unifacial tools is therefore the higher potential and use-life length of the first tools.

- For these reasons Keilmesser are the ideal strategic backed implements, mainly hand-held and useful for cutting, scraping and occasionally to produce smaller flakes from their flat surface. These tools and the so-called "Micoquian option" they represent are deeply interrelated to the Mousterian techno-tipological substrate and emerge in constrained ecological conditions related to patterns of frequent mobility. These conditions favored the production of highly strategic, long-life blanks and the ramification of knapping reduction sequences. In the absence of large-scale reliable seasonal data on the occupations, we can assert that possibly major climatic shifts are at the base of their emergence, framed around the end of MIS 5 and the beginning of MIS 3. Broad demographic dynamics, following the expansion and contraction of ice covering and related herds of herbivores, resulted in a widespread techno-cultural regionalization from which also *keilmessergruppen*/M.M.O. emerged. This is why Keilmesser, after their initial ecological and strategic signficance, could have then represented a tool equipped with cultural-related meaning. Sesselfelsgrotte G-complex seem to weakly document this kind of behavior. However, a broad comparison between early and late *Keilmessergruppen*/M.M.O. contexts is needed in order to verify this claim. Moreover, extensive chronometrical analyses will clarify the reliability of the long or short chronologies applied to *Keilmessergruppen*/M.M.O.

Future perspectives, in this sense, may also come from a more intense and targeted experimental protocol aimed at measuring the efficiency of Keilmesser in relation to other similar backed tools typical of other techno-complexes, in order to further quantify the ecological efficiency or the presumed cutural valence. The backing concept, here embedded in the raw blank use strategies, is particularly important in this chrono-cultural phase, when intentional backing by means of retouch appears in relation to specialized and precision tools [75]. In this sense, the bifacial shaping of back's angles and conjunctions seems to be a peculiar technical behavior, not documented elsewhere, that deserves an indepth investigation. The current research approach evaluates the technical objects in relation to their usefulness, environmental adaptation and also cognitive factors such as lateralization and planning depth. When this relationship isn't clear or is lacking, traditions and fashion may have played a more important role, especially in Late Middle Paleolithic Europe, fragmented into a technological, cultural and possibly biological mosaic.

## Supporting information

**S1 Data.**
(XLSX)

## Acknowledgments

The authors thank Prof. Marc Stamminger and Dr. Frank Bauer from the Informatik Department at FAU Erlangen-Nürnberg for the kind concession of David 3D scanner; the authors are also deeply grateful to Dr. Andreas Maier and Dr. Andreas Pastoors for productive discussion, to Dr. Atsushi Noguchi and an anonymous reviewer for positive feedback and helpful comments that considerably ameliorated the manuscript, and to Dr. Alex Pearse for the English proofreading.

## Author Contributions

**Conceptualization:** Davide Delpiano, Thorsten Uthmeier.

**Data curation:** Davide Delpiano.

**Formal analysis:** Davide Delpiano.

**Funding acquisition:** Thorsten Uthmeier.

**Investigation:** Davide Delpiano.

**Methodology:** Davide Delpiano.

**Resources:** Thorsten Uthmeier.

**Supervision:** Thorsten Uthmeier.

**Writing – original draft:** Davide Delpiano.

**Writing – review & editing:** Davide Delpiano, Thorsten Uthmeier.

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
