## [Decision Letter · Decision Letter 0]

19 Jun 2020

PONE-D-20-12918

Techno-functional and 3D shape analysis applied for investigating the variability of backed tools in the Late Middle Paleolithic of Central Europe.

PLOS ONE

Dear Dr. Delpiano,

Thank you for submitting your manuscript to PLOS ONE. After careful consideration, we feel that it has merit but does not fully meet PLOS ONE’s publication criteria as it currently stands. Therefore, we invite you to submit a revised version of the manuscript that addresses the points raised during the review process.

Reviewers greatly appreciated your work and they suggest only minor revisions. I would suggest to follow the comment of Rev. 1, who suggest to shorten some parts of the manuscript. Once the revisions will be submitted along with your rebuttal letter, I think i will be able to take a decision rapidly.

We look forward to receiving your revised manuscript.

Kind regards,

Andrea Zerboni, Ph.D.

Academic Editor

PLOS ONE

Journal Requirements:

2.  In your manuscript, please provide additional information regarding the specimens used in your study. Ensure that you have reported specimen numbers and complete repository information, including museum name and geographic location.

For more information on PLOS ONE's requirements for paleontology and archaeology research, see https://journals.plos.org/plosone/s/submission-guidelines#loc-paleontology-and-archaeology-research.

Reviewers' comments:

Reviewer's Responses to Questions

**Comments to the Author**

1. Is the manuscript technically sound, and do the data support the conclusions?

Reviewer #1: Yes

Reviewer #2: Yes

2. Has the statistical analysis been performed appropriately and rigorously? 

Reviewer #1: Yes

Reviewer #2: Yes

3. Have the authors made all data underlying the findings in their manuscript fully available?

Reviewer #1: No

Reviewer #2: Yes

4. Is the manuscript presented in an intelligible fashion and written in standard English?

Reviewer #1: Yes

Reviewer #2: Yes

5. Review Comments to the Author

Reviewer #1: Manuscript Number: PONE-D-20-12918

Title: Techno-functional and 3D shape analysis applied for investigating the variability of backed tools in the Late Middle Paleolithic of Central Europe

Authors: Delpiano, Davide

Uthmeier, Thorsten

The study “Techno-functional and 3D shape analysis applied for investigating the variability of backed tools in the Late Middle Paleolithic of Central Europe” by Davide Delpiano and Thorsten Uthmeier analyzes backed tools from the late Middle Paleolithic “G-Layers” of the southern German site “Sesselfelsgrotte”. They use a combined approach of techno-functional analyses and 3D geometric morphometrics. Their main finding is the relatedness of backed unifacial tools (“scrapers”) and bifacial tools (Keilmesser). As scrapers are mostly a characteristic of Mousterian assemblages, and bifacial tools of the central European Micoquian or Keilmessergruppen, the authors provide here a strong argument for the relatedness of these technocomplexes. Further, they interpret the emergence of Keilmesser as an ecological answer to the climatic shifts at the beginning of the last glacial cycle in Central Europe.

The paper represents a profound lithic analysis, combining several methods and lines of evidence to deal with the main research aims. The statistics are correct; the limit of the p-value is explained, and the study does not suffer from “over-testing”. The analysis is based on a large dataset from a rich site with various occupations. That represents a reasonable baseline for the interpretations given in the paper. I suggest the publication of the article after the authors have revised some minor issues.

General comments:

(1) I would strongly suggest capitalizing the noun “Keilmesser” as it is a specific name. Further, I advise the authors to remove the plural “s” from Keilmesser. In German, singular and plural of Keilmesser is the same (das Keilmesser, die Keilmesser) and I think this should be done in the same way in English (as it is a German term in an English text).

(2) When reading the paper, I can understand that every argument has its specific reason and makes sense. However, I think that parts of the paper are too long and can be shortened. As PLOS ONE accepts articles of any length, I leave the decision to the editor and the authors. Therefore, the following points are just suggestions. I could imagine merging the Introduction with chapter 2 and parts of chapter three. Other aspects of chapter 3, like the stratigraphy of Sesselfelsgrotte could be moved to the Materials section. Also, I do not think that it is necessary for the main plot of the paper to explain Richter’s models in detail here. Maybe there are some aspects of the site description that could also be moved to the SI? Furthermore, I think that chapter 7.4 is maybe a little bit too much. I don´t think that the issue of the emergence of the Keilmessergruppen could be solved with the data from the analysis presented here.

(3) Some of the plots and diagrams are done in R. But I could not find any citation or mention of the use of this software in the Methods section. I ask the authors to please add which steps (data management etc.) they did in R and which packages they used.

(4) I am missing a brief discussion within the 3DGM section about which aspects of shape change PC1 and PC2 account for. If I am correct, it is visible in the color coding of the displayed shapes in Fig. 13a. I think the analysis would benefit from adding some ideas here. It would be interesting, for example, to see where the different TFS plot in the PCA of shape space.

Specific Comments

Abstract:

a. “In the Late Middle Paleolithic of central Europe” – I think the authors can capitalize “Central”. They should look through the text as both versions occur.

Introduction:

a. “Königsaue was originally dated to MIS 5c [33]”: this is not correct. Königsaue was originally dated to Oderadde which correlates with MIS 5a. See here also the discussion by Jöris (citation [26]), or Mania 2002.

b. “Both the “long chronology” and the “short chronolgy” agree that a large number of sites from secure stratigraphical contexts and with reliable absolute dates fall into a period between the end of MIS 4 and the first part of MIS 3. Among these sites are Salzgitter-Lebenstedt [19,37], Lichtenberg [18], Sesselfelsgrotte [38,39], Verpillière I and II [40], Ciemna and Oblazowa [41,42], Pietraszyn 49 [43] and Kůlna [44]. (Fig. 1)”: I advise the authors to be cautious here with Lichtenberg and Salzgitter-Lebenstedt. The Lichtenberg artifacts come from a cryoturabted context which yielded TL-dates between late MIS 5 and MIS 3 (see Veil et al. 1994 for descriptions). The MIS 3 date is just a calculated model age, which results in a rather insecure age attribution for the find layer. The same applies for Salzgitter: although Pastoors 2001 (and 2009) provides good and reasonable evidence for a MIS 3 age of the site, the radiocarbon ages are close to the limit of the method and the stratigraphic context is far from simple (e.g., 19 geological horizons with finds). Alternative interpretations with a late MIS 5 age are also reasonable and should be considered (see here also Jöris [26]). Securely dated assemblages that are not mentioned here and that should be included (also in the map) are: Pouch, Saxony-Anhalt (MIS 3, OSL: Weiss 2015 Quartär and Weiss 2018 [60]), and Wroclaw Hallera Avenue, Poland (MIS 5 and MIS 3, OSL: Wisniewski et al. 2013, Occupation dynamics north of the Carpathians and Sudetes during the Weichselian (MIS5d-3): The Lower Silesia (SW Poland) case study).

c. “relationship between Mousterian and /Keilmessergruppen/M.M.O.”: please remove the “/” before Keilmessergruppen

Chapter 2: Relevance of backed tools…

a. “As suggested by the direct translation from the original german noun”: please capitalize “German”

b. “as well as in several polish assemblages being part of the so-called “Bockstein group”: “Polish”

c. “In the most recent Keilmessergruppen/M.M.O. sites, Levallois débitage is however usually dominant, in both its recurrent centripetal and recurrent parallel/unipolar variants, as attested in Sesselfelsgrotte, Königsaue, Salzgitter-Lebenstedt and Lichtenberg and other sites [9,18,19,33,35,50]” : Lichtenberg has only one prepared core. Based on facetted flake platforms, Veil et al. only suggest the presence of Levallois. However, they also note that the facetted platforms can equally result from bifacial reduction. Additionally, the authors state the same in chapter 7.3: “Within some open-air, short-term occupations, keilmessers and related artefacts can represent the whole assemblage, with the absence of core-reduction technologies (i.e. in Lichtenberg [18] or Zwolén [31])”

Chapter 3.1 Lithic assemblage data

a. Some references in this chapter (“Both the archaeology and the environmental studies were published in a monograph series (Weißmüller 1995; Richter 1997; Böhner 2008; Dirian 2004; Freund 1998; Freund and Richter 2017; 2018) or in articles (Richter 2002; Richter et al. 2000; Rathgeber 2006)”) are not formatted correctly

Chapter 3.2 Lithic assemblage data

a. Fig. 3: initial- and konsekutivinventar just with one “a” (not “Inventaar”)

Chapter 5.1 TFU analysis

a. “relevance of the techno-functional needs and the various ecological constraints that thir makers and users were confronted with”: “their”

Chapter 6.1.2 Metrical data

a. “Among the KMs, the overall smaller width mean and standard deviationcomes to the eye”: “deviation comes”

Chapter 6.2.1: Techno-functional schemes

a. “If bifacially worked, artifacts of the TFS 1 recall keilmesser types like Bocksteinmesser or Balver Messer.”: Readers not familiar with Keilmesser might ask what the last two terms are, as they are not explained in the text. I would suggest to either give a citation (e.g., Bosinski 1967), remove it, or explain it briefly.

b. “Techno-functional scheme 2 (TFS 2)”: “2a” is not labelled in the figure (Fig. 8), just “2” and “2b”. Further, already 2a, as illustrated in the figure, resembles a Lichtenberg Keilmesser with its oval outline, the elongated bow and the sharp distal tip. Although having a rounded tip, the specimen displayed in Fig. 8: 2b, is rather untypical for a Lichtenberg Keilmesser. See the definitions given in Veil et al. 1994 (“Idealtyp” and additional forms), as well as the drawings therein.

Chapter 6.3 3D morphological data

a. Table 9 & Table 10: What exactly is the number given in the Variability column? How is this number calculated and what does the number express about variability? E.g., what number relates to a low and what number to a high variability? Or is the interpretation of high and low variability just the numbers relative to each other? I would suggest adding one or two sentences to explain these values.

Chapter 7.1 One, no one and one hundred….

a. “3. One group with a curved or convex cutting edge (schemes 6/7 and 7), comprising 5 tools or 8.6%.”: I just like to add that group 1 also has a convex cutting edge (TFS 1 and 2)… maybe it would be better to say “heavily curved cutting edge”

b. “In this sense, keilmesser of scheme 2 could be modified by resharpening to scheme 2b in cases when the tip reshaped into a rounded outline (after it became blunt).” + further sentences: In my opinion this interpretation is a bit vague and may need further techno-functional investigations or experimental replication (I know this is beyond the scope of the paper). Just an example: Let`s say we start with something like 2a. If one thins the distal volume of the piece during resharpening, using the bow and the distal back as striking platform (like in figure 20 upper right specimen), the removal of volume from these edges -and moving towards the proximal parts of the artifact during the process- may result in a piece with a more elongated and longer point + shorter back (as in the Königsaue type). In other words, this strategy of resharpening can lead to results opposed to what the authors suggest (removal or rounding of the point). Generally, both variants may have existed. Maybe the authors should make it clearer that this paragraph is more hypothetical and that other reduction strategies may exist as well.

c. “Core tools have been also recognized in Neumark-Nord 2 and Konigsaue [109]”: Königsaue

Chapter 7.2 Differentiation and imitation:…

a. “weight, empirical index of their reduction degree”: I think that weight is not a good and precise measure of reduction. How deal the authors with pieces manufactured from small blanks?

b. “Even Jöris [23] states that not all keilmessers are of the core-tools type but they”: I strongly suggest removing the word “Even” here or replace it by “also” or an equivalent. “Even” expresses a specific opinion about or an evaluation of Jöris here and, despite the long chronology vs. short chronology debate, hypotheses of Jöris are not analyzed/discussed in this paper.

Chapter 7.3 On the Keilmessergruppen/MMO…

a. “Within some open-air, short-term occupations, keilmessers and related artefacts can represent the whole assemblage, with the absence of core-reduction technologies (i.e. in Lichtenberg [18] or Zwolén [31])”: Pietraszyn 49a is a good example here as well.

b. “According to the chronological distinction by Joris [23], after a debated phase A,”, and the paragraph below: “Jöris”

Reviewer #2: While microscopic approach especially to use-wear makes great achievement on functional study of stone tools, macroscopic approach on lithic morphology with techno- and mechano-functional aspects is rare. In this regard, this is recognized as an epoch making paper in the field. The concept and research design is excellent. Analyses conducted in the paper and discussion are sufficient. Introducing TFS in comparison with conventional classification (typology) is effective to explain and interpret technological system. Fig.12 is so nice to represent differences between Keilmessers and scrapers. These series of analyses achieve new perspective which can overcome dichotomy of different paradigm: morpho-typology for cultural history and functional evaluation with quantified approach.

Nevertheless, if the original data related to 3D morphology (including angle of edges) is available, it would be more sufficient with more transparency and reproducibility. Additional description or schematic illustration for procedure and controls in 3D morphology would help understanding of readers.

Despite of this additional request, this paper is significant on establishing a novel methodology of lithic morphology in general, not in the specific field of European Middle Palaeolithic.

I am fully recommended to publish this paper.

6. PLOS authors have the option to publish the peer review history of their article (what does this mean?). If published, this will include your full peer review and any attached files.

Reviewer #1: No

Reviewer #2: Yes: Atsushi Noguchi

---

## [Author Response · Author response to Decision Letter 0]

29 Jun 2020

All the replies to the reviewers’ comments are written below.

Reviewers' comments:

Reviewer's Responses to Questions

Comments to the Author

1. Is the manuscript technically sound, and do the data support the conclusions?

Reviewer #1: Yes

Reviewer #2: Yes

2. Has the statistical analysis been performed appropriately and rigorously?

Reviewer #1: Yes

Reviewer #2: Yes

3. Have the authors made all data underlying the findings in their manuscript fully available?

Reviewer #1: No

Reviewer #2: Yes

We can make fully available (as supplementary information) a simplified database of all the analysed records, including all the raw information used to develop the whole statistics. 

4. Is the manuscript presented in an intelligible fashion and written in standard English?

Reviewer #1: Yes

Reviewer #2: Yes

5. Review Comments to the Author

Reviewer #1: Manuscript Number: PONE-D-20-12918

Title: Techno-functional and 3D shape analysis applied for investigating the variability of backed tools in the Late Middle Paleolithic of Central Europe

Authors: Delpiano, Davide

Uthmeier, Thorsten

The study “Techno-functional and 3D shape analysis applied for investigating the variability of backed tools in the Late Middle Paleolithic of Central Europe” by Davide Delpiano and Thorsten Uthmeier analyzes backed tools from the late Middle Paleolithic “G-Layers” of the southern German site “Sesselfelsgrotte”. They use a combined approach of techno-functional analyses and 3D geometric morphometrics. Their main finding is the relatedness of backed unifacial tools (“scrapers”) and bifacial tools (Keilmesser). As scrapers are mostly a characteristic of Mousterian assemblages, and bifacial tools of the central European Micoquian or Keilmessergruppen, the authors provide here a strong argument for the relatedness of these technocomplexes. Further, they interpret the emergence of Keilmesser as an ecological answer to the climatic shifts at the beginning of the last glacial cycle in Central Europe.

The paper represents a profound lithic analysis, combining several methods and lines of evidence to deal with the main research aims. The statistics are correct; the limit of the p-value is explained, and the study does not suffer from “over-testing”. The analysis is based on a large dataset from a rich site with various occupations. That represents a reasonable baseline for the interpretations given in the paper. I suggest the publication of the article after the authors have revised some minor issues.

General comments:

(1) I would strongly suggest capitalizing the noun “Keilmesser” as it is a specific name. Further, I advise the authors to remove the plural “s” from Keilmesser. In German, singular and plural of Keilmesser is the same (das Keilmesser, die Keilmesser) and I think this should be done in the same way in English (as it is a German term in an English text).

Fixed as suggested

(2) When reading the paper, I can understand that every argument has its specific reason and makes sense. However, I think that parts of the paper are too long and can be shortened. As PLOS ONE accepts articles of any length, I leave the decision to the editor and the authors. Therefore, the following points are just suggestions. I could imagine merging the Introduction with chapter 2 and parts of chapter three. Other aspects of chapter 3, like the stratigraphy of Sesselfelsgrotte could be moved to the Materials section. Also, I do not think that it is necessary for the main plot of the paper to explain Richter’s models in detail here. Maybe there are some aspects of the site description that could also be moved to the SI? Furthermore, I think that chapter 7.4 is maybe a little bit too much. I don´t think that the issue of the emergence of the Keilmessergruppen could be solved with the data from the analysis presented here.

We agree that chapter 7.4 could seem a bit pretentious: since the issue of the emergence of Keilmessergruppen is widely beyond the scope of the paper, we decided to remove the subdivision as a chapter itself but we maintain part of the text as a point of discussion (in face of our analysis and recent bio-ecological data) merging it with the previous chapter (7.3).

About the other mentioned chapters, we slightly shorten some small parts but we didn’t considerably changed the manuscript since, as the reviewer says, every part has its reason for the economy of the work. Therefore, we are not sure about moving the “stratigraphy” section: we believe that it has to be presented before the in-depth presentation of the lithic assemblage and not after, together with the “Materials” section. On the other hand, we agree that the explanation of Richter’s model isn’t necessary here, so we removed it.

(3) Some of the plots and diagrams are done in R. But I could not find any citation or mention of the use of this software in the Methods section. I ask the authors to please add which steps (data management etc.) they did in R and which packages they used.

Added in the “Methods” section as suggested.

(4) I am missing a brief discussion within the 3DGM section about which aspects of shape change PC1 and PC2 account for. If I am correct, it is visible in the color coding of the displayed shapes in Fig. 13a. I think the analysis would benefit from adding some ideas here. It would be interesting, for example, to see where the different TFS plot in the PCA of shape space.

It is correct, in Fig.13a the yellow-reddish parts are the most changing shapes according to the 2 principal components. We added a brief sentence about this in the 6.3 chapter. 

About the TFS PCA plot, we decided not to add it because it appears rather messy and less clear if compared to the groups distance showed by cluster (Fig. 15a). The techno-functional schemes are too numerous, their ellipsoids overlap, and the most interesting feature is the centroid, better exposed in the cluster plot. However, if you are interested to see where the TFS plot in the PCA, we modified the figure and added the localization of the TFS centroids in the plot.

Specific Comments

Abstract:

a. “In the Late Middle Paleolithic of central Europe” – I think the authors can capitalize “Central”. They should look through the text as both versions occur.

Fixed

Introduction:

a. “Königsaue was originally dated to MIS 5c [33]”: this is not correct. Königsaue was originally dated to Oderadde which correlates with MIS 5a. See here also the discussion by Jöris (citation [26]), or Mania 2002.

Corrected as suggested

b. “Both the “long chronology” and the “short chronolgy” agree that a large number of sites from secure stratigraphical contexts and with reliable absolute dates fall into a period between the end of MIS 4 and the first part of MIS 3. Among these sites are Salzgitter-Lebenstedt [19,37], Lichtenberg [18], Sesselfelsgrotte [38,39], Verpillière I and II [40], Ciemna and Oblazowa [41,42], Pietraszyn 49 [43] and Kůlna [44]. (Fig. 1)”: I advise the authors to be cautious here with Lichtenberg and Salzgitter-Lebenstedt. The Lichtenberg artifacts come from a cryoturabted context which yielded TL-dates between late MIS 5 and MIS 3 (see Veil et al. 1994 for descriptions). The MIS 3 date is just a calculated model age, which results in a rather insecure age attribution for the find layer. The same applies for Salzgitter: although Pastoors 2001 (and 2009) provides good and reasonable evidence for a MIS 3 age of the site, the radiocarbon ages are close to the limit of the method and the stratigraphic context is far from simple (e.g., 19 geological horizons with finds). Alternative interpretations with a late MIS 5 age are also reasonable and should be considered (see here also Jöris [26]). Securely dated assemblages that are not mentioned here and that should be included (also in the map) are: Pouch, Saxony-Anhalt (MIS 3, OSL: Weiss 2015 Quartär and Weiss 2018 [60]), and Wroclaw Hallera Avenue, Poland (MIS 5 and MIS 3, OSL: Wisniewski et al. 2013, Occupation dynamics north of the Carpathians and Sudetes during the Weichselian (MIS5d-3): The Lower Silesia (SW Poland) case study).

We thank the reviewer for these suggestions: the references to Pouch and Wroclaw sites have been added in the text and the map. In addition, Salzgitter-Lebenstedt and Lichtenberg have been removed from the “securely MIS 3 sites” and a brief clarification has been added as suggested. 

c. “relationship between Mousterian and /Keilmessergruppen/M.M.O.”: please remove the “/” before Keilmessergruppen

Fixed

Chapter 2: Relevance of backed tools…

a. “As suggested by the direct translation from the original german noun”: please capitalize “German”

Fixed

b. “as well as in several polish assemblages being part of the so-called “Bockstein group”: “Polish”

Fixed

c. “In the most recent Keilmessergruppen/M.M.O. sites, Levallois débitage is however usually dominant, in both its recurrent centripetal and recurrent parallel/unipolar variants, as attested in Sesselfelsgrotte, Königsaue, Salzgitter-Lebenstedt and Lichtenberg and other sites [9,18,19,33,35,50]” : Lichtenberg has only one prepared core. Based on facetted flake platforms, Veil et al. only suggest the presence of Levallois. However, they also note that the facetted platforms can equally result from bifacial reduction. Additionally, the authors state the same in chapter 7.3: “Within some open-air, short-term occupations, keilmessers and related artefacts can represent the whole assemblage, with the absence of core-reduction technologies (i.e. in Lichtenberg [18] or Zwolén [31])”

Thank you for this specification, we missed the first reference to Lichtenberg, that has been now deleted.

Chapter 3.1 Lithic assemblage data

a. Some references in this chapter (“Both the archaeology and the environmental studies were published in a monograph series (Weißmüller 1995; Richter 1997; Böhner 2008; Dirian 2004; Freund 1998; Freund and Richter 2017; 2018) or in articles (Richter 2002; Richter et al. 2000; Rathgeber 2006)”) are not formatted correctly

This part has been removed since the references are correctly cited later in the text.

Chapter 3.2 Lithic assemblage data

a. Fig. 3: initial- and konsekutivinventar just with one “a” (not “Inventaar”)

Corrected

Chapter 5.1 TFU analysis

a. “relevance of the techno-functional needs and the various ecological constraints that thir makers and users were confronted with”: “their”

Fixed

Chapter 6.1.2 Metrical data

a. “Among the KMs, the overall smaller width mean and standard deviationcomes to the eye”: “deviation comes”

Fixed

Chapter 6.2.1: Techno-functional schemes

a. “If bifacially worked, artifacts of the TFS 1 recall keilmesser types like Bocksteinmesser or Balver Messer.”: Readers not familiar with Keilmesser might ask what the last two terms are, as they are not explained in the text. I would suggest to either give a citation (e.g., Bosinski 1967), remove it, or explain it briefly.

A citation was added and we leaved only the reference to Balver Messer since we moved the Bocksteinmesser reference to TFS type 6b, as showed by Bosinski (1967).

b. “Techno-functional scheme 2 (TFS 2)”: “2a” is not labelled in the figure (Fig. 8), just “2” and “2b”. Further, already 2a, as illustrated in the figure, resembles a Lichtenberg Keilmesser with its oval outline, the elongated bow and the sharp distal tip. Although having a rounded tip, the specimen displayed in Fig. 8: 2b, is rather untypical for a Lichtenberg Keilmesser. See the definitions given in Veil et al. 1994 (“Idealtyp” and additional forms), as well as the drawings therein.

Figures 8, 15 and 16 (not labelling 6a or 2a) have been corrected. Then, we think that both TFS 2a and 2b could fit into the definition of Lichtenberg Keilmesser (as we see from Bosinski 1967), and we modified the sentence accordingly to this.

Chapter 6.3 3D morphological data

a. Table 9 & Table 10: What exactly is the number given in the Variability column? How is this number calculated and what does the number express about variability? E.g., what number relates to a low and what number to a high variability? Or is the interpretation of high and low variability just the numbers relative to each other? I would suggest adding one or two sentences to explain these values.

The variability is calculated automatically by the software AGMT3-D: higher the number, higher the variability. As explained in Herzlinger & Grosman (2018) “the shape variability is measured as the mean multidimensional Euclidean distance of the items in the group from the group’s centroid (i.e. mean shape). The centroid size of each artifact is measured as the square root of the sum of squared Euclidean distances of all landmarks to the item’s centroid.” A brief description and a reference have been added.

Chapter 7.1 One, no one and one hundred….

a. “3. One group with a curved or convex cutting edge (schemes 6/7 and 7), comprising 5 tools or 8.6%.”: I just like to add that group 1 also has a convex cutting edge (TFS 1 and 2)… maybe it would be better to say “heavily curved cutting edge”

Yes, and the lack of a pointed end. We consequently changed the sentence.

b. “In this sense, keilmesser of scheme 2 could be modified by resharpening to scheme 2b in cases when the tip reshaped into a rounded outline (after it became blunt).” + further sentences: In my opinion this interpretation is a bit vague and may need further techno-functional investigations or experimental replication (I know this is beyond the scope of the paper). Just an example: Let`s say we start with something like 2a. If one thins the distal volume of the piece during resharpening, using the bow and the distal back as striking platform (like in figure 20 upper right specimen), the removal of volume from these edges -and moving towards the proximal parts of the artifact during the process- may result in a piece with a more elongated and longer point + shorter back (as in the Königsaue type). In other words, this strategy of resharpening can lead to results opposed to what the authors suggest (removal or rounding of the point). Generally, both variants may have existed. Maybe the authors should make it clearer that this paragraph is more hypothetical and that other reduction strategies may exist as well.

Of course, we are not talking about rules but possibilities: we added some clarifications about the hypothetical value of the paragraph. Several types of data (dimensions, angle of cutting edge, working-step on some pieces) would put TFS 2a as the “starting shape” of many keilmesser; but there are cases also with initial TFS 1 or 6, as well, since there are different initial possibilities (according to the shape of the raw plaquette/pebble) and reduction variants. In Fig. 16 we showed the most probable and/or common, but not a rule.

c. “Core tools have been also recognized in Neumark-Nord 2 and Konigsaue [109]”: Königsaue

Corrected

Chapter 7.2 Differentiation and imitation:…

a. “weight, empirical index of their reduction degree”: I think that weight is not a good and precise measure of reduction. How deal the authors with pieces manufactured from small blanks?

We strongly agree that this cannot be a rule since it depends on many variables. However, this can be a general rule if applied to Keilmesser in Sesselfelsgrotte since we are dealing with heavily reduced bifacial tools mainly manufactured on chert or quartzite plaquettes, available in comparable dimensions: in fact, within each Keilmesser group there is an increase of the active angles with the decrease in the blank size and weight, thus associated to the reshaping of the tool. The only exception could be represented by the tools made on Lydite and Radiolarite pebbles, that are considered apart since the different initial size and shape. But even considering these raw materials alone, the lighter and smaller tools coincide with the exhausted and more reduces blanks.

Given these premises, we found that scrapers do not follow this “rule”: since in the case of scrapers we are mainly dealing with flake-blanks with low degree of reduction, weight and size are likely to represent the initial blank rather than a reduction degree. We added a sentence at the end of the paragraph in order to clarify this assumption.

b. “Even Jöris [23] states that not all keilmessers are of the core-tools type but they”: I strongly suggest removing the word “Even” here or replace it by “also” or an equivalent. “Even” expresses a specific opinion about or an evaluation of Jöris here and, despite the long chronology vs. short chronology debate, hypotheses of Jöris are not analyzed/discussed in this paper.

If so, “also” is the correct word here.

Chapter 7.3 On the Keilmessergruppen/MMO…

a. “Within some open-air, short-term occupations, keilmessers and related artefacts can represent the whole assemblage, with the absence of core-reduction technologies (i.e. in Lichtenberg [18] or Zwolén [31])”: Pietraszyn 49a is a good example here as well.

Added

b. “According to the chronological distinction by Joris [23], after a debated phase A,”, and the paragraph below: “Jöris”

Fixed

Reviewer #2: While microscopic approach especially to use-wear makes great achievement on functional study of stone tools, macroscopic approach on lithic morphology with techno- and mechano-functional aspects is rare. In this regard, this is recognized as an epoch making paper in the field. The concept and research design is excellent. Analyses conducted in the paper and discussion are sufficient. Introducing TFS in comparison with conventional classification (typology) is effective to explain and interpret technological system. Fig.12 is so nice to represent differences between Keilmessers and scrapers. These series of analyses achieve new perspective which can overcome dichotomy of different paradigm: morpho-typology for cultural history and functional evaluation with quantified approach.

Nevertheless, if the original data related to 3D morphology (including angle of edges) is available, it would be more sufficient with more transparency and reproducibility. Additional description or schematic illustration for procedure and controls in 3D morphology would help understanding of readers.

Despite of this additional request, this paper is significant on establishing a novel methodology of lithic morphology in general, not in the specific field of European Middle Palaeolithic.

I am fully recommended to publish this paper.

As already pointed out, we can make fully available (as supplementary information) a simplified database of all the analysed records, including all the raw information (also edge angles) used to develop the whole statistics. On the other hand, it’s difficult to share original 3D shape files, as we are talking about nearly 200 models and 10 Gigabytes of data.

6. PLOS authors have the option to publish the peer review history of their article (what does this mean?). If published, this will include your full peer review and any attached files.

Do you want your identity to be public for this peer review? For information about this choice, including consent withdrawal, please see our Privacy Policy.

Reviewer #1: No

Reviewer #2: Yes: Atsushi Noguchi

---

## [Editor Report · Decision Letter 1]

10 Jul 2020

Techno-functional and 3D shape analysis applied for investigating the variability of backed tools in the Late Middle Paleolithic of Central Europe.

PONE-D-20-12918R1

Dear Dr. Delpiano,

We’re pleased to inform you that your manuscript has been judged scientifically suitable for publication and will be formally accepted for publication once it meets all outstanding technical requirements.

Kind regards,

Andrea Zerboni, Ph.D.

Academic Editor

PLOS ONE
---

## [Editor Report · Acceptance letter]

20 Jul 2020

PONE-D-20-12918R1 

Techno-functional and 3D shape analysis applied for investigating the variability of backed tools in the Late Middle Paleolithic of Central Europe. 

Dear Dr. Delpiano:

I'm pleased to inform you that your manuscript has been deemed suitable for publication in PLOS ONE. Congratulations! Your manuscript is now with our production department. 

Kind regards, 

on behalf of

Prof. Andrea Zerboni 

Academic Editor

PLOS ONE